# The oxic degradation of sedimentary organic matter 1400 Ma constrains atmospheric oxygen levels

Shuichang Zhang[1*], Xiaomei Wang[1], Huajian Wang[1], Emma U. Hammarlund[2], Jin Su[1], Yu Wang[1], Donald E. Canfield[2]

(1) Key Laboratory of Petroleum Geochemistry, Research Institute of Petroleum Exploration and Development, China National Petroleum Corporation, Beijing 100083, China, (2) Villum Investigator, Department of Biology and NordCEE, University of Southern Denmark, Campusvej 55, 5230 Odense M, Denmark

*Corresponding author: Email: sczhang@petrochina.com.cn, 86-10-83598360

**Abstract**

We studied sediments from the ca. 1400 million-year-old Xiamaling Formation from the northern China Block. The upper unit of this formation (unit 1) deposited mostly below storm wave base and contains alternating black and green-gray shales with very distinct geochemical characteristics. The black shales are enriched in redox-sensitive trace metals, have high concentrations of total organic carbon (TOC), high hydrogen index (HI) and iron speciation indicating deposition under anoxic conditions. In contrast, the green-gray shales show no trace metal enrichments, have low TOC, low HI and iron speciation consistent with an oxygenated depositional setting. Altogether, unit 1 displays alternations between oxic and anoxic depositional environments, driving differences in carbon preservation consistent with observations from the modern ocean. We combined our TOC and HI index results to calculate the differences in carbon mineralization and carbon preservation in comparing between the oxygenated and anoxic depositional environments. Through comparisons of these results with modern sedimentary environments, and by use of a simple diagenetic model, we conclude that the enhanced carbon mineralization under oxygenated conditions in unit 1 of the Xiamaling Formation required a minimum of 4% to 8% of present-day oxygen levels (PAL). These oxygen levels are higher than estimates based on chromium isotopes and reinforce the idea that the environment contained enough oxygen for animals long before their evolution.

**Keywords**: atmospheric oxygen, Mesoproterozoic, hydrogen index, carbon preservation, oxic respiration, anoxic, animal evolution, marine sediment

# 1    Introduction

The Mesoproterozoic Era (1600 to 1000 million years ago, Ma) was a time of profound biological transition. It witnessed the emergence of nascent eukaryote ecosystems, and more generally, it linked the dominantly prokaryote world of the Paleoproterozoic Era (2500 to 1600 Ma), and before, to the Neoproterozoic Era (1000 to 541 Ma), where eukaryotes greatly diverged and where animals first evolved (Butterfield, 2015; Knoll, 2014). In a widely held view, limited oxygen availability may have restricted the evolution and diversification of eukaryote clades, including animals, until a permissive environment emerged with a rise in oxygen levels in the late Neoproterozoic Era (Berkner and Marshall, 1965; Knoll, 2011, 2014; Nursall, 1959).

Unfortunately, there are few constraints on oxygen levels during the Mesoproterozoic Era. The idea that Mesoproterozoic oceans were largely anoxic below the surface mixed zone generated a model providing maximum oxygen concentrations in the range of 40% of present atmospheric levels (PAL) (Canfield, 1998). Subsequently, relatively low concentrations of redox-sensitive trace metals like uranium (U) and molybdenum (Mo) in Mesoproterozoic-aged black shales have reinforced the idea of wide-spread Mesoproterozoic ocean anoxia and of levels of atmospheric oxygen lower than today (Partin et al., 2013; Scott et al., 2008). Recently, chromium (Cr) associated with Mesoproterozoic-aged iron-enriched marine sediments has shown a lack of observable fractionation, suggesting no oxidative weathering of chromium minerals on land, and atmospheric oxygen levels of $\leq 0.1\%$ PAL (Planavsky et al., 2014). This idea was further reinforced by low Cr isotope fractionations preserved in Mesoproterozoic shales (Cole et al., 2016). In contrast, Cr associated with a number of 900 to 1100 Ma marine carbonates reveal highly fractionated isotopes consistent with oxidative weathering of Cr from land under elevated atmospheric oxygen concentrations (Gilleaudeau et al., 2016).  Thus, taken

at face value, Cr isotopes offer an equivocal view of the Mesoproterozoic history of atmospheric oxygen levels.

The chemistry of ancient soils, paleosols, offers other possible constraints on Mesoproterozoic Era atmospheric oxygen levels. Thus, Zbinden et al. (1988) modelled the retention and oxidation of iron during the weathering of the 1100 Ma Sturgeon Hills paleosol, developed on hydrothermally altered Keweenawan basalt, obtaining a minimum atmospheric oxygen concentration of 0.1% PAL. Other studies of the same paleosol profile, however, have not reproduced the same chemistry (Mitchell and Sheldon, 2010), indicating that further work on this paleosol is likely required.

Similarly aged paleosols developed on fluvium derived from weathered Keweenawan basalt (Mitchell and Sheldon, 2009). These paleosols formed in over-bank river sediments and adjacent pond environments that were alternatively flooded and air-exposed (Mitchell and Sheldon, 2009). The chemistry of these paleosols is thus complicated by an admixture of oxidative (during exposure) and reductive (during flooding) processes. When ratioed against the Ti content of unweathered Keweenawan basalt, Fe was apparently lost during paleosol formation, although Fe oxides are also prominent in the paleosols (Mitchell and Sheldon, 2009). In contrast, if paleosol chemistry is ratioed against Keweenawan basalt Al content, Fe was either fully retained or even enriched in the paleosols (Mitchell and Sheldon, 2009). Thus, interpretations of element loss or gain during weathering of these paleosols are highly dependent on the choice of the "immobile" element. Titanium is typically associated with dense mineral phases such as rutile ($TiO_2$) and ilmentite ($FeTiO_2$), and these minerals may undergo sorting during fluvial transport (Chen et al., 2013). Therefore, the magnitude of element mobility during the weathering of the Keweenawan basalt-derived fluvium is uncertain. Even with this uncertainty,

we are aware of no attempt to model atmospheric oxygen levels from the chemistry of these paleosols.

In a different approach, studies of unit 3 of the 1400 Ma Xiamaling Formation from the North China Block demonstrated trace metal and biomarker signatures consistent with deposition in oxygenated waters below an ancient oxygen-minimum zone (OMZ) (Zhang et al., 2016). A simple ocean water-column carbon-cycle model was constructed to determine the amount of atmospheric oxygen that would have been required to oxygenate these deep waters. Model results revealed a minimum value of $\geq 4\%$ PAL oxygen (Zhang et al., 2016).

Taken together, the studies described above show that available constraints on Mesoproterozoic levels of atmospheric oxygen are apparently contradictory. One could argue that the apparent uncertainty in Mesoproterozoic Era atmospheric oxygen levels actually reflects temporal variability. This is a valid point, but still, focusing on studies of the 1400 Ma Xiamaling Formation, and as mentioned above, shale Cr isotope results support low levels of atmospheric oxygen (Cole et al., 2016) of $< 0.1\%$ PAL, while modelling OMZ conditions in unit 3 of the Xiamling Formation revealed oxygen levels of $\geq 4\%$ PAL (Zhang et al., 2016). Clearly, other lines of evidence are required to constrain the evolution of Mesoproterozoic Era atmospheric oxygen and its role in biological evolution.

Here, we focus on evidence from unit 1 of the ca. 1400 Ma Xiamaling Formation. This unit shows transitions between sediment deposition under oxygenated and anoxic water-column conditions as revealed through trace metal systematics, iron speciation and organic geochemical results: in particular, total organic carbon contents (TOC) and the hydrogen index (HI). The results from the Xiamaling Formation are reviewed in the context of modern sediments, allowing

the construction of a simple diagenetic model that constrains atmospheric oxygen to a minimum of 4% to 8% PAL. These new results, thus, support elevated atmospheric oxygen levels at 1400 Ma.

## 2    Study site and methods

### 2.1    Study site

We explored rocks from unit 1 the Xiamaling Formation of the North China Block. The Xiamaling Formation is part of Paleoproterozoic to Mid-Mesoproterozoic sedimentary sequence, depositing onto Paleoproterozoic crystalline rocks that were likely formed during the breakup of supercontinent Columbia (Meng et al., 2011). The sedimentary sequence begins as an opening rift basin that developed into a passive margin and eventually a back-arc setting (Meng et al., 2011; Qu et al., 2014). The Xiamaling Formation itself contains relatively few volcanoclastic layers, and was first intruded by diabase sills at $1323 \pm 21$ Ma (Li et al., 2009). These intrusives are taken to indicate back-arc development, but they occurred some 60 to 70 million years after sediment deposition in units 2 and 3. This is evident from high-precision thermal ionization mass spectrometry (TIMS) dating yielding an age of $1384.4 \pm 1.4$ Ma for a tuff layer located at 210 m depth in unit 2, and an age of $1392.2 \pm 1.0$ Ma for a bentonite layer in unit 3, 52 m below the unit 2 tuff layer (Zhang et al., 2015). Thus, the Xiamaling Formation likely represented deposition in a passive-margin setting before later back-arc development (Meng et al., 2011; Qu et al., 2014).

Overall, the Xiamaling Formation has a total thickness of about 500 m, and is composed of highly-laminated sediments deposited, mostly, in deeper quiet waters below storm wave base (>100 m) through its history (Zhang et al., 2015). Paleogeographically, the Xiamaling Formation deposited in a tropical to sub-tropical setting between $10^o$ N and $30^o$ N latitude (Evans and

Mitchell, 2011; Zhang et al., 2012), and the patterns of sediment lamination and chemistry are consistent with the influence of climate forcing on sedimentation dynamics (Zhang et al., 2015). The sediments are also of exceptionally low thermal maturity, likely never heated to greater than 90°C (Zhang et al., 2015).

We focus here on the sediments deposited in unit 1. This unit is differentiated from underlying unit 2 by the first occurrence of TOC-poor green-gray shale layers, in a background of TOC-rich black shales (Figure 1) (Zhang et al., 2015). The green-gray shales become more prominent in moving up-section, and by 40 to 45 m depth in the stratigraphy, green-gray and black shales alternate regularly with individual layer thicknesses of 1 to several cm's. Sedimentation is continuous between the black and green-gray layers, and both represent fine-grained muddy silts. A deep-water setting is indicated through most of the unit 1, but at about 15 m depth, there appear occasional strata with hummocky-cross bedding indicating the influence of storm waves on deposition. From here, and upwards, sediments deposited at or above storm wave base, which can range in depth from about 50 to 200 meters (Immenhauser, 2009). Thus, overall, unit 1 likely deposited in waters in the depth range of $100 \pm 50$ meters, with deeper waters towards the bottom of the unit and shallower waters towards the top. There is no precise dating of unit 1 sediments.

Previous work placed unit 1 in the downwelling limb of an ancient Hadley Cell. It was argued that in this setting fluctuations in sediment chemistry resulted from periodic changes in Hadley Cell placement and the location of the Intertropical Convergence Zone (ITCZ) as these influenced patterns of trade wind intensity and ocean circulation (Zhang et al., 2015).

## 2.2    Sample collection and analytical methods

Both outcrop and core samples from the Xiamaling Formation were used in this study. Outcrop

samples were collected at ∼0.5m intervals along road cuts within 2 to 4 y after the outcrops were

exposed. Black shales and green-gray shales were easy to discriminate in outcrop, and all

samples were collected after removal of the weathered outer layer (see further, Zhang et al.,

2015). Core samples were collected using a diamond drill lubricated with fresh water to

minimize contamination from drilling fluids (see further, Zhang et al., 2016). Core depths were

correlated to outcrop height based on reconstructions from drilling depth and angle, and depth

were further cross calibrated against geochemical parameters such as trace element geochemistry

(e.g., Zhang et al., 2015; Zhang et al., 2016). For geochemical analyses, samples were rinsed

with purified water, dried, and then crushed to fine powder (less than 74 µm) using a stainless

steel puck mill, which was cleaned between samples by grinding with baked quartz sand multiple

times. All of the geochemical data were obtained from the homogeneous powder.

Trace metal concentrations were measured by ICP-MS following the methods outlined in

(Zhang et al., 2015). Accuracy and precision were tested with multiple runs of international

standards (GBW07309, GBW07310, GBW07312, GBW07104, GBW07106) that were included

with our sample runs (Table S1). With multiple analyses of each of these standards, the accepted

values for vanadium (V), molybdenum (Mo) and uranium (U) were all reproduced to within <

1%, and the standard deviation of individual analyses was in the range of 2.1% to 3.8% (Table

S1). For the outcrop samples, Al and Fe were determined by X-ray fluorescence following the

methods outlined in Zhang et al. (2015). Some of the total Fe data from the core samples were

also obtained by this method, but total Fe was also obtained with a hand-held XRF (HHXRF),

calibrated against a range of certified standards (see methods outlined in Zhang et al., 2016).

Overall, HHXRF Fe had a precision of about 1.5%, and an accuracy of > 95% when compared

with total Fe for an international standard (PACS-3) and a Xiamaling Formation samples whose Fe content was determined with traditional XRF (Table S1). In some instances, total Fe was also determined using the hot hydrochloric acid (HCl) boiling method of Aller et al. (1986). Repeated analyses (n=22) of the certified sediment standard PACS-2 (NRC) by this method showed a recovery of >95% of the total Fe. The different methods used for total Fe determinations reflect the evolution of total Fe methodology in the lab during the course of our Fe speciation data collection. Total Fe results are coupled to their respective analytical method in Table S3.

The hydrogen index (HI) expresses the amount of bound hydrocarbon-like compounds released during sample pyrolysis, ratioed against the total amount of organic carbon (TOC) in the sample (Espitalie et al., 1977). HI is defined as $S_2(mg/g_{rock})*100/wt\%TOC$, where $S_2$ represents a specific peak generated during pyrolysis that is generally assumed to comprise of the longer-chained, non-volatile hydrocarbons cracked and liberated from kerogen (Espitalié, 1986; Tissot and Welte, 1984). Pyrolysis was accomplished by programmed heating of samples in a Rock-Eval 6 instrument (Vinci Techologies, France), where the hydrocarbons were liberated and measured by flame ionization detection. The initial and final pyrolysis temperatures were 300 °C and 650 °C, respectively, and the programmed heating rate was 25 °C/min. Pyrolysis was conducted under a $N_2$ atmosphere and data were obtained and interpreted with the software ROCKINT. The instrument was calibrated using standard material [GBW (E) 070064∼070066]. The TOC for HI calculations were determined as described below. These HI analyses were previously reported in Zhang et al. (2015).

Isolation of kerogen involves successive removal of soluble organic matter (bitumen), mineral matter, and water from the shale, such that predominantly kerogen remains. To obtain kerogen, sediment powders were extracted for 72 h using a Soxhlet apparatus (9:1 v/v

DCM/MeOH) to remove soluble bitumen. Minerals in the sediments were then removed with the following procedure (Durand and Nicaise, 1980): 1) carbonates were dissolved by reacting with 6 M HCl at 60 to 70 °C for 1 to 2 h, 2) silicates were dissolved by reacting with a mixture of 40% HF and 6 M HCl (3:2 v/v) at 60 ~ 70 °C for 2 h, 3) newly formed fluorides were removed with 3% $HNO_3$. After each step, the samples were flushed with deionized water to remove soluble material. The kerogen was finally obtained as a coarse malleable mass after drying overnight at 90 °C. According to Durand and Nicaise (1980), the loss of kerogen from shales during this procedure is typically around 5.8% (n=45), where most of this loss is most likely from sample handling and not a result of chemical digestion (Durand and Nicaise, 1980).

The isotopic composition ($\delta^{13}C_{org}$) of dry kerogen carbon was measured with a Delta V Advantage mass spectrometer (Thermo Scientific Co. Ltd.) after the carbon was first combusted to carbon dioxide using a Flash EA 1112 HT. The mass spectrometer was standardized with NBS-18 ($\delta^{13}C$ = -5.014‰) and Chinese standards GBW04405 ($\delta^{13}C$ = 0.57‰) and GBW04407 ($\delta^{13}C$ = -22.40‰) with a relative standard deviation of 0.2‰ based on replicate analyses of the standards (n=5). Isotopic compositions are reported relative to the Pee Dee Belemnite (PDB).

The measurement of total organic carbon (TOC) on outcrop samples was performed at the Key Laboratory of Petroleum Geochemistry in China. All samples were powdered and de-carbonated (1M HCl for 2 h), and subsequently dried in an oven at < 40 °C. TOC concentrations were measured with a LECO CS-230HC carbon-sulfur analyzer after standardization with certified standard materials. Replicate analyses of standards gave a standard deviation of < 5%. TOC concentrations on core samples were determined after de-carbonation (same procedure as for outcrop samples) at the University of Southern Denmark, on a Thermo Analytical element

analyzer Flash 2000 after calibration against standard materials with and standard deviation of <

5%.

Iron speciation was performed on powders of samples collected from fresh core material.

The analytical method followed that of Poulton and Canfield (2005). In the Fe speciation

technique, four different pools of highly reactive iron (FeHR) are extracted from the sediment.

These are: carbonate associated iron (FeCARB; siderite and ankerite), ferric oxide and ferric

oxyhydroxide minerals (FeOX; ferrihydrite, lepidocrocite, goethite, hematite), magnetite

(FeMAG) and sulfidized iron, mainly pyrite (FePY). The concentrations of the non-sulfidized

iron pools were quantified after extraction by atomic adsorption spectroscopy (AAS), and the

analytical error for each iron extraction was less than 5% (as monitored through comparisons

with the internally calibrated extractions of NRC PACS-2 and PACS-3 sediment standards, n=5

per set of samples). Pyrite sulfur was extracted by chromium digestion, where the sulfide was

trapped as $Ag_2S$, and its concentration determined gravimetrically (Canfield et al., 1986; Zhabina

and Volkov, 1978). Replicate chromium digestions of the sediment standard NRC PACS-2

indicate an analytical error for evaluating pyrite iron contents of less than 9% (n=6 per set core

sample set). The FeHR is typically normalized to the total concentration of Fe in the sample (FeT;

the determination of which is described above.), yielding the ratio FeHR/FeT.

## 3    Results

Geochemical parameters for the outcrop samples are shown in Figure 2. Many of the

geochemical signals correlate with rock type and with TOC content (Figure 2 and Table 1). Thus,

the black shales show elevated TOC, HI, and ratios of Mo/Al, V/Al and U/Al, compared to the

green-gray shales, where the ratios of Mo/Al, V/Al are very near the crustal average values

(using crustal averages from (Rudnick, 2004)). In the black shales, the ratio of Fe/Al tends towards higher values (Figure 2, Table 1), particularly below 15 m in the stratigraphy (Figure 2). Iron speciation from the core materials shows that elevated ratios of FeHR/FeT are generally associated with samples containing high-TOC (Figure 3a). The results for TOC concentration, $\delta^{13}C$, HI and our trace metal analyses for the outcrop material are shown in Table S2, while Fe speciation results are shown in Table S3.

## 4      Discussion

### 4.1      Mesoproterozoic Era sedimentary organic matter

Much of the discussion to follow is based on patterns of organic carbon preservation as revealed in our geochemical data. Therefore, we begin with a short discussion of the nature of the Mesoproterozoic Era carbon cycle. Generally, the Mesoproterozoic Era saw the emergence of eukaryotic organisms (Javaux, 2011; Knoll, 2014), and by 1400 Ma there is compelling evidence for eukaryotic algae in marine ecosystems (Javaux, 2011; Knoll, 2014; Zhu et al., 2016). Still, fossil eukaryotes from the Mesoproterozoic Era are rare, and there is little well-verified biomarker evidence for marine eukaryotes at or prior to 1400 Ma. Therefore, while eukaryotes, including algae, likely populated marine ecosystems by 1400 Ma, there is little evidence that they were a major part of the carbon cycle (Brocks and Banfield 2009). Rather, the carbon cycle was likely dominated by prokaryotic organisms, with cyanobacteria as the most important primary producers. Indeed, by 1400 Ma the fossil record reveals a variety of cyanobacterial forms ranging from single coccoidal cells and coccoid colonies, to multicellular filaments (e.g., Golubic and Seong-Joo, 1999). In addition to cyanobacteria, anoxic water- column settings also supported anoxygenic phototrophic bacteria living off the oxidation of chemically-reduced species such as $Fe^{2+}$ and $H_2S$ (Brocks et al., 2005; Zhang et al., 2016). The carbon cycle would

have also included the myriads of heterotrophic and autotrophic prokaryotes involved in elemental cycling.

From a biochemical perspective, prokaryotes, including cyanobacteria, are composed primarily of carbohydrates, lipids, and proteins, just as eukaryotic algae, although in different proportions (e.g., Hedges et al., 2002; Mouginot et al., 2015). Indeed, the biggest difference in chemical composition among photosynthetic organisms is between cyanobacteria and algae, on one hand, and land plants, on the other, where land plants contain significant proportions of lignin and cellulose. These compound classes have very different elemental stoichiometries than aquatic phototrophs (Sterner and Elser, 2002), and are much more resistant to diagenetic decomposition (e.g., Cowie et al., 1992). However, terrestrial land plants emerged around a billion years after the deposition of the Xiamaling Formation, and therefore, would not have influenced the Mesoproterozoic Era carbon cycle.

The carbon cycle of the Mesoproterozoic Era produced sedimentary organic carbon concentrations ranging from very low, nearly undetectable, to 20 wt% or more (e.g., Cox et al., 2016; Strauss et al., 1992; Zhang et al., 2015; Zhang et al., 2016) very similar to the range observed in modern sediments (e.g., Jahnke, 1996). However, in comparing organic carbon concentrations in modern and Mesoproterozoic Era sediments, one must consider the possibility that low concentrations of atmospheric oxygen could have inhibited the weathering of sedimentary organic carbon on land (e.g., Bolton et al., 2006; Daines et al., 2017), thus providing elevated concentrations of recycled ancient organic matter to marine sediments. There is no evidence for a significant contribution of recycled organic matter to Xiamaling unit 1 sediments as Rock-Eval analysis of both low and high-TOC samples from unit 1 of the Xiamaling Formation produced similar Tmax values in the range of 430 to 440 $^{\circ}$C (Zhang et al., 2015). This

range of Tmax values is characteristic of immature to early mature organic matter just entering the oil production window (Espitalié, 1986). In contrast, one would expect much higher maturity, and Tmax values, for recycled organic matter having experienced many cycles of deposition, burial and weathering. Therefore, there is no evidence for the recycling of ancient continental organic matter into unit 1 Xiamaling Formation sediments. Overall, organic matter cycling during the Mesoproterozoic Era appears to reflect processes and dynamics that we can relate to modern marine environments.

## 4.2    Water column chemistry

Our geochemical data reveal fluctuating water column conditions during the deposition of unit 1. Thus, as mentioned above, the black shales are enriched in TOC compared to the green-gray shales (Figure 2; Table 1). The black shales are also enriched in all of the redox-sensitive trace metals V, Mo and U (Figure 2; Table 1) compared to both the green-gray shales and compared to crustal average values. Enrichments in these trace metals, and TOC, are typical for deposition under anoxic water-column conditions (Algeo and Rowe, 2012). There is also some indication of Fe enrichment (as expressed through the Fe/Al ratio) in the black shales (Figure 2, Table 1) compared to the green-gray shales. Such enrichments are indicative of the water-column mobilization of Fe and its deposition under anoxic bottom- water conditions (Lyons and Severmann, 2006). Therefore, trace metal results, and patterns in Fe/Al ratios, are fully consistent with black shale deposition under an anoxic water column. In contrast, the lack of enrichment in trace metals during in the green-gray sediments are consistent with deposition under oxygenated bottom waters (e.g., Piper and Calvert, 2009; Tribovillard et al., 2006). Trace metal concentrations, however, also typically correlate with TOC concentration (e.g., Algeo and

Lyons, 2006; Tribovillard et al., 2006). Therefore, the low TOC content of the green-gray shales could partially explain the low trace metal abundance in these shales, and alternative geochemical indicators of bottom-water chemistry would strengthen our geochemical interpretations.

Sequential Fe extraction results offer another assessment of water column chemistry. Indeed, sequential Fe extractions have become a standard tool for evaluating bottom water chemistry during sediment deposition (Poulton and Canfield, 2005b; Raiswell and Canfield, 2012, 1998). Thus, from a compilation of data from modern environments, the ratio of highly reactive iron over total iron (FeHR/FeT) rarely exceeds 0.38 during deposition in oxygenated marine waters (Raiswell and Canfield, 1998). In contrast, when FeHR/FeT values exceed 0.38, this indicates sediment deposition below anoxic water columns, both in modern and ancient depositional settings (Poulton and Raiswell, 2002; Raiswell and Canfield, 1998).

Our Fe extractions were performed on fresh core material where it was not always easy to distinguish between black shales and green-gray shales, as was straightforward in the outcrop samples. Therefore, we have organized our extraction results as a function TOC concentration (Figure 3), recalling that in outcrop, TOC concentrations of > 2 wt% easily distinguished black shales, whereas the green-gray shales were easily distinguished at TOC values of mostly < 0.5 wt% (Figure 2, Table S2). We collected very few samples from the outcrop with TOC concentrations of between 0.5 and 2 wt%, so the shale color (type) in this TOC range is uncertain.

From our Fe extraction results, the ratio FeHR/FeT exceeds 0.38 for sediments with TOC exceeding 2 wt% (Figure 3a). As mentioned above, FeHR/FeT values of greater than 0.38 indicate sediment deposition under bottom-water anoxia (Raiswell and Canfield, 2012, 1998).

These results, therefore, reinforce our conclusions from trace metal dynamics that the black shales of unit 1 deposited in anoxic waters. In addition, euxinic (sulfidic) water-column conditions are indicated when FePY/FeHR > 0.7-0.8 for sediments deposited in anoxic waters, and when FePY/FeHR < 0.7-0.8, ferruginous conditions are indicated (Raiswell and Canfield, 2012). The chemical nature of anoxic deposition is not a focus here, but from the data in Table S3, it is clear that the anoxic waters of unit 1 contained a mixture of euxinic and ferruginous chemistry.

Unlike the high-TOC sediments, those with low-TOC, particularly with TOC concentrations of < 0.5wt%, have FeHR/FeT values of less than 0.38. These FeHR/FeT values are compatible with sediment deposition under oxygenated bottom waters, providing further evidence, in addition to the trace metals, that the green-gray shales deposited under oxygenated bottom water conditions. As we will see below, our assessments of bottom water chemistry during unit 1 deposition is compatible with additional organic geochemical constraints.

**4.3    HI and organic carbon preservation**

Hydrogen index (HI) is often used to assess organic matter maturity and state of organic matter preservation. Thus, high HI values are associated with better preserved organic matter with lower maturity, while low values of HI are associated with poorly preserved and organic mature with organic matter of high maturity (Espitalie et al., 1977; Tissot and Welte, 1984). From the outcrop materials, the HI is considerably higher in the black shales compared to the green-gray shales (Table 1, Figure 2, 3c), and overall, HI correlates with TOC concentration (Figure 3c). Despite these differences, the green-gray and black shales share similar organic matter $\delta^{13}C$ values (Figure 1) consistent with a similar source of organic carbon for each sediment type. As

mentioned above, high degrees of thermal maturity can reduce the HI (Espitalie et al., 1977; Tissot and Welte, 1984), but unit 1 sediments have all experienced the same thermal history, so this cannot account for differences in the HI between the different sediment types.

The magnitude of the HI has often been linked with the degree of organic carbon preservation. In general, a higher HI is associated with more H-rich aliphatic organic matter and better organic matter preservation, while low HI is associated with poorer organic matter preservation (Espitalie et al., 1977; Tissot and Welte, 1984). In Phanerozoic-aged examples, alternations in bottom water oxygenation have been argued to explain stratigraphically controlled differences in TOC and HI; similar differences to those observed in unit 1 of the Xiamaling Formation. Thus, in one example, TOC-rich, laminated to micro-burrowed shales from the Cretaceous Greenland Formation deposited with high HI, whereas moderately-to-highly bioturbated low-TOC shales deposited with low HI (Pratt, 1984). Palynological and organic geochemical analyses revealed a limited contribution of terrestrial organic matter to all sediment types, and differences in HI were attributed to the influence of oxygen on organic carbon preservation. In particular, oxygen was much more available to the bioturbated sediments compared to the laminated and micro-burrowed sediments, and more oxygen availability resulted in more extensive organic matter decomposition (Pratt, 1984), yielding both lower TOC and lower HI values. In another example, careful palynological and organic-geochemical analyses from the Upper Jurassic Kashpir shales of the Volga Basin, Russia, revealed that TOC-poor low-HI sediments were most likely associated with intensive oxic organic matter decomposition, whereas TOC-rich high-HI sediments were likely deposited in a continuously anoxic environment (Riboulleau et al., 2003).

In addition, alternating black and green claystone sequences from Cretaceous-aged sediments of the proto-North Atlantic (Kuypers et al., 2004) displayed dynamics in TOC and HI that are highly reminiscent of those from unit 1 of the Xiamaling Formation. Thus, the black shales contained organic matter predominantly of marine origin, and biomarker evidence demonstrated the presence of sulfide-oxidizing phototrophs in the water column (Kuypers et al., 2002). This evidence, coupled with trace metal enrichments in the black claystones, demonstrated water-column anoxia during black claystone deposition. The green claystones, some of which were heavily bioturbated, and thus clearly deposited in oxygenated waters, had low concentrations of TOC and low HI values. In addition, biomarker evidence showed an enhanced contribution of relatively refractory biomarkers such as n-alkanes in the green claystones, compared to the black claystones, where more labile hopanoids and steroids were much more abundant (Kuypers et al., 2002). These biomarker patterns were argued to reflect greater oxygen exposure times and more extensive organic matter decomposition in the green vs black claystones (Kuypers et al., 2002). Thus, at least in part, differences in HI and TOC between the green and black claystones reflected differences in carbon preservation as controlled by oxygen availability. Some of the low HI in the green claystones may have resulted from a relatively higher contribution of terrestrial organic matter to these sediments (Kuypers et al., 2002). But, the high terrestrial organic matter contribution in the green claystones was likely only evident due to extensive decomposition of the more labile marine organic carbon pool by oxygen.

There are also examples where relationships between HI and oxygen availability are not so straightforward. For example, in surface sediments of the eastern Arabian Sea, an intense oxygen minimum zone (OMZ) impinges on the sediment surface at water depths between about

100 and 700 m, with oxygenated water above and below (Naqvi et al., 2005). In these sediments, HI does not correlate with TOC, and HI values are equally high in sediments deposited in the OMZ and those deposited in oxygenated waters above and below the OMZ (Calvert et al., 1995). In this case, organic carbon dynamics are heavily affected by the sorting associated with active water currents and hydrodynamic processes (Cowie, 2005; Cowie et al., 2014). But even here, patterns of biomarker preservation and other indices of organic matter preservation suggest that organic matter is more heavily degraded under well-oxygenated conditions compared to low-oxygen to anoxic conditions in the heart of the OMZ (Cowie et al., 2014; Damste et al., 2002).

Returning to the Xiamaling Formation, and in reference to the studies mentioned above, the patterns of HI in unit 1 Xiamaling sediments indicate enhanced organic matter preservation in the high-TOC black shales compared to the low-TOC green-gray shales. Thus, patterns of TOC and HI in Xiamaling unit 1 are best understood in terms of differences in carbon preservation as driven by the presence or absence of oxygen during sediment organic matter mineralization. In this way, fluctuations between TOC-rich black shales and TOC-poor shale resulted from fluctuations between anoxic and oxic depositional conditions. This conclusion is completely compatible with, and indeed supports, the patterns of bottom-water oxygenation as revealed from trace metal and Fe speciation results as discussed above.

## 4.4 Modern studies of organic carbon preservation

The relationships between oxygen availability and organic carbon preservation as explored above are completely consistent with experimental observations of decomposing organic matter. Thus, in one study, decomposition experiments were conducted on algae that was pre-decomposed for 40 days (to about one half of its initial biomass and thus representing the type of

"aged" organic matter that deposits onto shelf sediments). The study found that organic matter in the presence of oxygen decomposed at rates 5 to 10 times greater than organic matter decomposed anaerobically (Kristensen and Holmer, 2001). These experiments were not continued until all of the labile organic matter was exhausted (this would have taken many years), but the results strongly indicate enhanced organic matter preservation under anoxic conditions compared to its preservation in the presence of oxygen.

These experimental results are further supported by the observations of organic carbon preservation in modern marine sediments (Canfield, 1994; Hartnett et al., 1998). Thus, in one approach, organic carbon preservation was compiled for marine sediments across a wide range of sedimentary environments, from continental shelf to the deep sea, and for oxygenated, low-oxygen, and fully anoxic bottom-water conditions (Canfield, 1994). Here, carbon preservation (%) is defined as:

$$\%C_{pres} = 100*C_{bur}/C_{dep} = 100*C_{bur}/(C_{bur} + C_{resp}) \qquad (1)$$

Where $\%C_{pres}$ is the percent of organic matter falling onto the sediment surface that is buried and preserved, $C_{bur}$ is the burial flux of organic carbon, and $C_{dep}$ is the flux of organic carbon depositing onto the sediment surface. For practical reasons, $C_{dep}$ is usually determined as the sum of the organic carbon burial flux ($C_{bur}$) and the rate of organic carbon respiration ($C_{resp}$), as determined, for example, by oxygen and/or $CO_2$ flux measurements (Canfield, 1994, 1989). This compilation is shown in Figure 4a, and we see that at the same rate of sedimentation (and for sedimentation rates <0.1 g cm$^{-2}$ y$^{-1}$), sediments depositing in anoxic and low-oxygen environments preserve considerably more organic carbon compared to sediments depositing in oxygenated environments.

In another approach, the degree of organic carbon preservation (as derived in equation 1) were related to amount of time the surface sediments were exposed to oxygen, the so-called oxygen exposure time (Hartnett et al., 1998). The oxygen exposure time ($O_2$-exp) is calculated from the depth of oxygen penetration into the sediment ($O_2$-pen) and the linear sedimentation rate (Linear rate)

$$O_2\text{-exp (y)} = O_2\text{-pen (cm)} / \text{Linear rate (cm y}^{-1}) \qquad (2)$$

In the original publication by Hartnett et al. (1998), calculations of oxygen exposure times were mostly based on calculated oxygen penetration depths. In this calculation, oxygen penetration was derived from a simple model where measured rates of sediment oxygen uptake were assumed to be driven by a linear decrease in oxygen concentration in the sediment. Normally, however, oxygen will penetrate much deeper than a linear gradient derived from the sediment surface would indicate (e.g., Glud, 2008). For this reason, we have compiled our own database (Table S4), that relies on actual measurements of oxygen penetration depth and for which carbon preservation (burial efficiency) is also calculated. Our compilation includes data from many parts of the global ocean and is summarized in graph form in Figure 4b. Consistent with Hartnett et al. ( 1998), however, lower carbon preservation accompanies greater oxygen exposure times. As mentioned above, this idea is consistent with the experimental observations of enhanced organic matter mineralization in the presence of oxygen (Kristensen and Holmer, 2001) and the observations of carbon preservation from Figure 4a as discussed above.

## 4.5 Organic carbon decomposition in the sediment and water column

Our geochemical results indicate enhanced oxic organic matter decomposition during the deposition of the green-gray shales compared to black shales in unit 1. As outlined above, the HI

is much reduced in the green-gray shales compared to the black shales. Furthermore, the TOC concentration averages 0.29 wt% in the green-gray shales compared to 3.1 wt% for the black shales, just over 10 times reduced (Table 1, Figure 2c). In the modelling that follows, we convert these trends in carbon preservation to sediment organic matter mineralization rates, and from here, to the minimum levels of atmospheric oxygen needed to drive these rates of mineralization. Therefore, to provide the best sediment model for carbon mineralization, we must evaluate the comparative histories of organic matter decomposition under oxic and anoxic conditions from the water column to the sediment.

Beginning in the water column, Keil et al. (2015) provides one of the few studies to compare organic carbon transport through oxic and anoxic waters. In this study, Keil et al. (2015) explored sediment traps the composition of particles settling through waters of the Arabian Sea. In two stations, the water column was nitrite-containing and completely anoxic between 130-150 meters to > 500 meters water depth. In contrast, at a third site, the water also became anoxic at about 150 meters depth, but oxygen began to accumulate at about 200 meters water depth, below a narrow anoxic zone of some 50 meters depth. In sediment traps at 500 meters depth, TOC averaged about 11 wt% for particles settling through oxygenated waters and 15 wt% for particles settling through anoxic OMZ waters (Keil et al., 2015). Thus, in the Arabian Sea, there is a relatively small difference (27%) in the carbon content of particles settling through oxic and anoxic waters to 500 meters depth.

This difference could relate to differences in the relative efficiencies of oxic vs anoxic mineralization, or to differences in the initial composition of the particles originating at the different sites. If oxic vs anoxic decomposition is the main factor driving these TOC differences, then the differences would likely be even smaller for particles settling to the shallower water

depths of 50 to 200 meters as we surmise for unit 1 of the Xiamaling Formation. The Arabian

Sea results also reinforce a general observation that throughout the global ocean, particles

settling though the upper 100s of meters of the marine water column are quite TOC-enriched,

with values much closer (typically 3 to 15 wt%; Armstrong et al., 2002; Honjo et al., 1982) to

those observed in the black shales of unit 1 of the Xiamaling Formation than to those observed in

the green-gray shales (Table 1). Overall, we argue that the differences in the TOC content

between the green-gray and black shales in unit 1 were likely driven mostly by differences in

sediment organic carbon preservation, as controlled by the presence or absence of bottom-water

oxygen, and not by differences in water column processes. This assessment is based on: 1) the

relatively small differences in the TOC content of particles settling through oxic and anoxic

waters of the Arabian Sea, and 2) the observation that the green-gray shales of unit 1 have TOC

contents much reduced compared to particles settling through the upper 100s of meters of the

modern marine water column.

## 4.6    Constraining oxygen levels

Our goal now is to determine the levels of bottom-water oxygen required to account for the

differences in carbon preservation between the green-gray and black shales of unit 1, which we

assume, from the discussion above, to be a factor of 10 (although we also relax this assumption

in the modelling that follows). Our model is constrained from modern observations through a

multi-step process. Our first step is to revisit the observation that organic carbon preservation in

modern marine sediments is controlled by both sedimentation rate and sedimentary environment

as shown in Figure 4a. To utilize the trends In Figure 4a, we must first estimate the rate of

sediment deposition for Xiamaling Formation unit 1. From precise zircon dating, we previously

determined an average linear (after compaction and lithification) sedimentation rate of $6.7 \times 10^{-4}$

cm $y^{-1}$ for a 52 meter section encompassing upper unit 3 into lower unit 2 of the Xiamaling

Formation (Zhang et al., 2015). This linear sedimentation rate translates into a mass

accumulation rate of $1.7 \times 10^{-3}$ g $cm^{-2}$ $y^{-1}$, assuming an average rock density of 2.5 g $cm^{-3}$ (a

density similar to quartz at 2.65 g $cm^{-3}$ and typical for marine sediments). We call this the XML

rate. We cannot be certain that this rate applies to unit 1, which is undated, but to accommodate

this uncertainty, we will also consider sedimentation rates of one half of the XML rate and 10

times greater than this rate.

To demonstrate our approach, we begin with a sedimentation rate consistent with the

XML rate ($1.7 \times 10^{-3}$ g $cm^{-2}$ $y^{-1}$). In modern anoxic environments, sediments at this

sedimentation rate experience carbon preservation of between about 20% and 30% as seen by

extrapolating between existing data points in Figure 4a. This degree of carbon preservation

would be relevant for the black shales of unit 1 of the Xiamaling Formation. With a factor of 10

times lower organic carbon preservation (reflecting a 10 times lower TOC content) for the green-

gray shales, the carbon preservation ranges from between 2% and 3% (see also Figure 4a).

Similar calculations for 10 times the XML sedimentation rate, and ½ this rate, are shown in

Table 2. Our derived organic carbon preservation percentages for the green-gray shales compare

reasonably well with observations from modern oxic marine environments (Figure 4a).

From here, we determine the amount of oxygen exposure that sediments require to

achieve the degrees of carbon preservation we have determined. These oxygen exposure times

($O_2$-exp) are obtained from the compilation in Figure 4b, where we utilize the full range of

oxygen exposure times at a given percentage of carbon preservation as determined in modern

environments. These oxygen exposure times are summarized in Table 2.

With the oxygen exposure times we have determined, we can now calculate the depth of oxygen penetration necessary to generate these exposure times. To do this, we must also know the linear sedimentation rate (Linear rate) which is related to the mass sedimentation rate (mass rate) through the following expression:

$$\text{Linear rate (cm y}^{-1}) = \text{mass rate (g cm}^{-2}\text{ y}^{-1})*[1/(1-\phi)*\rho \text{ (g/cm}^3)] \qquad (3)$$

where $\phi$ is sediment porosity and $\rho$ is dry sediment density. Sediment porosity and dry density are often measured, but rarely reported and not compiled, to our knowledge, for surface muds (top few cm's). As mentioned above, and from our experience, a value of 2.5 g cm$^{-3}$ is a good approximation for the dry density of sediment particles. Surface sediment porosities can vary, and in our experience, a range of 0.7 to 0.9 for the upper couple of centimeters encompasses our dozens of observations from a variety of different marine muds. Exceptions include organic-rich euxinic sediments that can have porosities of > 0.95, and sands which have porosities in the range of 0.4 to 0.5. The Xiamaling Formation sediments of unit 1, however, are relatively fine-grained silty muds, so comparisons with sands are inappropriate. In our calculations, we explore a range of porosities from 0.7 to 0.9. The oxygen penetration depth ($O_2$-pen) is determined by rearranging Equation 2 and is given as:

$$O_2\text{-pen (cm)} = \text{Linear rate (cm y}^{-1})*O_2\text{-exp (y)} \qquad (4)$$

Calculations of oxygen penetration depths at different porosities and at different oxygen exposure times for the XML sedimentation rate are provided in Table 3. From here, we determine how much bottom water oxygen ($[O_2]_{BW}$) is required to generate the oxygen penetration depths ($O_2$-pen) we have calculated. Generally, oxygen penetration will depend on the concentration of bottom water oxygen, the rate of sediment oxygen uptake, and the kinetics

of organic carbon mineralization including any oxygen dependency on mineralization and the depth-distribution of organic matter quality. Numerous models have been proposed relating these parameters (e.g., Hartnett et al., 1998; Hulth et al., 1994; Rasmussen and Jørgensen, 1992), and we will build on the simple model proposed by Rasmussen and Jørgensen (1992), and shown in Equation 5, rearranged to yield $[O_2]_{BW}$ .

$$[O_2]_{BW} = O_2\text{-pen}*O_2\text{-flux}/A*\phi D_{sed} \tag{5}$$

where, in addition to the terms already named, $O_2$-flux (mmole cm$^{-2}$ y$^{-1}$) is the flux of oxygen into the sediment, $D_{sed}$ (cm$^2$ y$^{-1}$) is the sediment diffusion coefficient for oxygen approximated as $D*\phi^2$ (Ullman and Aller, 1982), where D is the free diffusion coefficient, which we take as 536 cm$^2$ y$^{-1}$, the value for seawater at 15$^o$C (Broecker and Peng, 1982) and A is a variable that we will explore below.

When A=1, Equation 5 is consistent with a linear oxygen profile in the sediment as assumed, for example, by Hartnett et al. (1998). This formulation represents sediments with a source of oxygen from the overlying water and a fixed sink at the depth of oxygen penetration, but no oxygen removal in between. A value of A=2 generates an equation consistent with a constant rate of oxygen removal with depth in the sediment, but no dependency of oxygen removal rate on oxygen concentration (thus zero-order reaction kinetics on oxygen concentration) as developed in Rasmussen and Jørgensen (1992). Other values for A may also be chosen as explored below.

To evaluate these equations, and to determine most appropriate value for A, we have compiled a database on the relationship between oxygen uptake rate and oxygen penetration depth for a broad range of marine sediments depositing in a range of different bottom water

oxygen concentrations (Figure 4c; data in Table S5; note that some of this data is also presented in Table S4). We see that a value of A=2 clearly underestimates the oxygen penetration depth at a given rate of oxygen uptake (and A=1 is even worse; not shown). The data, however, are consistent with a value of A=4, which generates a relationship between oxygen penetration depth and oxygen uptake rate very similar to the best-fit power function of the data ($O_2$ uptake = $0.203O_2$-pen$^{-0.868}$; $R^2$=0.7526). With the value of A=4, Equation 5 also seems to work through the range of oxygen concentrations explored in Figure 4c. Therefore, in subsequent modelling as described below, we will use Eq. 5 with a value of A=4 and consider it reliable for the range of oxygen concentrations explored. We note, however, that with a value of A=4, Equation 5 is simply an empirical fit of the data in Figure 4c and is not based on first-principle diagenetic relationships as is true when A=1 and when A=2.

Our next step is to determine rates of sediment oxygen uptake ($O_2$-flux, mmol cm$^{-2}$ y$^{-1}$) for the Xiamaling Formation green-gray shales for each of our model scenarios. In modern sediments, the oxygen uptake rate is approximately equivalent to the total carbon mineralization rate ($C_{resp}$; see Equation 1) as described, for example, in Canfield et al. (1993). The equivalency arises because oxygen is also used to oxidize the reduced products of anaerobic mineralization. Therefore, from rearranging Equation 1, we can isolate $C_{resp}$:

$$O_2\text{-flux} = C_{resp} = (100*C_{bur}/\%C_{pres}) - C_{bur} \qquad (6)$$

The values for $C_{resp}$ (and thus $O_2$-flux) obtained this way are internally consistent mass-balance values. $O_2$-flux results are shown in Table 4 for the different modeling scenarios.

With our calculations of $O_2$-pen and $O_2$-flux, we use the Equation 5 with a value of A=4 to calculate bottom water oxygen levels at different sediment porosities. The results are shown in

Table 4. Our estimates for bottom water oxygen concentration vary widely, and are highest for scenarios with the longest sediment oxygen exposure times. Long oxygen exposure times accompany deep oxygen penetration (see Table 3), and higher concentrations of bottom water oxygen are required to balance the oxygen flux into the sediment against a small oxygen gradient as required by the deeper oxygen penetration. In many cases, our calculations with high oxygen exposure times yield bottom water oxygen concentrations that exceed modern values by a factor of 10 or more. One could view these as upper estimates for bottom water oxygen concentrations using our modeling approach, but this is not particularly useful, as such high oxygen concentrations during the Mesoproterozoic Era are very unlikely.

Just as our high estimates for bottom water oxygen concentrations are unrealistically high, our minimum estimates are probably also too low. However, as our goal here is to constrain minimum oxygen levels during Xiamaling Formation unit 1 deposition, we view these values as highly informative. Thus, for both the case of XML sedimentation rates and sedimentation rates one half of these (XML*0.5), a minimum estimate of 18 to 19 µM bottom water oxygen is obtained. These oxygen concentrations translate into an atmospheric oxygen concentration of 7% to 8% PAL assuming that the bottom water is in equilibrium with modern atmospheric oxygen at a temperature of $15^{o}C$ (yielding an equilibrium concentration of 250 µM). This calculation of atmospheric oxygen concentration does not account for any reduction in bottom water oxygen concentration that might have occurred due to respiration as particles settled through the oxic water column. Accommodating this oxygen loss would increase our atmospheric oxygen concentration estimates.

One potential criticism of our approach is that the factor of 10 difference in carbon preservation indicated between the black shales and the green-gray shales of unit 1 is an

overestimate. Thus, if the organic matter deposited onto the green-gray shales with a lower concentration than the black shales, then the difference in carbon preservation could be less than indicated by the difference in TOC concentration between the sediment types. As discussed above, we do not believe the differences would have been significant, but we still must entertain this possibility.

Thus, we have also calculated carbon preservation, oxygen exposure times, oxygen-penetration depths and, finally, estimates for bottom water oxygen for XML sedimentation rate and a factor of 5 difference in preservation between the black and green-gray shales (Figure 4a, Tables 2-4). These results yield a lower minimum bottom water oxygen concentration of 4.4 $\mu$M and about 2 % PAL atmospheric levels. This value, however, is likely too low for at least two reasons. First, these low bottom water oxygen estimates are accompanied by steep oxygen gradients and shallow oxygen penetration in the sediment (Table 3). In this case, one must also consider that oxygen is supplied to the sediment surface, and subsequently into the sediment, through a viscous boundary layer, which varies in thickness from 0.04 to 0.08 cm in continental margin sediments (Glud, 2008). Transport through this boundary layer is by molecular diffusion. Thus, strictly speaking, our oxygen estimate of 4.4 $\mu$M (Table 4) is the oxygen concentration at the sediment surface, below the viscous boundary layer. We calculate that a minimum $[O_2]_{BW}$ of between 10.1 $\mu$M (with a 0.04 cm boundary layer) and 15.7 $\mu$M (with a 0.08 cm boundary layer) is required to supply the 4.4 $\mu$M of oxygen to the sediment surface. These $[O_2]_{BW}$ are calculated from Equation 5 using the benthic boundary layer thickness for $O_2$-pen, A=1 (as would be true through a viscous boundary layer above the sediment), the free diffusion coefficient for oxygen and after adding the 4.4 $\mu$M oxygen concentration at the sediment surface. The bottom water oxygen concentrations of 10.1 to 15.7 $\mu$M transfer to atmospheric oxygen levels between 4% to

6% PAL. These should be considered the proper calculation values. A consideration of benthic boundary layer diffusion is not important for any of our other calculations.

Secondly, we note that low values of $[O_2]_{BW}$ in the range of even 10 μM are at odds with modern observations. Thus, when compared to anoxic settings, modern sediments depositing between the XML sedimentation rate and the XML*10 rate do not show enhanced degradation of organic matter for sediments depositing in oxygenated bottom water with < 25 μM $O_2$ (Figure 4a). Indeed, this observation alone might suggest that our higher bottom water oxygen estimates of 18 to 19 μM are also too low. Thus, while 10 μM to 19 μM (4% to 8% PAL) is the range of minimum bottom water oxygen concentrations produced by our model, modern observations suggest that this range may be too low and that bottom water oxygen levels of >25 μM, translating to atmospheric levels of 10% PAL, are a more realistic minimum estimate.

## 5.      Conclusions and Perspectives

We combined observations of trace metal dynamics, iron speciation, and TOC and HI dynamics to determine that unit 1 of the Xiamaling Formation experienced alternating periods of deposition in oxygenated and anoxic waters. The relationship between TOC and HI indicates substantial oxic mineralization of organic matter when sediments deposited in oxygenated water. We utilized observations from modern sediment organic matter dynamics to constrain the levels of atmospheric oxygenation required to generate the differences in organic matter preservation we observed between oxic and anoxic deposition in the Xiamaling Formation. Our modeling indicates minimum atmospheric oxygen levels at the time of Xiamaling unit 1 deposition of 4% to 8% PAL. Based on further observations from modern sediments, we believe that our estimate of 8% PAL is likely even too low.

Generally, our estimates of Mesoproterozoic atmospheric oxygen levels are consistent with the higher values of atmospheric oxygen ($\geq 4\%$ PAL) as constrained from ocean modeling (Zhang et al., 2016) while inconsistent with the lower levels of atmospheric oxygen ($\leq 0.1\%$ PAL) as constrained from chromium isotope systematics (Planavsky et al., 2014; Cole et al., 2016). We note, however, that the marine geochemistry of chromium, and its isotopes, are poorly known, and we have also previously documented concerns (Zhang et al., 2016) that the samples reported in the study of Planavsky et al. (2014) have a substantial, if not dominant, detrital chromium component. A strong detrital component would potentially compromise the interpretation of the chromium isotopes signal.

In any event, observations of low atmospheric oxygen concentration during this time (<0.1% PAL) do not square with the necessity of much higher oxygen levels to drive the sedimentary carbon dynamics that we observe in the Xiamaling Formation (Zhang et al., 2016). As noted above, we have previously reported evidence for minimum atmospheric oxygen levels of 4% PAL from unit 3 of the Xiamaling Formation (Zhang et al., 2016). While we do not have precise dating of unit 1 in the Xiamaling Formation, with a deposition rate from unit 2-3, the separation in time between unit 1 and 3 would be in the range of 20 to 25 million years. Therefore, relatively elevated levels of atmospheric oxygen appear to have been a persistent feature of the Mesoproterozoic geochemical environment for seemingly tens of millions of years. As noted previously (Zhang et al., 2016), these higher levels of atmospheric oxygen would have been sufficient to fuel animal respiration, at least at this time window in Earth history, and some 700 to 800 million years before animals first evolved.

**Data Availability**: All data used in this paper is provided in table form in the supplement.

**Team list**: Shuichang Zhang (SZ), Xiaomei Wang (XW), Huajian Wang (HW), Emma U. Hammarlund (EUH), Jin Su (JS), Yu Wang (YW), Donald E. Canfield (DEC).

**Author contribution**: SZ, XW, HW and DEC conceived of the project, SZ, DEC, XW, HW, EUH, JS, YW, did the research and SZ, XW, EUH and DEC wrote the paper.

**Competing interests**: The authors have no competing interested in this work.

**Acknowledgments**

We wish to thank Richard Boyle for discussions as well as Devon Cole and two anonymous reviews for very helpful comments. In addition, we wish to thank Heidi Jensen and Susanne Møller for expert laboratory assistance. We also acknowledge generous funding from the State Key Program of National Natural Science Foundation of China (41530317), the Scientific Research and Technological Development Project of China National Petroleum Corporation (CNPC 2016A-0204), the Danish National Research Foundation (Grant DNRF53), the ERC (Oxygen, grant 267233), the Danish Foundation for Basic Research (FNU) and the Villum Foundation.

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

Table 1. Averages for geochemical parameters in outcrop samples

| Shale type | Org-C wt% | Fe/Al | V/Al | Mo/Al | U/Al | HI mg/gTOC |
|---|---|---|---|---|---|---|
| black | 3.07±0.67 | 0.49±0.13 | 26.5±6.2 | 0.75±0.29 | 0.65±0.12 | 323±67 |
| green-gray | 0.29±0.17 | 0.28±0.11 | 12.1±2.0 | 0.16±0.09 | 0.43±0.06 | 113±56 |
| crustal ave.[a] | | 0.43 | 11.9 | 0.13 | | |

[a]from Rudnick (2004)

Table 2. Carbon preservation at various rates of sediment deposition

| scenario | Sed rate | %C pres | %C pres | $O_2$ exposure time |
|---|---|---|---|---|
| | g cm$^{-2}$y$^{-1}$ | black shale | gray shale | y |
| XML*0.5 | $0.8x10^{-3}$ | 12 | 1.2 | 700-6000 |
| XML | $1.7x10^{-3}$ | 20-30 | 2-3 | 400-5000 |
| XML*10 | $1.7x10^{-2}$ | 40-80 | 4-8 | 150-2000 |
| XML (factor 5) | $1.7x10^{-3}$ | 20-30 | 4-6 | 200-2000 |

Table 3. Linear sedimentation rates and oxygen penetration depths ($O_2$ pen) for the different mass fluxes explored in our modelling.

| XML*0.5 | | | |
|---|---|---|---|
| $O_2$ exposure time (y) | | 700 | 6000 |
| Porosity ($\phi$) | Sed rate | $O_2$ pen | $O_2$ pen |
| | cm y$^{-1}$ | cm | cm |
| 0.7 | $1.1 \times 10^{-3}$ | 0.77 | 7.7 |
| 0.8 | $1.7 \times 10^{-3}$ | 1.19 | 11.9 |
| 0.9 | $3.4 \times 10^{-3}$ | 2.38 | 23.8 |
| XML sed rate | | | |
| $O_2$ exposure time (y) | | 400 | 5000 |
| Porosity ($\phi$) | Sed rate | $O_2$ pen | $O_2$ pen |
| | cm y$^{-1}$ | cm | cm |
| 0.7 | $2.2 \times 10^{-3}$ | 0.88 | 11.0 |
| 0.8 | $3.4 \times 10^{-3}$ | 1.36 | 17.0 |
| 0.9 | $6.8 \times 10^{-3}$ | 2.72 | 34.0 |
| XML *10 | | | |
| $O_2$ exposure time (y) | | 150 | 2000 |
| Porosity ($\phi$) | Sed rate | $O_2$ pen | $O_2$ pen |
| | cm y$^{-1}$ | cm | cm |
| 0.7 | $2.2 \times 10^{-2}$ | 3.3 | 44 |
| 0.8 | $3.4 \times 10^{-2}$ | 5.1 | 68 |
| 0.9 | $6.8 \times 10^{-2}$ | 10.2 | 136 |
| XML (factor 5) | | | |
| $O_2$ exposure time (y) | | 200 | 2000 |
| Porosity ($\phi$) | Sed rate | $O_2$ pen | $O_2$ pen |
| | cm y$^{-1}$ | cm | cm |
| 0.7 | $2.2 \times 10^{-3}$ | 0.44 | 4.42 |
| 0.8 | $3.4 \times 10^{-3}$ | 0.68 | 6.8 |
| 0.9 | $6.8 \times 10^{-3}$ | 1.36 | 13.6 |

Table 4. Calculations of $[O_2]_{BW}$ (µM) for our different assumptions of sedimentation rate (lowest value for each sedimentation rate in **red**).

| XML*0.5 | | | | |
|---|---|---|---|---|
| Oxygen exposure (y) | 700 | 6000 | | |
| Carbon preservation (%) | 1.2 | 1.2 | | |
| $O_2$-flux (mmol cm$^{-2}$ y$^{-1}$) | 0.019 | 0.019 | | |
| | | | | |
| Porosity ($\phi$) | $[O_2]_{BW}$ | $[O_2]_{BW}$ | | |
| | µM | µM | | |
| 0.7 | **19** | 160 | | |
| 0.8 | 19 | 170 | | |
| 0.9 | 27 | 230 | | |
| **XML** | | | | |
| Oxygen exposure (y) | 400 | 5000 | 400 | 5000 |
| Carbon preservation (%) | 2 | 2 | 3 | 3 |
| $O_2$-flux (mmol cm$^{-2}$ y$^{-1}$) | 0.029 | 0.029 | 0.015 | 0.015 |
| | | | | |
| Porosity ($\phi$) | $[O_2]_{BW}$ | $[O_2]_{BW}$ | $[O_2]_{BW}$ | $[O_2]_{BW}$ |
| | µM | µM | µM | µM |
| 0.7 | 35 | 440 | **18** | 230 |
| 0.8 | 36 | 450 | 19 | 230 |
| 0.9 | 51 | 630 | 26 | 330 |
| **XML*10** | | | | |
| Oxygen exposure (y) | 150 | 2000 | 150 | 2000 |
| Carbon preservation (%) | 4 | 8 | 4 | 8 |
| $O_2$-flux (mmol cm$^{-2}$ y$^{-1}$) | 0.11 | 0.11 | 0.053 | 0.053 |
| | | | | |
| Porosity ($\phi$) | $[O_2]_{BW}$ | $[O_2]_{BW}$ | $[O_2]_{BW}$ | $[O_2]_{BW}$ |
| | µM | µM | µM | µM |
| 0.7 | 500 | 6600 | **240** | 3200 |
| 0.8 | 510 | 6800 | 250 | 3300 |
| 0.9 | 720 | 9600 | 350 | 4600 |
| | | | | |
| **XML (factor 5)** | | | | |
| Oxygen exposure (y) | 200 | 2000 | 200 | 2000 |
| Carbon preservation (%) | 4 | 4 | 6 | 6 |
| $O_2$-flux (mmol cm$^{-2}$ y$^{-1}$) | 0.011 | 0.011 | 0.073 | 0.073 |
| | | | | |
| Porosity ($\phi$) | $[O_2]_{BW}$ | $[O_2]_{BW}$ | $[O_2]_{BW}$ | $[O_2]_{BW}$ |
| | µM | µM | µM | µM |
| 0.7 | 6.6 | 66 | **4.4 (10.1-15.7)[a]** | 44 |
| 0.8 | 6.8 | 68 | 4.5 | 45 |
| 0.9 | 9.6 | 96 | 6.4 | 64 |

[a]values in parenthesis after considering diffusion through the benthic boundary layer

**Figure Captions:**

Figure 1. General stratigraphy for the upper 4 units of the Xiamaling Formation (abstracted from Zhang et al. 2015) with a more detailed stratigraphy for the upper 45 meters of unit 1.

Figure 2. Total organic carbon (TOC), HI, $\delta^{13}C$ (relative to PDB) and metal data (Mo/Al, V/Al and U/Al, Fe/Al for unit 1 of the Xiamaling Formation. The dashed line represents upper crust values from Rudnick (2004).

Figure 3. a) TOC vs the ratio of highly reactive to total iron (FeHR/FeT) from fresh core material in unit 1 of the Xiamaling Formation. The horizontal dashed line represents a FeHR/FeT of 0.38. The range of TOC values for green-gray shales from outcrop samples is shown in the green rectangular field, while the range in values for the black shales from outcrop is shown in the gray field. b) TOC vs HI for outcrop material, with black and green-gray shales separately indicated. The red dots mark the averages for the black and green-gray shale groups.

Figure 4. a) Preservation of organic carbon in modern marine sediments calculated as % of carbon buried in a sediment compared to the carbon deposited to the sediment surface. Redrafted from (Canfield, 1994).The vertical lines represent the different sedimentation rates used in the modelling. The upper red rectangles highlight the carbon preservation for the anoxic environments in the compilation, while the lower blue rectangles are 10 times less than this, representing the estimated range of carbon preservation in Xiamaling oxic sediments. For the XML sedimentation rate, a blue rectangle at 5 times less carbon preservation is also shown, b) Oxygen exposure time versus organic carbon preservation in marine sediments. The horizontal blue boxes reflect the range of oxic sediment carbon preservation at the different sedimentation rates from used in the modelling (see Figure 4a), while the dark blue fields outline the range of

associated oxygen exposure times. Oxygen exposure time data summarized in Table S4. c)

oxygen penetration depth versus oxygen uptake rate from modern marine sediments with

variable bottom-water oxygen concentrations. Data is from Table S5. Black line indicates the

best power-function fit to the data. Red lines indicate fits from Equation 4 to the data at different

bottom water oxygen concentrations and A=4. Green line represents model fit from Equation 4

with A=2 and 250 μM $O_2$.

Figure 1.

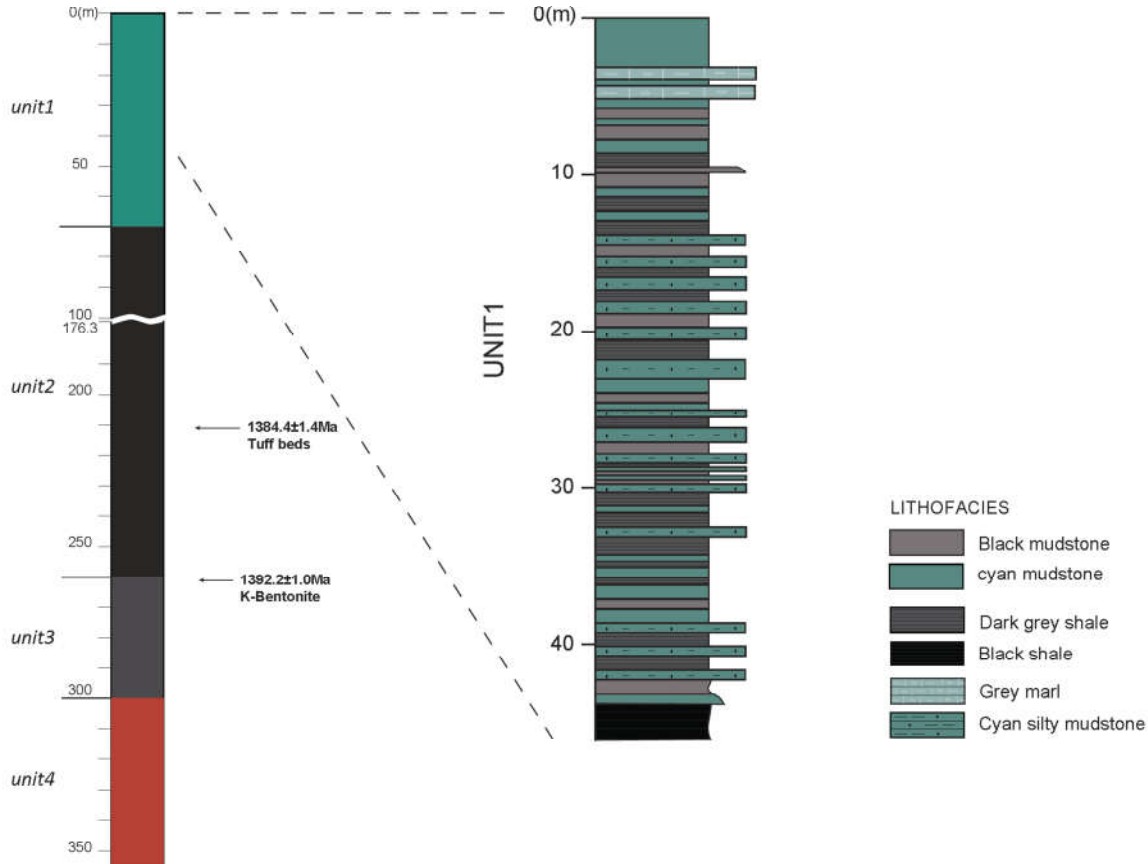

Figure 2.

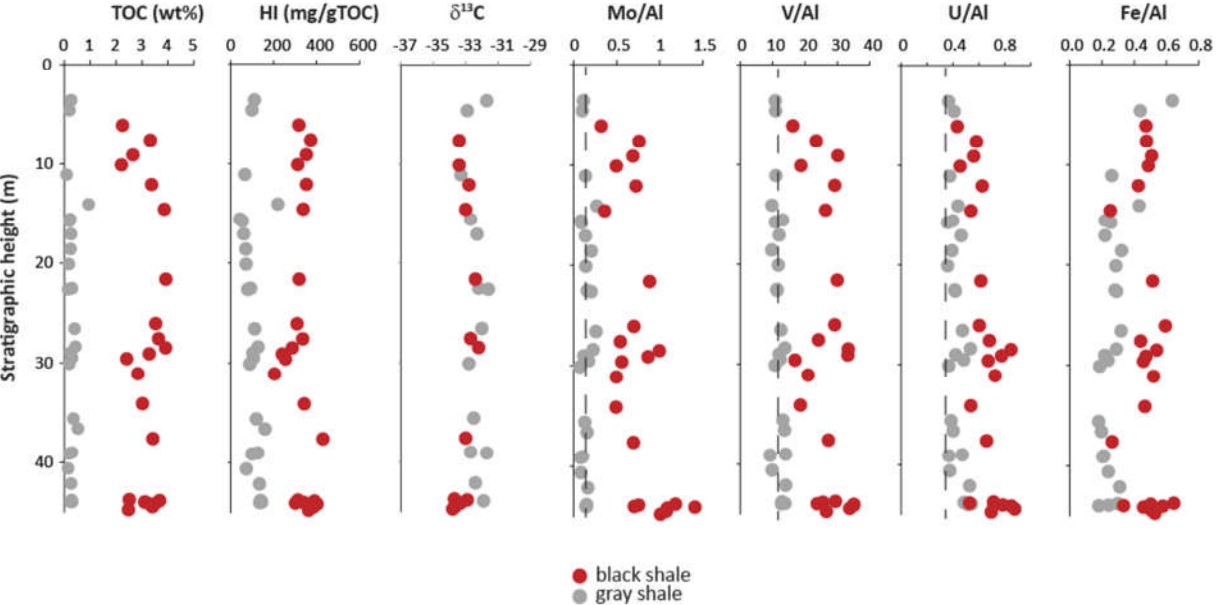

Figure 3.

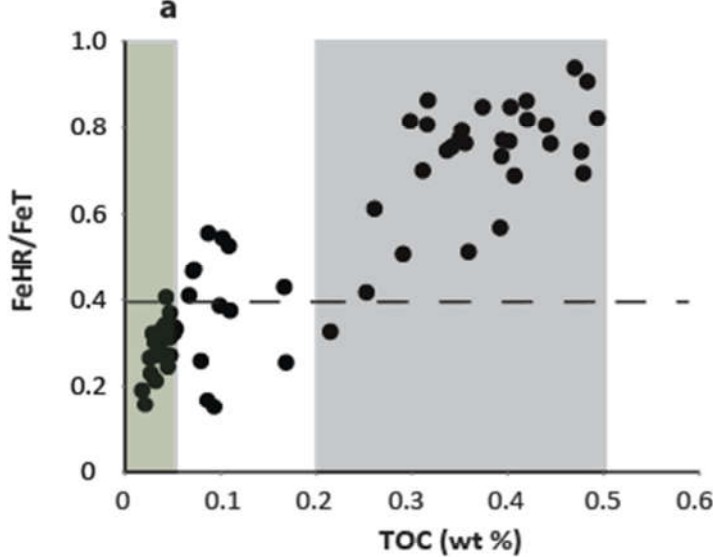

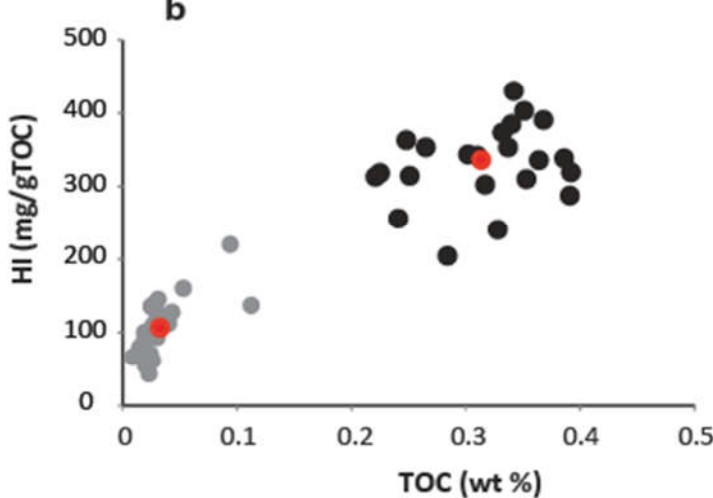

Figure 4.

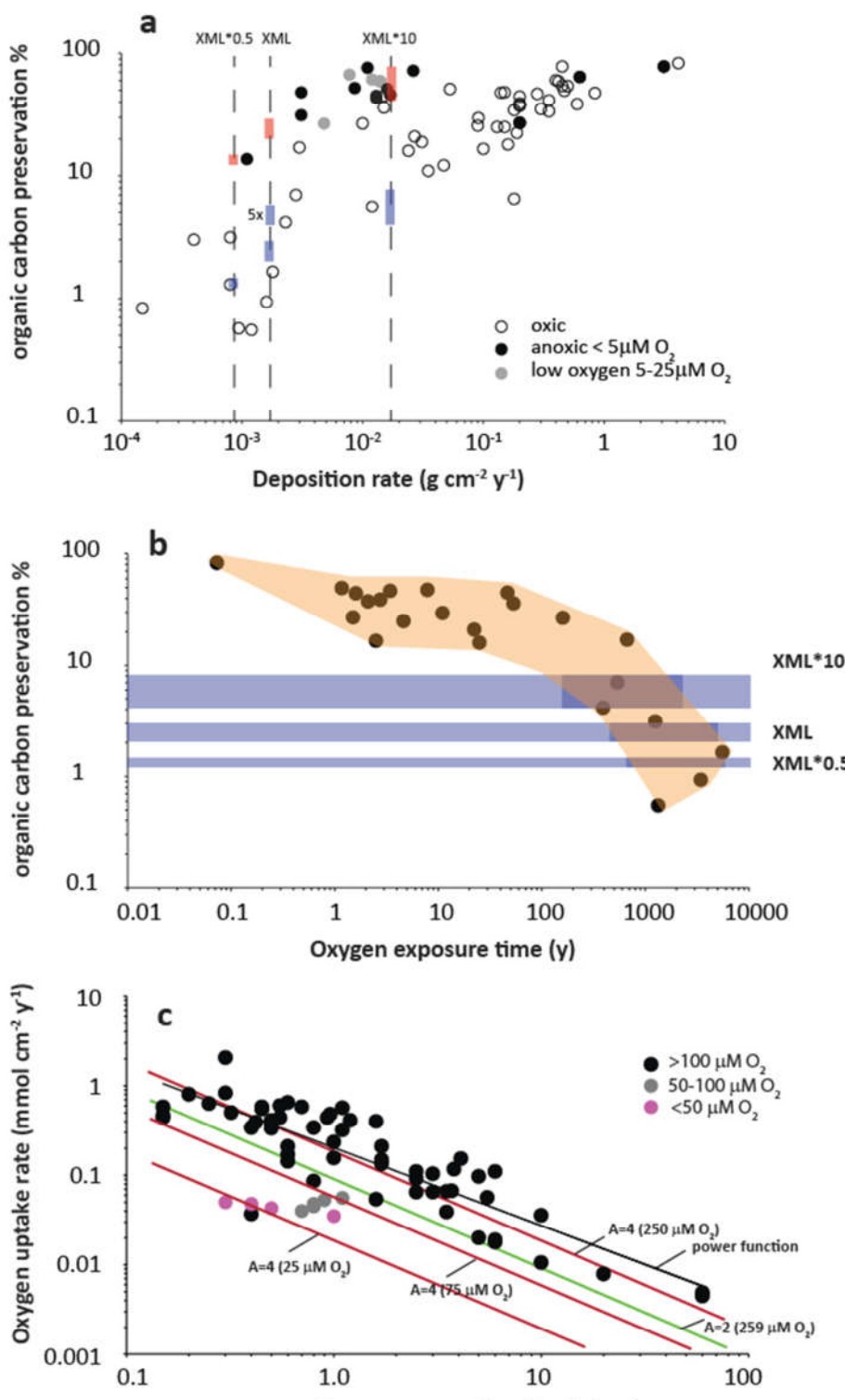

