# Peer review of "The oxic degradation of sedimentary organic matter 1400 Ma constrains atmospheric oxygen levels"

_Biogeosciences, 2016_

## Referee Comment (RC1) · Anonymous Referee #1 · 16 Nov 2016

Review of Zhang et al., Bioegosciences Discussion DOI: 10.5194/bg-2016-413

General Comments:

This paper provides analysis for support of a compelling idea that oxygen was sufficient in the mid-Proterozoic ocean for evolution of higher-order species long before their actual evolution, suggesting that biological evolution was not limited by oxygen levels, as has been the suggested and currently accepted paradigm. This idea has been the subject of much debate in recent publications with recent work on Cr isotopes published in Science and Nature suggesting the O2 levels in the Mesoproterozoic are even lower (<1% PAL) than is currently widely accepted ($\sim$1% PAL), and, therefore, this topic certainly warrants further study and evidence to determine oxygen levels in this

important Eon. The authors are leaders in their field, and have presented other lines of evidence to support the view of an oxygenated Mesoproterozoic world in related recent publications. However, in general, I feel that the overall presentation of this paper could be clearer, better structured and more fluent. Often both the sentence structure and the argument structure are confusing, weakening the overall presentation of very novel and interesting work. The discussions presented in this paper rely heavily on a diagenetic model to determine oxygen necessary for estimated carbon mineralization (inferred from measured [TOC] and HI) in the paleo-ocean. I applaud the efforts of the authors for their efforts to calibrate the model to many modern "analogues". However, the presentation of this model and the subsequent conclusions are often convoluted, and do not present strong enough arguments to back the authors claims, especially in regards to the recent controversial evidence from other paleo-reconstruction work of oxygen levels in this Eon. Therefore, it is hard to follow the applied methodology (diagenetic model) from the way the discussion is currently presented. As much of the conclusions of this work rely on heavy data interpretation, I would recommend a clearer dialogue throughout the manuscript, which would strengthen the conclusions this work has in our understanding of Mesoproterozoic oxygen levels.

Specific Comments:

Comments for throughout the manuscript-

It seems like there are many "missing" references for methodology used and some discussion points. This opens room for debating the usage of the approaches outlined in the paper. e.g. lines 105, 110, 135-138 for methodology lines 211, 220, 250, 293 to support specific claims made in the paper.

1 Introduction-

Authors must keep in mind the broad readership of Biogeosciences as a journal and define the background and significance of this work in a bit more detail. The intro feels short and incomplete, especially in regards to framing the current state of the research

and preparing the readers for the in-depth models that follow.

2 Methods-

2.1 Study Site: While the authors have obviously been working with the Xiamaling Formation and are vary familiar with its stratigraphy, the readers would benefit from a figure defining the "units" of the formation and it's overall place in the paleo-record (ie. dating).

2.2 Sample Collection and Analytical Methods:

Lines 103-105: for the geochemical data, do you have a reference for the preparation of these samples?

Line 109: Are the accuracies presented adequate? For example, it is often accepted in organic geochemical analyses that over 10% variability is "too high". Is this acceptable for trace metal work? Do you have a reference to support that your methodology is acceptable?

Line 110: Define what major elements you looked at with X-ray fluorescence and a reference to the method used.

Line 113: Redefine HI, as you only have defined it in the abstract

Line 115: How was TOC and S2 measured? Pyrolysis? This description is a little confusing

Line 125: How did you dry your samples?

Line 125-130: Is this TOC measured on the C-S analyzer and the EA used for the HI presented in the paragraph above? If so, you should present this first so the readers aren't wondering where the data came from. Is the uncertainty presented for the same standards an inter-lab comparison, or is it for the uncertainty between replicates in the same lab?

Lines 131- 139: Are these methods previously published? References?

Line 138: Has there been any work done on losses or transformations of kerogen OM in regards to the removal methods of the carbonates, silicates, and fluorides? For example, in modern terrestrial and aquatic samples, significant losses and alterations to OM content, composition and isotopic fractionation have been observed with use of concentrated acids and rinse steps (See e.g.: Gélinas et al., Organic Geochemistry, 2001 DOI: 10.1016/S0146-6380(01)00018-3 and discussion in Brodie et al., Chemical Geology, 2011 DOI: 10.106/j.chemgeo.2011.01.007)

Lines 161-172: What is the rational behind the 3 methods of iron determination? Why did you not do a split comparison on the unit 1?

Lines 173-182: This should not be in the Methods section, please move to results and/or discussion.

3 Results

Line 184: Please describe a bit of the results rather than just referring me to a supplemental table. For example, averages and standard deviations could be given for black and green-gray shales within the text.

Figure 1: How does the stratigraphic height on the Y axis relate within your or to your unit number and the lithology? Descriptions of variations in the constituents with stratigraphic heights are largely lacking. For example in lines 188-189 the U/Al ratios are not always near the crustal values (e.g. around 30 m). Why?

Lines 189-191: What is the significance of the higher Fe/Al values? No context for what a "higher value" even is- for example, is the difference observed between the ratios with shale type significant? Then what's the explanation of the differences with depth, which is often greater than differences between shale types?

Figure 2: This figure could benefit greatly from color separation of the green-gray and the black shales in ALL panels (such as in Figure 1). The figure description and much

of the subsequent text is not clearly supported by the actual figure (e.g. the conclusion in line 309 is impossible to tell from the Figure 2 as is). Color-coding would help alleviate some of this confusion (for example discussions on lines 299-305, lines 307-309). Don't refer to another figure in a figure caption! The figure caption should be self-sufficient to the figure. Also, please explain in the text the significance of the line denoting the TOC wt % of 1%.

4 Discussion

Line 210: The figures referenced in this sentence don't support the claim. The best figure to support this statement is probably Figure 2, but based on this figure, HI and TOC don't look that correlated among like-shale types, as the different shale types separate out completely. The "linear" relationship across the two shale types don't support your conclusions here.

Line 215-217: What are the potential sources of OM for these age sediments?

Lines 223-232: I don't understand the purpose of this whole discussion. How can you compare your sediments to these when the terrestrial/aquatic assemblages are totally different and the depositional regimes were likely vastly different? For example, you stated in line 215 that your sediments were deposited before land-plant evolution, so how can you compare the HI index (even with relation to O2 availability) you see in your samples to these other, newer sediments? Couldn't the HI index be similar/different for completely different reasons at this point? If this is not the case, please make a less confusing and sounder argument for why high TOC and HI samples are deposited under anoxic conditions and how this can be utilized for older sediments. This become especially important when you are discussing processes that can influence HI in more modern sediments (Lines 274-278), as these depositional processes could have been very different for the sediments you are comparing and using to build your argument. As the vary least, you need to discuss that the depositional regimes were likely similar for the sediments even though Eons apart. I feel that much of this discussion in

Section 4.1 could be condensed, strengthened and over all clarified. I feel like your conclusion in Lines 289-291, that there are fluctuations between anoxic and oxic deposition conditions, seems a bit of a leap from the arguments presented in the section above.

Lines 292-293- Do you have a figure or reference to support this claim?

Lines 295-300- How do you know this is Fe enrichment and not Al depletion? A depletion in Al would give you the same ratio as an enrichment as Fe, potentially. What is happening at the 15 and 35 m mark in your core? The ratios also seem to overlap within the same range between the two types of shales. How do you know the variation you see is significant and indicative of the changes in depositional environment?

Line 408: Is the use of this density value in this context something that is already published in the literature? Cite if yes, but if not, this needs to be further validated!

Line 426-430 (Equation 3): A is not defined in the equation description

Lines 519-527: I don't follow the logic of this argument. How can you compare your oxygen estimates with modern observations? Isn't it likely the mid-Proterozoic ocean was different (in terms of oxygen penetration depth especially) than your modern analogues, as we now live in a more oxygen rich world? So how does this comparison support your conclusions?

Line 550: I feel like it should be mentioned that there is no precise dating of unit 1 MUCH earlier in the text!

Technical Corrections:

This paper would benefit from a careful proofreading and calibration of citation software used. For example, there are many citation-related errors that probably have resulted from the citation software used: e.g. Lines 99, 147, etc. where parenthetical citations are inappropriate, lines 115-116 where the exact citations are presented twice in a row, lines 594-599 where the same exact reference is presented as two distinct references

(Cole et al., 2016a and 2016b), line 542 where two references in a row are presented separately "(Planavsky et al., 2014) (Cole et al., 2016a)" as opposed to "(Planavsky et al., 2014; Cole et al., 2016)".

Lines 41-43: Sentence starting with "The original idea..." is not a complete thought. Line 43, 44, 46, etc: Please define the element names before using their symbols at first use. There is a shift back and forth between the element name and symbol (e.g. in line 46 "chromium-associated" and "Fe-enriched"). Please at least be consistent with the usages.

Line 50, 56: Please provide more background on your "unit" numbers. The lack of context what unit 3 is vs unit 1 is confusing for the readers to follow. A figure might help with this!

Lines 68-72: Sentence starting with "These intrusive sills..." is wordy and confusing!

Line 73: "...like the sediments depositing just before" I would consider rewording this phrase as it sounds a bit weird.

Line 79-80: Add space between the degree symbol and N

Lines 84-88: Sentence starting with "Previous work....". This sentence is 5 lines long! Break into shorter sentences.

Line 103: This should read "...drilling depth, angle, and cross calibration..."

Line 115: "HI is defined as S2*100/TOC" (i.e. HI = S2*100/TOC) This is an equation, shouldn't this have an equation number?

Line 148: "...highly iron..." do you mean highly reactive iron?

Line 159: You already defined FeHR on line 148

Lines 218-220: This not a complete thought.

Line 220-223: This sentence is confusing !

Line 266: After organic carbon burial flux maybe place the abbreviation used in the equation? Cbur?

Line 354: This transition "Returning to the sediment model..." is awkward. I would consider re-wording it.

Line 543: "...chrome, and it's isotopes..." Do you mean chromium? "It's isotopes" should read "its isotopes", as it's used as a possessive pronoun here.

Line 546: "...chrome component" do you mean chromium?

Line 548: the usage of "square" is awkward. I would consider rewording this sentence.

---

## Referee Comment (RC2) · Anonymous Referee #2 · 22 Dec 2016

"General comments"

This paper aims to determine oxygen levels prevailing in oceans 1.4 Gyr ago. This topic is of interest and debated as evidenced by the numerous publications in high-rank journals. As stressed by the authors, the use of the chromium isotopes as a redox indicator can be discussed implying the need for complementary or new approaches as the one published by the same authors in PNAS (Zhang et al., 2016). However, this manuscript shares the same conclusion than Zhang et al., 2016. There are too many similarities between the two manuscripts to fully consider this one as a new manuscript. In my opinion, the manuscript must refocus on the approach rather than the already published conclusion.

[Figure]

The two studies are only distinguished by (i) the studied geological units and (ii) the determination of oxygen exposure time and penetration depth: (i) Their previous study dealt with units 2,3 and 4 while this one focus on the unit 4. Unfortunately and as stressed by the authors (L550), there is no precise dating of unit 1 in the Xiamaling Formation. Hence, the main original conclusion of the manuscript (persistent atmospheric oxygen over million years) is not convincing especially in the light of the alternating gray (oxidizing conditions) and black shales (anoxic conditions; see Zhang et al., 2016). (ii) Although very interesting, providing a fair review on the determination oxygen exposure time and penetration depth is too difficult in the current state of the manuscript. The overall presentation is too confusing as for the distinction between anoxic and oxic depositional environments

Finally, the quality of figures and tables does not stand for the publication standard.

In my opinion, this manuscript is not fully original (because of its redundant conclusion with the PNAS paper) but presents an innovating approach. I suggest the authors to modify this manuscript focusing on the new unique feature of this version (TOC-derived calculations) with a significant improvement of the overall presentation.

"Specific comments":

Introduction The introduction is very short. There is an overall lack of contextualization. Moreover, authors finalize their introduction by evidencing that they will present equivalent results than their former publication. It is quite destabilizing. L19-20: Where? It is not so clear in the manuscript. L43: Uranium (U) L44: Molybdenum (Mo) L49 "in contrast, sediments from unit 3": It is probably too specific for the global purpose of an introduction. Do you have other references than yours ?

Study site and methods L113-122: What were the samples analyzed by Rock-Eval pyrolysis ? kerogens or crushed rocks ? According to the logic of the manuscript, it may be crushed rocks. L113 HI: Hydrogen Index L114: Using the HI index required the determination of the TOC by Rock-Eval Pyrolysis. To determine the TOC, carbonaceous compounds are also determined during the combustion step (oxidation oven). It must be clarified. Moreover, why don't you use the Oxygen Index ? It cannot be as a consequence of the presence of carbonates since Rock-Eval device provide the possibility to distinguish oxygen from both carbonates and organic matter sources (Baudin et al., 2015; organic geochemistry). OI is often used as a proxy of oxidation of the organic matter during early diagenesis. L116-117: This assertion is not true. Following cited references, S2 corresponds to the amount of hydrocarbon released upon pyrolysis without any distinction about the molecular weight of the hydrocarbons. L131-139: Isolation procedure does not correspond to the classical procedure, why ? L131-139: have you investigated the effect of the isolation procedure on the preservation of organic matter ? HCl procedure can lead to artefactual degradation of aliphatic moieties implying in turn, a shift in the carbon isotope composition.

Results Figures: The quality of Figure 1 and 2 is not acceptable.

Discussion 4.1 Organic carbon preservation and water column chemistry This discussion section is too long and not really pertinent. The TOC and HI high values are attributed to the preservation of organic matter under anoxic conditions. In turn, I have the feeling that the TOC and HI low values are allocated to oxic environments by default. Factually, the FeHR/FeT is the best criterion to distinguish between oxic and anoxic depositional environments. As the distinction between the oxic and anoxic environments is crucial to sustain the determination of O2 level, it is essential to provide a more thorough argumentation. For instance, why their results about element traces are not compared with data from literature ? Moreover, interpretation of element traces is by far more complicated. In this case, there is again a lack of contextualization.

L210 "HI has often been linked": References are needed L211: higher=>high L211: more => better preservation of L212 poorer => a low L218-232: This paragraph can be shortened. In its present form, it is confusing and not really interesting. L248-261: the good preservation of organic matter in anoxic environment is not new and does not deserve such a large paragraph. L269-270: This sentence is in contradiction

with your previous statement (see comment on L210) L274; If HI values are "blurred" by sediment re-suspension or transport, why is it not the case for the TOC values? Indeed, HI and TOC characterize the same organic matter. L279-280: It is redundant with a previous sentence. L280: $\delta$13C was determined on the insoluble organic matter not on the whole bulk rock in contrast to Rock-Eval parameters. During the isolation of the kerogen, the use of HCl can degrade then aliphatic content leading in turn to a bias in the determination of the $\delta$13C. Have you any evidence that such a bias does not affect the $\delta$13C values ? Anyway, I don't really understand the relevance of the $\delta$13C.

For the 4.2 discussion section, an extensive rewriting effort is required to simplify the whole structure. Indeed, several readings of the draft text are required to fully understand the approach. Otherwise, I've no concern about the calculations.

---

## Short Comment (SC1) · 26 Dec 2016

We are very pleased that the reviewer sees the value of this work and appreciates the approach that we have used to constrain levels of Mesoproterozoic atmospheric oxygen. The reviewer also offers a number of excellent comments which will surely improve the manuscript during the revision stage. We thank the reviewer for these comments and look forward to implementing them during our revisions.

---

## Short Comment (SC2) · 26 Dec 2016

In what follows, we offer a brief response to the main criticisms of Reviewer 2. The main criticism of this reviewer is reproduced below:

"This paper aims to determine oxygen levels prevailing in oceans 1.4 Gyr ago. This topic is of interest and debated as evidenced by the numerous publications in highrank journals. As stressed by the authors, the use of the chromium isotopes as a redox indicator can be discussed implying the need for complementary or new approaches as the one published by the same authors in PNAS (Zhang et al., 2016). However, this manuscript shares the same conclusion than Zhang et al., 2016. There are too many similarities between the two manuscripts to fully consider this one as a new manuscript.

In my opinion, the manuscript must refocus on the approach rather than the already published conclusion."

With all due respect to the reviewer's concerns, we are puzzled by the idea that "there are too many similarities between the two manuscripts[this manuscript and an earlier one by our group published in PNAS] to fully consider this one is a new manuscript" because "...this manuscript shares the same conclusion than Zhang et al. 2016." We stress: 1) that there is no consensus on levels of oxygen during the Mesoproterozoic Era. Basically, there are a pair of chromium isotope studies suggesting very low levels of atmospheric oxygen, and our PNAS contribution suggesting much higher levels, 2) that the relationship, therefore, between the history of atmospheric oxygen and animal evolution is currently unresolved, 3) the current manuscript uses a completely different methodology to establish similar oxygen concentrations to the PNAS paper on a completely different section of the Xiamaling Formation, 4) the methodology explored in this manuscript is completely novel and is the first to evaluate the consequences of oxygen exposure on carbon preservation during the Precambrian.

Therefore, we view this manuscript as a novel contribution to an important scientific debate. The fact that our two contributions, utilizing completely different methods, offer similar conclusions as to levels of Mesoproterozoic atmospheric oxygen strengthens the idea that atmospheric oxygen levels were higher than predicted from chromium isotope studies and reinforce the idea that sufficient oxygen for animal respiration was available in the environment long before the evolution of animals themselves.

We might understand the reviewer's concerns if the scientific problem we were exploring was well resolved and trivial. However, this is not the case with the Mesoproterozoic history of atmospheric oxygen and its relationship to animal evolution.

We also offer a brief response to some of the other reviewer's concerns.

The reviewer state "The two studies are only distinguished by (i) the studied geological units and (ii) the determination of oxygen exposure time and penetration depth"

[Figure]

Yes, the studies do look at different parts of the Xiamaling Formation, but the methodology used to constrain atmospheric oxygen is completely and fundamentally different. In the PNAS paper we utilized a water column model to determine the minimum levels of atmospheric oxygen required to allow bottom water oxygenation. In the present manuscript we utilize a sediment diagenetic approach to explore the minimum levels of bottom water oxygen required to generate the amounts of carbon oxidation needed to reproduce the HI values we observe in the sediments.

The reviewer states: "Unfortunately and as stressed by the authors (L550), there is no precise dating of unit 1 in the Xiamaling Formation. Hence, the main original conclusion of the manuscript (persistent atmospheric oxygen over million years) is not convincing especially in the light of the alternating gray (oxidizing conditions) and black shales (anoxic conditions; see Zhang etal., 2016)."

Yes, there is no precise dating of unit 1. However, there is a well-dated interval of 8 million years separating unit 3, from which the calculations of oxygen concentration in the PNAS paper were derived, and 220 meter in unit 2. There is a further 180 meter of stratigraphy between the last dated interval in unit 2 and the interval in unit 1 where we make our oxygen calculations in the present paper. Therefore, even if this 180 m of stratigraphy deposited instantaneously (which is completely unreasonable) we observe elevated oxygen over a time interval separated by 8 million years. We agree that we have not determined persistent oxygenation over millions of years (although this is likely) but that oxygen is found in intervals spanning millions of years. We believe that this point is significant and will be better developed in the revised manuscript.

Outside of these comments, the reviewer's comments are very helpful, and together with those of reviewer 1, will help to significantly improve the manuscript.

———————————————————

---

## Short Comment (SC3) · 6 Jan 2017

D. Cole

devon.cole@yale.edu

This study by Zhang et al. seeks to improve our understanding of mid-Proterozoic oxygen levels by examining the 1.4 Ga Xiamaling Formation of North China. Atmospheric oxygen through this interval has been the subject of debate, and recent work on the Xiamaling Formation has also incited discussion (Planavsky et al., 2016; Zhang et al., 2016a; Zhang et al., 2016b). This study takes a somewhat different approach than that of Zhang et al. (2016a) to tracking global oxygen signals preserved in the Xiamaling, but comes to the same conclusions. It appears, however, that there are a number of issues in this manuscript that have resulted in potentially spurious conclusions. The introduction of the paper also does not present a very balanced view of work on Pro-

terozoic pO2 evolution.

The introduction of this paper does not cover all of the basic background of our current understanding of mid-Proterozoic oxygen levels, failing to acknowledge much of the significant work that has previously laid the groundwork on which recent studies are based. Traditional estimates for mid-Proterozoic (1.8 to 1.4 Ga) oxygen levels are typically considered to be those based on detrital pyrite and paleosol records (Holland, 2006). While there are not detrital pyrite occurrences in the mid-Proterozoic, the only well documented paleosols through this interval are characterized by Fe and Mn loss (Mitchell and Sheldon, 2009). This suggests, using the traditional framework for quantifying atmospheric oxygen levels from paleosols, that atmospheric oxygen levels were less than ∼ 1% PAL (e.g., Crowe et al., 2013; Rye and Holland, 1998; Zbinden et al., 1988). More recently, additional geochemical proxies such as chromium isotopes and Zn/Fe ratios have also yielded estimates for mid-Proterozoic atmospheric oxygen as low as < 1% PAL (Cole et al., 2016; Liu et al., 2016; Planavsky et al., 2014). The 40% PAL upper estimate for pO2 levels of Canfield (1998) rests on a number of assumptions—such as roughly constant productivity through Earth history, despite changing oxygen levels—that have been questioned by multiple researchers. Foremost, both Laakso and Schrag (2014) and Derry (2015) have articulated that the only means to achieve a low-oxygen ocean–atmosphere system is to greatly reduce productivity. More importantly, regardless if the 40% PAL estimate is considered valid, it is a "maximum estimate," which, by definition, does not exclude any lower estimates. This estimate is, therefore, perhaps not the most useful framework to introduce and consider low oxygen levels in the mid-Proterozoic.

While assumptions must be made in any model-based attempt at reconstructing paleoenvironmental parameters, the simplification of natural fluxes and phenomena must be carefully chosen, justified, and explored. Unfortunately, I fear the authors here have made assumptions that oversimplify the processes that shaped the paleoenvironment of the Xiamaling Formation and have failed to explore the results of these choices.

Namely, the authors assume both constant organic matter input and a constant sedimentation rate throughout the deposition of the unit (on the order of millions of years). Instead, dramatic variations in both of these fluxes would, over a range of time scales, be expected in a continental margin setting, and it is likely that these variations would be the primary drivers of changes in preserved organic matter (e.g., Liu et al., 2000, and references therein). It is reasonable to consider that alternating organic-rich and organic-poor layers in shallow marine settings can be driven by increased or pulsed sedimentation, pulsed productivity, variations in the extent of degradation, or any combination thereof. Instead, the authors here assume all variations are tied to changes in the extent of degradation. Further, the assumption of a constant organic flux is embedded into the remineralization estimates that are used to derive the pO2 estimates. More plainly, the authors make the likely dubious assumption of static conditions and this assumption directly affects their conclusions.

These expected variations are also likely important in the interpretation of the hydrogen index (HI), which has not been fully explored in this manuscript. Importantly, the HI reflects organic matter degradation, alteration, and composition broadly, not just the effects of aerobic marine alteration. Varying amounts of initial organic matter coupled with the same extent of degradation (from aerobic remineralization, iron reduction, etc.) will result in varying bulk sediment HI. Further, the organic-rich samples are likely to have been deposited under a sulfidic water column (as evidenced by Mo enrichments presented by the authors), in contrast to the less organic-rich sediments, which could have been deposited under oxic or anoxic but non-sulfidic conditions. As pyrite abundances are anomalously not reported (see below), Fe speciation cannot be used to determine if the setting was euxinic. The higher Mo enrichments in the more organic-rich sediments can, however, be roughly tied to sulfide availability. Greater sulfide availability can then subsequently change the redox state of organic matter oxidation and in turn, the HI. In sum, the HI data by no means require essentially constant organic matter fluxes with variable extents of early diagenetic remineralization.

The authors also assume no continentally derived detrital organic carbon input, despite a proximal setting. Non-negligible fluxes of detrital organic matter to marine systems are found under even modern atmospheric oxygen levels, and such fluxes will be higher under lower atmospheric oxygen levels (e.g., Derry, 2014). Continentally derived detrital organic material would be expected to have a lower HI than fresh marine organic matter. With lower TOC samples the effects from detrital organic matter will be more pronounced. Therefore, the observed correlation between HI and TOC enrichment could also be tied in part to mixing. Due to its potential influence, detrital organic matter should not be excluded from the model, especially since an oxidizing atmosphere should not be assumed in the starting conditions if that is the central conclusion, introducing a potential circularity to the model.

More broadly, numerous authors have suggested that HI does not correlate well with bottom-water oxygen availability in many cases, suggesting variations can be driven by a number of other factors including sedimentation rate, sediment dilution and mixing, changes in organic matter delivery, grain size, and hydrodynamic effects (e.g., Calvert et al., 1995; Cowie, 2005; Rao and Veerayya, 2000) and these factors are not discussed.

The inclusion of a detailed stratigraphic column would be a great (and likely necessary) addition to the paper. Further, there is no mention of any lithology besides shale in this unit, but it is not possible to form HCS without grains coarser in size than mud or clay (Cheel and Leckie, 1993; Dott and Bourgeois, 1982; Dumas et al., 2005). Deposition of silt or sand (as required by the presence of HCS) in a continental margin setting will, even assuming relatively constant (e.g., myr-scale) fluxes, be dynamic (i.e., pulsed) on the scale of individual beds, presenting a significant conflict with the assumptions of constancy made in this study.

Finally, on a more technical front, the authors have failed to report Fepy, which is not only a standard component of iron speciation, but could also contribute important information about sulfide availability in the section (which, as stated above, can have
important ramifications in the interpretation of the HI signal). Pyrite to oxide ratios are also a means of gauging whether there was recent groundwater alteration. Also, troublingly, samples were crushed in a steel mill, which is not standard practice for an iron speciation or trace metal study as steel can easily contaminate many trace metals of interest (Hickson and Juras, 1986; Takamasa and Nakai, 2009). At a minimum, some information on how blanks were monitored must be provided. Lastly, the error on the U concentration analyses of 17% makes the small variations and slight enrichments observed in this study nearly meaningless.

References

Calvert, S., Pedersen, T., Naidu, P. and Von Stackelberg, U. (1995) On the organic carbon maximum on the continental slope of the eastern Arabian Sea. Journal of Marine Research 53, 269-296.

Canfield, D.E. (1998) A new model for Proterozoic ocean chemistry. Nature 396, 450-453. Cheel, R. and Leckie, D. (1993) Hummocky cross-stratification. Wright, VP (ed.) Sedimentology review, 1: 103-122. Blackwell.

Cole, D.B., Reinhard, C.T., Wang, X., Gueguen, B., Halverson, G.P., Gibson, T., Hodgskiss, M.S.W., McKenzie, N.R., Lyons, T.W. and Planavsky, N.J. (2016) A shale-hosted Cr isotope record of low atmospheric oxygen during the Proterozoic. Geology.

Cowie, G. (2005) The biogeochemistry of Arabian Sea surficial sediments: A review of recent studies. Progress in Oceanography 65, 260-289.

Crowe, S.A., Dossing, L.N., Beukes, N.J., Bau, M., Kruger, S.J., Frei, R. and Canfield, D.E. (2013) Atmospheric oxygenation three billion years ago. Nature 501, 535-538.

Derry, L. (2014) Organic Carbon Cycling and the Lithosphere. Treatise on geochemistry 12.9.

Derry, L.A. (2015) Causes and consequences of mid-Proterozoic anoxia. Geophysical Research Letters 42, 8538-8546.

Dott, R.H. and Bourgeois, J. (1982) Hummocky stratification: Significance of its variable bedding sequences. Geological Society of America Bulletin 93, 663-680.

Dumas, S., Arnott, R.W.C. and Southard, J.B. (2005) Experiments on Oscillatory-Flow and Combined-Flow Bed Forms: Implications for Interpreting Parts of the Shallow-Marine Sedimentary Record. Journal of Sedimentary Research 75, 501-513.

Hickson, C.J. and Juras, S.J. (1986) Sample contamination by grinding. Canadian Mineralogist 24, 585-589.

Holland, H.D. (2006) The oxygenation of the atmosphere and oceans. Philosophical Transactions of the Royal Society of London B: Biological Sciences 361, 903-915.

Kump, L.R. (2008) The rise of atmospheric oxygen. Nature 451, 277-278.

Laakso, T.A. and Schrag, D.P. (2014) Regulation of atmospheric oxygen during the Proterozoic. Earth and Planetary Science Letters 388, 81-91.

Liu, K., Iseki, K. and Chao, S. (2000) Continental margin carbon fluxes. The Changing ocean carbon cycle: A midterm synthesis of the Joint Global Ocean Flux Study 5, 187.

Liu, X., Kah, L., Knoll, A., Cui, H., Kaufman, A., Shahar, A. and Hazen, R. (2016) Tracing Earth's O2 evolution using Zn/Fe ratios in marine carbonates. Geochemical Perspectives Letters 2, 24-34.

Mitchell, R.L. and Sheldon, N.D. (2009) Weathering and paleosol formation in the 1.1 Ga Keweenawan Rift. Precambrian Research 168, 271-283.

Planavsky, N.J., Cole, D.B., Reinhard, C.T., Diamond, C., Love, G.D., Luo, G., Zhang, S., Konhauser, K.O. and Lyons, T.W. (2016) No evidence for high atmospheric oxygen levels 1,400 million years ago. Proceedings of the National Academy of Sciences.

Planavsky, N.J., Reinhard, C.T., Wang, X., Thomson, D., McGoldrick, P., Rainbird, R.H., Johnson, T., Fischer, W.W. and Lyons, T.W. (2014) Low Mid-Proterozoic atmospheric oxygen levels and the delayed rise of animals. Science 346, 635-638.

[Figure]

Rao, B.R. and Veerayya, M. (2000) Influence of marginal highs on the accumulation of organic carbon along the continental slope off western India. Deep Sea Research Part II: Topical Studies in Oceanography 47, 303-327.

Rye, R. and Holland, H.D. (1998) Paleosols and the evolution of atmospheric oxygen: a critical review. American journal of science 298, 621-672.

Takamasa, A. and Nakai, S.I. (2009) Contamination introduced during rock sample powdering: Effects from different mill materials on trace element contamination. Geochemical Journal 43, 389-394.

Zbinden, E., Holland, H., Feakes, C. and Dobos, S. (1988) The Sturgeon Falls paleosol and the composition of the atmosphere 1.1 Ga BP. Precambrian Research 42, 141-163.

Zhang, S., Wang, X., Wang, H., Bjerrum, C.J., Hammarlund, E.U., Costa, M.M., Connelly, J.N., Zhang, B., Su, J. and Canfield, D.E. (2016a) Sufficient oxygen for animal respiration 1,400 million years ago. Proceedings of the National Academy of Sciences.

Zhang, S., Wang, X., Wang, H., Bjerrum, C.J., Hammarlund, E.U., Dahl, T.W. and Canfield, D.E. (2016b) Reply to Planavsky et al.: Strong evidence for high atmospheric oxygen levels 1,400 million years ago. Proceedings of the National Academy of Sciences.

---

## Short Comment (SC4) · 10 Jan 2017

We thank Devon Cole (hereafter DC) for her comments and welcome the opportunity to comment on them. The comments of DC will be listed in italics. (See .pdf for proper formating of this response)

This study by Zhang et al. seeks to improve our understanding of mid-Proterozoic oxygen levels by examining the 1.4 Ga Xiamaling Formation of North China. Atmospheric oxygen through this interval has been the subject of debate, and recent work on the Xiamaling Formation has also incited discussion (Planavsky et al., 2016; Zhang et al., 2016a; Zhang et al., 2016b). This study takes a somewhat different approach than that of Zhang et al. (2016a) to tracking global oxygen signals preserved in the Xiamaling,

but comes to the same conclusions. It appears, however, that there are a number of issues in this manuscript that have resulted in potentially spurious conclusions.

Please see our comments below.

The introduction of the paper also does not present a very balanced view of work on Proterozoic pO2 evolution.

We agree that the introduction was too short and could better provide a more nuanced discussion of the history of the problem and a better background for our approach. This issue will be addressed during manuscript revisions.

The introduction of this paper does not cover all of the basic background of our current understanding of mid-Proterozoic oxygen levels, failing to acknowledge much of the significant work that has previously laid the groundwork on which recent studies are based.

Agreed, see above.

Traditional estimates for mid-Proterozoic (1.8 to 1.4 Ga) oxygen levels are typically considered to be those based on detrital pyrite and paleosol records (Hol- land, 2006). While there are not detrital pyrite occurrences in the mid-Proterozoic, the only well documented paleosols through this interval are characterized by Fe and Mn loss (Mitchell and Sheldon, 2009). This suggests, using the traditional framework for quantifying atmospheric oxygen levels from paleosols, that atmospheric oxygen levels were less than 1% PAL (e.g., Crowe et al., 2013; Rye and Holland, 1998; Zbinden et al., 1988).

The paleosols referred to by DC (Mitchell and Sheldon, 2009) are around 1.1 Ga in age. They are very interesting paleosols in they are weathered fluvial deposits, where the soils themselves formed in river overbanks and in ponds when the sediments were periodically exposed. Thus, these paleosols have a complex history and hydrology. We are not aware of any attempts to model atmospheric oxygen levels from these paleosols, but we note that each paleosol must be considered individually with mod-

eling based on the chemical composition of the parent material, soil depth, pCO2, hydrology, among other things. Therefore, one cannot simply set an atmospheric oxygen (< 1% PAL as DC does) level without a detailed understanding of the paleosol chemistry (including mineral redox state) and a carefully-considered model. We note also that these paleosols are typically hematite-cemented, indicating oxidative weathering of the sediments when weathering occurred. Furthermore, the extent of iron loss (or enrichment) in these weathered sedimentary deposits depends very much on the element one chooses as immobile. Choosing aluminum as immobile (as has been typically done in paleosol oxygen reconstructions, see (Rye and Holland, 1998) for example), Fe becomes enriched in almost all of the paleosols. Both iron and aluminum are mobilized in choosing Zr, as Mitchell and Sheldon (2009) did, and the mobilization iron and aluminum is much less in choosing Ti as immobile. Indeed, in fluvial deposits, due to sorting of heavy grains and the initial weathering of rock materials before fluvial transport, it is difficult to decide which element to choose as an "immobile" baseline, but Zr and Ti, largely associated with heavy minerals, may be poor choices. We also note that the near contemporaneous Sturgeon Falls paleosol, as reported by Zbinden et al (1988) displays iron retention and extensive iron oxidation during weathering. This paleosol offers a minimum atmospheric oxygen concentration estimate, not a maximum one, as implied by DC. All of these paleosols also formed contemporaneously with independent evidence for elevated atmospheric oxygen including elevated chromium isotope compositions extracted from carbonates (Gilleaudeau et al., 2016). We suspect that DC will have some issues with this later contribution, but, all of these points underscore the importance of applying additional methodologies, as we do in our contribution, to understanding the evolution of Proterozoic atmospheric oxygen.

More recently, additional geochemical proxies such as chromium iso- topes and Zn/Fe ratios have also yielded estimates for mid-Proterozoic atmospheric oxygen as low as < 1% PAL (Cole et al., 2016; Liu et al., 2016; Planavsky et al., 2014).

There is much to discuss here. First, the Planavsky et al (2014) paper promotes a maximum mid-Proterozoic atmospheric oxygen level < 0.1% PAL ("These data suggest that atmospheric O2 levels were at most 0.1% of present atmospheric levels."). As Cole et al. (2016) did not perform any additional modeling or offer additional quantitative constraints, we assume that DC still supports < 0.1% PAL as a maximum mid-Proterozoic oxygen level. If not, we don't understand how and why their maximum estimate has changed and whether it is based on any type of solid modeling or just a hunch? But, let's consider each of these three contributions in turn.

Starting with Planavsky et al. (2014), we have already expressed our concerns that the chromium from mid-Proterozoic samples measured in the Planavsky et al. (2014) had a large, if not dominant, detrital component, and are therefore inappropriate for understanding the isotopic composition of chrome in contemporaneous seawater (Zhang et al., 2016).

We also have serious misgivings about the Liu et al (2016) paper. These misgivings are fully explored in a contribution in review by one of us (Canfield, in review). However, the short of the story is that the Liu et al (2016) paper requires that calcium carbonates capture dissolved iron in seawater at concentrations in equilibrium with atmospheric oxygen. In the (Canfield, in review) contribution it is shown that the equilibrium iron concentrations are many orders of magnitude lower than those predicted from the measured iron contents of the carbonates used in the Liu et al (2016) study. Indeed, in our view, this contribution is fundamentally flawed and we will not cite it.

As for Cole et al (2016), we also have concerns about a dominant detrital chromium isotope contribution in their chromium isotope record. In this paper, the authors used a "selective" extraction technique to dissolve authigenic chrome from the shale samples they analyzed. As in the original contribution by Planavsky et al. (2014), the chromium isotopes showed little variation from crustal values, leading the authors to argue that there was little oxidative weathering of the chromium supplied to the oceans and therefore low concentrations of atmospheric oxygen. In the figure below, taken from (Wang et al., in review), we have re-plotted the data from the Cole et al (2016)

contribution together with data from Gueguen et al (2016), which includes a summary of chromium isotope contributions from modern sediments. In the upper panel, we see that both the isotopic compositions and concentrations of chrome in Mesoproterozoic shale samples fall exactly in line with those from modern oxic sediments where no authigenic chrome enrichment is believed to occur. These results do not give confidence that Cole et al (2016) have captured a non-fractionated authigenic component in their Mesoproterozoic shale samples; rather, a non-fractionated detrital contribution seems likely. Furthermore, chrome is also leached during the "selective" extraction procedure from modern oxic sediments Gueguen et al (2016) (therefore, the extraction is not as selective as one would hope), and the isotopic composition of this chrome is similar to the whole rock. This would be expected if a small amount of the detrital component was leached during the extraction. The same is true for the Mesoproterozoic samples, with the leach and the whole rock samples showing almost the same isotopic composition. This interpretation does not prove that Cole et al (2016) did not leach a small amount of authigenic chrome from the shale samples they analyzed (although, by analogy with the modern oxic sediments, it's hard to imagine that no detrital chrome was included in this leach). However, we do not believe that Cole et al (2016) have sufficiently demonstrated that they have extracted an authigenic chrome component, and we believe that their data is completely consistent with a detrital chrome source in the Mesoproterozoic shales.

The 40% PAL upper estimate for pO2 levels of Canfield (1998) rests on a number of Assumptions such as roughly constant productivity through Earth history, despite changing oxygen levelsthat have been questioned by multiple researchers. Fore- most, both Laakso and Schrag (2014) and Derry (2015) have articulated that the only means to achieve a low-oxygen ocean–atmosphere system is to greatly reduce pro- ductivity. More importantly, regardless if the 40% PAL estimate is considered valid, it is a "maximum estimate," which, by definition, does not exclude any lower estimates. This estimate is, therefore, perhaps not the most useful framework to introduce and consider low oxygen levels in the mid-Proterozoic.

We're not quite sure of the problems that DC has with our referencing the Canfield (1998) paper. This was the first quantitative constraint on Mesoproterozoic atmospheric oxygen concentrations as well as a new insight into Mesoproterozoic ocean chemistry. It is also been the inspiration for many, if not most, subsequent studies of the evolution of Proterozoic atmospheric and oxygen chemistry. We in no way mean to consider 40% as a "valid" estimate for Proterozoic atmospheric oxygen levels; we have always promoted this as a maximum model-based estimate. Again, we don't quite see where DC is coming from here.

While assumptions must be made in any model-based attempt at reconstructing pale-oenvironmental parameters, the simplification of natural fluxes and phenomena must be carefully chosen, justified, and explored. Unfortunately, I fear the authors here have made assumptions that oversimplify the processes that shaped the paleoenvironment of the Xiamaling Formation and have failed to explore the results of these choices. Namely, the authors assume both constant organic matter input and a constant sedi-mentation rate throughout the deposition of the unit (on the order of millions of years).

This is patently incorrect. We have explored sedimentation rates ranging over a factor of 20, with organic matter input rates constrained by the concentrations of organic matter in the sediments and our constraints on carbon oxidation from our analysis of Xiamaling sediments and modern analogs. Organic matter input rates also vary by a factor of 20.

Instead, dramatic variations in both of these fluxes would, over a range of time scales, be expected in a continental margin setting, and it is likely that these variations would be the primary drivers of changes in preserved organic matter (e.g., Liu et al., 2000, and references therein). It is reasonable to consider that alternating organic-rich and organic-poor layers in shallow marine settings can be driven by increased or pulsed sedimentation, pulsed productivity, variations in the extent of degradation, or any com-bination thereof. Instead, the authors here assume all variations are tied to changes in the extent of degradation. Further, the assumption of a constant organic flux is embedded into the remineralization estimates that are used to derive the pO2 estimates.

We agree that fluxes in organic matter input can vary over time, which is one of the reasons that we explored sedimentation rates spanning a factor of 20. However, we disagree that changes in organic matter deposition should have consequences for the state of preservation of the organic matter at the time of deposition. We note also that in hundreds of sediment trap experiments throughout the global ocean, the organic matter concentration in the sediment in particles is never less than 2 wt% for sediments settling through the upper couple hundred meters of the water column, and much more commonly in the 4 to >10 weight % range (data from http://usjgofs.whoi.edu/mzweb/data/Honjo/sed_traps.html). These concentrations represent the concentrations of organic matter settling to the sediment surface. Therefore, organic matter concentrations in the low range we observe in the Xiamaling Formation unit 1 (0.1 to 0.3 wt%) are very unlikely without substantial sediment TOC decomposition. The fact that these low organic matter concentrations are associated with: 1) low HI index, 2) low trace metal content, 3) and a lack of Fe enrichment gives us good confidence that these low concentrations of organic matter are associated with oxygenated bottom waters and aerobic oxidation of organic matter in the sediment; completely consistent with modern observation on the controls of organic carbon preservation. Conversely, the high concentrations of TOC that we observe are associated with, 1) high HI, 2) high trace metal content, and iron speciation consistent with anoxic depositional conditions. We cannot imagine a sedimentalogically controlled scenario that would accommodate these observations, unless we just started to make things up.

More plainly, the authors make the likely dubious assumption of static conditions and this assumption directly affects their conclusions.

As stated above, we have modeled a large range of possible carbon preservation amounts and carbon depositional fluxes based on the complete range viewed in modern sediments. Our model, therefore, has explored an enormous range of possibilities, and our final oxygen estimates are based on the lowest, most conservative estimates

that we obtained.

These expected variations are also likely important in the interpretation of the hydrogen index (HI), which has not been fully explored in this manuscript. Importantly, the HI reflects organic matter degradation, alteration, and composition broadly, not just the effects of aerobic marine alteration.

We agree that many factors can control hydrogen index. These sediments deposited before the evolution of land plants, so the influence of lignin and cellulose can be ignored. We expect that all algal (although there is no evidence for eukaryotic algae in these deposits) and prokaryotic sources will have characteristics, as expressed through HI, of type 1 to 2 organic matter as expressed in van Krevelen diagrams. There is no difference in the isotopic composition of organic matter between the high and low TOC sediments, suggesting (but we admit not proving) a similar organic carbon source. Furthermore, during Rock-Eval analysis, there is no systematic differences in the Tmax values between the green-gray shales and the black shales (Zhang et al., 2015), demonstrating similar thermal maturity for organic matter in both of the sediment types. The Tmax values are also low, in the range of 430-440 oC demonstrating the low thermal maturity for all unit 1 organic matter.

Varying amounts of initial organic matter coupled with the same extent of degradation (from aerobic remineralization, iron reduction, etc.) will result in varying bulk sediment HI.

The point, based on comparisons of modern observations, is that there are large differences in the organic matter preservation depending on whether organic matter is decomposed in the presence or absence of bottom water oxygen. We refer DC to (Canfield, 1994), (Hartnett et al., 1998) and (Blair and Aller, 2012) for discussions.

Further, the organic-rich samples are likely to have been deposited under a sulfidic water column (as evidenced by Mo enrichments presented by the authors), in contrast to the less organic-rich sediments, which could have been deposited under oxic or

anoxic but non-sulfidic conditions.

The iron speciation, trace metal abundances, and HI are consistent with deposition under oxic conditions for the bulk of the low-TOC shales.

As pyrite abun- dances are anomalously not reported (see below), Fe speciation cannot be used to de- termine if the setting was euxinic.

All of our iron speciation data is reported in the supplement, and cross plots of FeHR/FeT vs FePy/FeHR show a combination of euxinic and ferruginous conditions when FeHR/FeT > 0.38. We did not feel that it was important to discuss water column chemistry in any detail in the present manuscript as the important point is the presence or absence of oxygen, and in particular, the role of oxygen in enhancing organic matter decomposition. In extensive work throughout the Xiamaling Formation we see similar enrichments in trace metals under both euxinic and ferruginous conditions (Wang et al., in review).

The higher Mo enrichments in the more organic-rich sediments can, however, be roughly tied to sulfide availability. Greater sulfide avail- ability can then subsequently change the redox state of organic matter oxidation and in turn, the HI. In sum, the HI data by no means require essentially constant organic matter fluxes with variable extents of early diagenetic remineralization.

Sorry, but we really don't follow the argument here. Please see our above responses for more discussion on the role of oxygen in organic matter preservation.

The authors also assume no continentally derived detrital organic carbon input, despite a proximal setting. Non-negligible fluxes of detrital organic matter to marine systems are found under even modern atmospheric oxygen levels, and such fluxes will be higher under lower atmospheric oxygen levels (e.g., Derry, 2014). Continentally derived de- trital organic material would be expected to have a lower HI than fresh marine organic matter. With lower TOC samples the effects from detrital organic matter will be more

pronounced. Therefore, the observed correlation between HI and TOC enrichment could also be tied in part to mixing.

It is true, we do not assume any continentally organic matter contributing to the Xiamaling Formation in unit 1. This is a good point and will be elaborated on further in our revised manuscript. It is, however, highly unlikely that the continents supplied organic matter to this unit of the Xiamaling Formation. As noted above, both high and low TOC units of unit 1 have preserved relatively immature TOC, with similar Tmax values. One would expect continentally derived recycled organic matter to display a higher degree of maturity than observed in these relatively immature rocks, and if one argued for a disproportionate amount of this organic matter in the low-TOC sediments, then this should be easily seen in the Tmax values, and probably also the carbon isotopic compositions. But, as noted above, there are no systematic differences. The degree of continentally-derived organic matter recycling will depend very much on the amount of oxygen one assumes there was an atmosphere. If the values were < 0.1%, then the organic matter recycling would likely be extremely important. This becomes less important as atmospheric oxygen concentrations increase.

Due to its potential influence, detrital organic mat- ter should not be excluded from the model, especially since an oxidizing atmosphere should not be assumed in the starting conditions if that is the central conclusion, intro- ducing a potential circularity to the model.

See above. There is no support for recycled continentally-derived organic matter in unit 1 of the Xiamaling. Assuming otherwise would just be making things up, and counter to the available evidence.

More broadly, numerous authors have suggested that HI does not correlate well with bottom-water oxygen availability in many cases, suggesting variations can be driven by a number of other factors including sedimentation rate, sediment dilution and mixing, changes in organic matter delivery, grain size, and hydrodynamic effects (e.g.,

Calvert et al., 1995; Cowie, 2005; Rao and Veerayya, 2000) and these factors are not discussed.

We do not feel it is our job to discuss all of the situations and hydrological circumstances where HI does not correlate well with bottom water oxygen. We do, however, in the MS offer a discussion of this issue, and note that in sediments off the Peruvian margin the relationship between HI and oxygen is complicated by sediment transport. In contrast, there are many situations, particularly in the geologic record, where the relationship between HI and oxygen availability is excellent. As noted above, HI correlates with other geochemical indicators of the presence or absence of bottom water oxygen, and thus we feel that the evidence is overwhelming that HI was most strongly influenced by the presence or absence of oxygen.

The inclusion of a detailed stratigraphic column would be a great (and likely necessary) addition to the paper.

Agreed

Further, there is no mention of any lithology besides shale in this unit, but it is not possible to form HCS without grains coarser in size than mud or clay (Cheel and Leckie, 1993; Dott and Bourgeois, 1982; Dumas et al., 2005). Deposition of silt or sand (as required by the presence of HCS) in a continental margin setting will, even assuming relatively constant (e.g., myr-scale) fluxes, be dynamic (i.e., pulsed) on the scale of individual beds, presenting a significant conflict with the assumptions of constancy made in this study.

Indeed, the sediments from unit 1 are mostly muds and silts, and we will provide a more detailed description of the sedimentology in the revised manuscript. There are some sand intervals as well, but these are mostly concentrated near the top of the unit. We don't get the point, however about constancy in the model. Our model is based on modern observations which experience the same stochastic dynamics in sedimentation that likely occurred in the Xiamaling. We note again, that in generating
our model results, we incorporate the whole range of observed carbon preservation rates in the modern environment at a given sedimentation rate and the whole range of oxygen exposure times at our different degrees of carbon preservation. This range of considerations generates an enormous breadth of model results and we always picked the most conservative lowest oxygen level as our reference. We believe that DC has misunderstood our modeling and its intentions.

Finally, on a more technical front, the authors have failed to report Fepy, which is not only a standard component of iron speciation, but could also contribute important information about sulfide availability in the section (which, as stated above, can have important ramifications in the interpretation of the HI signal). Pyrite to oxide ratios are also a means of gauging whether there was recent groundwater alteration.

These values were all reported in the supplemental information. Indeed, our core material was fresh, and the oxide contents are small indicating a lack of weathering or groundwater alteration.

. Also, trou- blingly, samples were crushed in a steel mill, which is not standard practice for an iron speciation or trace metal study as steel can easily contaminate many trace metals of interest (Hickson and Juras, 1986; Takamasa and Nakai, 2009). At a minimum, some information on how blanks were monitored must be provided.

Good point. We have also been sloppy in how we have described our trace metal methodology. All of the samples from the outcrop were crushed with a steel mill, (figure 1) while the vanadium samples from core material presented in Figure 2 were crushed in agate ball mill. Our blanks for the steel mill are very low and will be reported in the revised manuscript together with a more accurate description of our methodology.

Lastly, the error on the U concentration analyses of 17% makes the small variations and slight enrichments observed in this study nearly meaningless.

Our bad. Through rather careless cutting and pasting of methods, we have botched

our methods descriptions. All of the samples from outcrop (Figure 1) were measured with ICP-MS with precisions of better than 1.5%. Some of our V samples from the core (Figure 2) were also measured this way with the same uncertainties, but some are also measured with a hand-held XRF standardized against three different international standard materials with a precision of < 5%. An accurate accounting of methods and standard deviations will be reported in the revised MS, as well as an accounting in the SI of with method was used for the V outcrop samples (we did not report U as indicated in our description).

References Calvert, S., Pedersen, T., Naidu, P. and Von Stackelberg, U. (1995) On the organic carbon maximum on the continental slope of the eastern Arabian Sea. Journal of Marine Research 53, 269-296. Canfield, D.E. (1998) A new model for Proterozoic ocean chemistry. Nature 396, 450- 453. Cheel, R. and Leckie, D. (1993) Hummocky cross-stratification. Wright, VP (ed.) Sedimentology review, 1: 103-122. Blackwell. Cole, D.B., Reinhard, C.T., Wang, X., Gueguen, B., Halverson, G.P., Gibson, T., Hodgskiss, M.S.W., McKenzie, N.R., Lyons, T.W. and Planavsky, N.J. (2016) A shale- hosted Cr isotope record of low atmospheric oxygen during the Proterozoic. Geology. Cowie, G. (2005) The biogeochemistry of Arabian Sea surficial sediments: A review of recent studies. Progress in Oceanography 65, 260-289. Crowe, S.A., Dossing, L.N., Beukes, N.J., Bau, M., Kruger, S.J., Frei, R. and Canfield, D.E. (2013) Atmospheric oxygenation three billion years ago. Nature 501, 535-538. Derry, L. (2014) Organic Carbon Cycling and the Lithosphere. Treatise on geochem- istry 12.9. Derry, L.A. (2015) Causes and consequences of mid-Proterozoic anoxia. Geophysical Research Letters 42, 8538-8546. C5 Dott, R.H. and Bourgeois, J. (1982) Hummocky stratification: Significance of its variable bedding sequences. Geological Society of America Bulletin 93, 663-680. Dumas, S., Arnott, R.W.C. and Southard, J.B. (2005) Experiments on Oscillatory-Flow and Combined-Flow Bed Forms: Implications for Interpreting Parts of the Shallow- Marine Sedimentary Record. Journal of Sedimentary Research 75, 501-513. Hickson, C.J. and Juras, S.J. (1986) Sample contamination by grinding. Canadian Mineralogist 24, 585-589. Holland, H.D. (2006) The oxygenation of the atmosphere and oceans. Philosophical Transactions of the Royal Society of London B: Biological Sciences 361, 903-915. Kump, L.R. (2008) The rise of atmospheric oxygen. Nature 451, 277-278. Laakso, T.A. and Schrag, D.P. (2014) Regulation of atmospheric oxygen during the Proterozoic. Earth and Planetary Science Letters 388, 81-91. Liu, K., Iseki, K. and Chao, S. (2000) Continental margin carbon fluxes. The Changing ocean carbon cycle: A midterm synthesis of the Joint Global Ocean Flux Study 5, 187. Liu, X., Kah, L., Knoll, A., Cui, H., Kaufman, A., Shahar, A. and Hazen, R. (2016) Tracing Earth's O2 evolution using Zn/Fe ratios in marine carbonates. Geochemical Perspectives Letters 2, 24-34. Mitchell, R.L. and Sheldon, N.D. (2009) Weathering and paleosol formation in the 1.1 Ga Keweenawan Rift. Precambrian Research 168, 271-283. Planavsky, N.J., Cole, D.B., Reinhard, C.T., Diamond, C., Love, G.D., Luo, G., Zhang, S., Konhauser, K.O. and Lyons, T.W. (2016) No evidence for high atmospheric oxygen levels 1,400 million years ago. Proceedings of the National Academy of Sciences. Planavsky, N.J., Reinhard, C.T., Wang, X., Thomson, D., McGoldrick, P., Rainbird, R.H., Johnson, T., Fischer, W.W. and Lyons, T.W. (2014) Low Mid-Proterozoic atmospheric oxygen levels and the delayed rise of animals. Science 346, 635-638. C6 Rao, B.R. and Veerayya, M. (2000) Influence of marginal highs on the accumulation of organic carbon along the continental slope off western India. Deep Sea Research Part II: Topical Studies in Oceanography 47, 303-327. Rye, R. and Holland, H.D. (1998) Paleosols and the evolution of atmospheric oxygen: a critical review. American journal of science 298, 621-672. Takamasa, A. and Nakai, S.I. (2009) Contamination introduced during rock sample powdering: Effects from different mill materials on trace element contamination. Geo- chemical Journal 43, 389-394. Zbinden, E., Holland, H., Feakes, C. and Dobos, S. (1988) The Sturgeon Falls paleosol and the composition of the atmosphere 1.1 Ga BP. Precambrian Research 42, 141-163. Zhang, S., Wang, X., Wang, H., Bjerrum, C.J., Hammarlund, E.U., Costa, M.M., Con- nelly, J.N., Zhang, B., Su, J. and Canfield, D.E. (2016a) Sufficient oxygen for animal respiration 1,400 million years ago. Proceedings of the National Academy of Sciences. Zhang, S., Wang, X., Wang, H., Bjerrum, C.J., Hammarlund, E.U., Dahl,

T.W. and Can- field, D.E. (2016b) Reply to Planavsky et al.: Strong evidence for high atmospheric oxygen levels 1,400 million years ago. Proceedings of the National

Additional references

Blair, N. E. and Aller, R. C.: The fate of terrestrial organic carbon in the marine environment, Annual Review of Marine Science, 4, 401-423, 2012. Canfield, D. E.: Factors influencing organic carbon preservation in marine sediments, Chemical Geology, 114, 315-329, 1994. Canfield, D. E.: Trace metals in carbonates and atmospheric oxygen, a cautionary tale, in review. in review. Gilleaudeau, G. J., Frei, R., Kaufman, A. J., Kah, L. C., Azmy, K., Bartley, J. K., Chernyavskiy, P., and Knoll, A. H.: Oxygenation of the mid-Proterozoic atmospheric: clues from chromium isotopes and carbonates, Geochemical Perspectives Letters, 2, 178-187, 2016. Hartnett, H. E., Keil, R. G., Hedges, J. I., and Devol, A. H.: Influence of oxygen exposure time on organic carbon preservation in continental margin sediments, Nature, 391, 572-574, 1998. Rye, R. and Holland, H. D.: Paleosols and the evolution of atmospheric oxygen: A critical review, Am J Sci, 298, 621-672, 1998. Wang, X., Zhang, S., Wamng, H., Bjerrum, C. J., Hammarlund, E. U., Haxen, E. R., Su, S., Wang, Y., and Canfeld, D. E.: Oxygen, climate and the chemical evolution of a 1400 million year old tropical marine setting, in review. in review. Zhang, S., Wang, X., Hammarlund, E. U., Wang, H., Costa, M. M., Bjerrum, C. J., Connelly, J. N., Zhang, B., Bian, L., and Canfie'ld, D. E.: Orbital forcing of climate 1.4 billion years ago, PNAS, 112, E1406-E1413, 2015. Zhang, S. C., Wang, X. M., Wang, H. J., Bjerrum, C. J., Hammarlund, E. U., Costa, M. M., Connelly, J. N., Zhang, B. M., Su, J., and Canfield, D. E.: Sufficient oxygen for animal respiration 1,400 million years ago, Proceedings of the National Academy of Sciences of the United States of America, 113, 1731-1736, 2016.

Please also note the supplement to this comment:
http://www.biogeosciences-discuss.net/bg-2016-413/bg-2016-413-SC4-supplement.pdf

[Figure]

[Figure]

**Fig. 1.** Cr isotopes and concentrations from anciemt shales and modern sediments

---

## Author Comment (AC1) · 24 Jan 2017

Response to reviewer's comments

We thank the reviewers for their detailed comments and criticisms of our manuscript. Please find our detailed response to these comments below.

Anonymous Referee #1 Review of Zhang et al., Bioegosciences Discussion DOI: 10.5194/bg-2016-413

General Comments:

This paper provides analysis for support of a compelling idea that oxygen was sufficient
in the mid-Proterozoic ocean for evolution of higher-order species long before their actual evolution, suggesting that biological evolution was not limited by oxygen levels, as has been the suggested and currently accepted paradigm. This idea has been the subject of much debate in recent publications with recent work on Cr isotopes published in Science and Nature suggesting the O2 levels in the Mesoproterozoic are even lower (<1% PAL) than is currently widely accepted (1% PAL), and, therefore, this topic certainly warrants further study and evidence to determine oxygen levels in this important Eon. The authors are leaders in their field, and have presented other lines of evidence to support the view of an oxygenated Mesoproterozoic world in related recent publications. However, in general, I feel that the overall presentation of this paper could be clearer, better structured and more fluent. Often both the sentence structure and the argument structure are confusing, weakening the overall presentation of very novel and interesting work. The discussions presented in this paper rely heavily on a diagenetic model to determine oxygen necessary for estimated carbon mineralization (inferred from measured [TOC] and HI) in the paleo-ocean. I applaud the efforts of the authors for their efforts to calibrate the model to many modern "analogues". However, the presentation of this model and the subsequent conclusions are often convoluted, and do not present strong enough arguments to back the authors claims, especially in regards to the recent controversial evidence from other paleo-reconstruction work of oxygen levels in this Eon. Therefore, it is hard to follow the applied methodology (diagenetic model) from the way the discussion is currently presented. As much of the conclusions of this work rely on heavy data interpretation, I would recommend a clearer dialogue throughout the manuscript, which would strengthen the conclusions this work has in our understanding of Mesoproterozoic oxygen levels.

Response: We appreciate these comments and the general support for our approach. We will strive to provide better focus to our discussion as illuminated in our response to the points below.

Specific Comments: Comments for throughout the manuscript- It seems like there are

many "missing" references for methodology used and some discussion points. This opens room for debating the usage of the approaches outlined in the paper. e.g. lines 105, 110, 135-138 for methodology lines 211, 220, 250, 293 to support specific claims made in the paper.

Response: Thanks for this comment, we will supply the "missing" references.

1 Introduction- Authors must keep in mind the broad readership of Biogeosciences as a journal and define the background and significance of this work in a bit more detail. The intro feels short and incomplete, especially in regards to framing the current state of the research and preparing the readers for the in-depth models that follow.

Response: We agree. This was also a comment of Reviewer 2. We will work to expand the introduction and to place our work more solidly in the context of what is known not only of the history of atmospheric oxygen, but also the methods we have used to reconstruct the history and the present manuscript.

2 Methods- 2.1 Study Site: While the authors have obviously been working with the Xiamaling Formation and are vary familiar with its stratigraphy, the readers would benefit from a figure defining the "units" of the formation and it's overall place in the paleorecord (ie. dating).

Response: Good point. We will supply a stratigraphic overview of the Xiamaling Formation in the revised manuscript.

2.2 Sample Collection and Analytical Methods: Lines 103-105: for the geochemical data, do you have a reference for the preparation of these samples?

Response: These are pretty standard techniques, but we will supply reference on revised MS.

Line 109: Are the accuracies presented adequate? For example, it is often accepted in organic geochemical analyses that over 10% variability is "too high". Is this acceptable for trace metal work? Do you have a reference to support that your methodology is

acceptable?

Response: Prompted by this comment, we have gone back to reevaluate our methods and methods description. We realize that we should have been more careful in our presentation of the methods and their uncertainties. By far the most of our trace metal data was obtained by ICP-MS with uncertainties in the 1% range, not the ranges reported in the manuscript. We also generated some of our results with a hand-held XRF calibrated against numerous international standards and several splits of Xiamaling sediment independently calibrated with ICP-MS. Our uncertainties with this method were all better than 5%. We will carefully redraft our methods description to reflect what we have done, and in the supplementary information we will note which method would use to generate the different trace metal analyses.

Line 110: Define what major elements you looked at with X-ray fluorescence and a reference to the method used.

Response: Good point. Will do.

Line 113: Redefine HI, as you only have defined it in the abstract.

Response: Good point. Will do.

Line 115: How was TOC and S2 measured? Pyrolysis? This description is a little Confusing.

Response: TOC was measured with an elemental analyzer, while S2 was measured with standard Rock-Eval pyrolysis. We will clarify this description.

Line 125: How did you dry your samples?

Response: Samples were dried in a muffle furnace with the temperature <40°C

Line 125-130: Is this TOC measured on the C-S analyzer and the EA used for the HI presented in the paragraph above? If so, you should present this first so the readers aren't wondering where the data came from. Is the uncertainty presented for the same

standards an inter-lab comparison, or is it for the uncertainty between replicates in the same lab?

Response: Good point, we will reorganize our methods description as suggested. The uncertainties in TOC were presented for replicates run in the same lab.

Lines 131- 139: Are these methods previously published? References?

Response: Yes, (Durand and Nicaise, 1980).

Line 138: Has there been any work done on losses or transformations of kerogen OM in regards to the removal methods of the carbonates, silicates, and fluorides? For example, in modern terrestrial and aquatic samples, significant losses and alterations to OM content, composition and isotopic fractionation have been observed with use of concentrated acids and rinse steps (See e.g.: Gélinas et al., Organic Geochemistry, 2001 DOI: 10.1016/S0146-6380(01)00018-3 and discussion in Brodie et al., Chemical Geology, 2011 DOI: 10.106/j.chemgeo.2011.01.007).

Response: We have followed standard protocols in all of our analytical techniques. It is true that these techniques could have removed or transformed some organic matter during their application. However, previous studies have shown that organic matter hydrolysis during the recovery of kerogen by our methods can lead to carbon loss on the order of less than 5% (Durand and Nicaise, 1980). We doubt, however, whether this removal will have seriously impacted our isotopic interpretations, particularly as the methods were applied in the same way throughout the stratigraphic column and we are most interested in relative differences in isotopic compositions between the different sediment types rather than the absolute compositions. Also, simple acid treatment, typically, has a negligible influence on sediment organic isotopic composition (Könitzer et al., 2012). Finally, when we compare between the isotopic composition of bulk sediment TOC and kerogen TOC for unit 1 organic matter, the isotopic values are almost always within 1 per mil of one another, and typically within 0.5 per mil (see figure 1). We can view these differences as the maximum influence of analyses conducted in

different laboratories (bulk sediment in Odense and kerogen in Beijing) as well as all of the accumulated extraction steps.

Lines 161-172: What is the rational behind the 3 methods of iron determination? Why did you not do a split comparison on the unit 1?

Response: The rationale is that we did not have access to ICP-MS for all of our samples as this is a very expensive method. The utilization of concentrated HCl after sample heating is a common method for "total" Fe, and we find that it routinely extracts >95% of the total iron in standard reference materials. We prefer this to routine methods utilizing concentrated HF acid. Recognizing, however, that the HCl method leaves a small Fe residual, we have also begun to calibrate and use our hand-held x-ray fluorescence unit for total iron analyses. This device is excellent for iron with great reproducibility, stability, and accuracy when properly calibrated. This data set has evolved over the span of two years as we have added more data to best understand its geochemistry. Thus, the evolution of our methodology is in part due to the evolution of the data set and our understanding of unit 1. We have done many comparisons of our data utilizing these three different methods both on international standards and samples from unit 1 and other units within the Xiamaling Formation. We will add some of these comparisons to our supplemental information.

Lines 173-182: This should not be in the Methods section, please move to results and/or discussion.

Response: OK

3 Results Line 184: Please describe a bit of the results rather than just referring me to a supple- mental table. For example, averages and standard deviations could be given for black and green-gray shales within the text.

Response: OK.

Figure 1: How does the stratigraphic height on the Y axis relate within your or to your

unit number and the lithology? Descriptions of variations in the constituents with strati-graphic heights are largely lacking. For example in lines 188-189 the U/Al ratios are not always near the crustal values (e.g. around 30 m). Why?

Response: The lithology changes on a centimeter-decimeter scale which is way too detailed to indicate in a lithological reconstruction. Because of the tight correlation be-tween lithology and TOC content, the TOC content of a sample is an excellent indicator of its lithology, be it black shale or green-gray shale. There could be many reasons as to why U/Al is not always at the crustal average. One could be small differences in the U/Al ratio of the depositing clastic material. Another could be that U can also be en-riched in sediments also depositing in oxygenated environments (Barnes and Cochran, 1990). The main point, however, is the difference between uranium enrichment in the gray versus black shales. We agree that it would be important to discuss how U is also enriched in many modern sediments depositing under oxygenated environments, and will do so in the revised manuscript.

Lines 189-191: What is the significance of the higher Fe/Al values? No context for what a "higher value" even is- for example, is the difference observed between the ratios with shale type significant? Then what's the explanation of the differences with depth, which is often greater than differences between shale types?

Response: Higher values of Fe/Al are taken to indicate iron enrichments in a similar fashion to the ratio of FeHR/FeT. This has been discussed in the literature, and we reference these discussions (also referenced in the text; e.g. (Lyons and Severmann, 2006)). However, Fe/Al has not been as carefully calibrated as the FeHR/FeT, and is not generally the method of choice for determining iron enrichments and for elucidat-ing bottom water chemical conditions. However, in the present case, there is a clear trend of higher Fe/Al ratios in the outcrop samples experiencing also large trace metal enrichments. Therefore, the Fe/Al is consistent with our trace metal determinations in differentiating oxic and anoxic depositional conditions. We did not perform our standard iron extraction procedure on the outcrop samples, as these were somewhat weathered,

which could compromise the results. We do not believe that the Fe/Al ratio is critical to our interpretation, but it is a way that we can bind the outcrop and core results together.

Figure 2: This figure could benefit greatly from color separation of the green-gray and the black shales in ALL panels (such as in Figure 1). The figure description and much of the subsequent text is not clearly supported by the actual figure (e.g. the conclusion in line 309 is impossible to tell from the Figure 2 as is). Color-coding would help alleviate some of this confusion (for example discussions on lines 299-305, lines 307-309). Don't refer to another figure in a figure caption! The figure caption should be self-sufficient to the figure. Also, please explain in the text the significance of the line denoting the TOC wt % of 1%.

Response: These are all good points. One of the difficulties in the core samples is that the black and green-gray shales were not always easy to distinguish. This color differentiation becomes much more vivid and easy to see in the field samples where weathering has influenced the coloration of the rocks. However, for the field samples, 1 wt% TOC differentiates the black shales from the green-gray shales, and it is for this reason that we have highlighted this value of TOC in the Figure 2A. We will re-plot the data in Figure 2 to differentiate between the high TOC and low TOC samples, clarifing these points in the revised text.

4 Discussion Line 210: The figures referenced in this sentence don't support the claim. The best figure to support this statement is probably Figure 2, but based on this figure, HI and TOC don't look that correlated among like-shale types, as the different shale types separate out completely. The "linear" relationship across the two shale types don't support your conclusions here.

Response: Perhaps "relationship" is a better word than "correlation". But, it is clear that there is a strong "relationship" between HI and TOC among different shale types and this is the main point; that high TOC shales are associated with high HI and Low TOC shales are associated with low HI. Such a "relationship" is common in Phanerozoicaged sediments when the source of the organic matter is not heavily influenced by terrestrial plant material. We will change this in the revised manuscript and we agree that figure 2 best supports this discussion.

Line 215-217: What are the potential sources of OM for these age sediments?

Response: Good point. The potential sources are prokaryotic biomass, including cyanobacteria and other microbes involved in producing and degrading organic matter, as well as eukaryotic algae. There is no biomarker evidence for eukaryotes in this formation, but we cannot rule out the possibility that they were a part of the ecosystem as they likely have evolved by 1.4 Ga. We will clarify in the manuscript the potential sources of OM.

Lines 223-232: I don't understand the purpose of this whole discussion. How can you compare your sediments to these when the terrestrial/aquatic assemblages are totally different and the depositional regimes were likely vastly different? For example, you stated in line 215 that your sediments were deposited before land-plant evolution, so how can you compare the HI index (even with relation to O2 availability) you see in your samples to these other, newer sediments?

Response: We believe that this discussion is highly relevant. Please note that in both of the cases from Phanerozoic sediments that we highlight, terrestrial plant material is a minor component of the TOC pool. Thus, relationships between TOC and HI in these cases are a result of the influence of oxygen on marine organic matter degradation. The same principles would have influenced organic matter preservation in the Mesoproterozoic. Indeed, there is nothing particularly weird to be expected about this marine microbial biomass. It would have contained lipids, proteins and carbohydrates like Phanerozoic eukaryotic algae, and all expectations would be that it decomposes like any other lignin and cellulose-free algal biomass. There will of course be some differences in the exact nature of and relative proportions of these biomass components. But still, these are basic biomass components. We have performed some elemental

analysis of kerogens from unit 1, and they contain elemental ratios of H/C and O/C completely consistent with Type 1 biomass when plotted on a van Krevelen diagram (Figure 2). This would also be expected from microbial biomass. Thus, we have every reason to believe that unit 1 biomass fits well into what we know from the characteristics of Phanerozoic algal biomass (excluding land plants). Will include figure 2 into a revised version of the manuscript together with a more complete and nuanced discussion of the likely nature of unit 1 biomass. Remember also, it is critical that the trends in HI in the Xiamaling Fm correlate with other independent geochemical indicators of bottom water oxygenation, precisely as would be expected based on modern analogues.

Couldn't the HI index be similar/different for completely different reasons at this point? If this is not the case, please make a less confusing and sounder argument for why high TOC and HI samples are deposited un- der anoxic conditions and how this can be utilized for older sediments.

Response: We have absolutely no alternative explanation as to our observed trends in HI is related to bottom water oxygenation. As noted above, the trends in HI are completely compatible with independent geochemical evidence for the presence and absence of bottom water oxygenation and with the expectations based on modern analogues. Honestly, we don't feel that it gets much better or more compelling than this. One can argue that because the sediments deposited a long time ago, when the dominant sources of organic matter were different than today, that we cannot apply modern analogues to this ancient record. However, this would be an assertion without any justification, and it would not square with our independent geochemical observations of bottom water oxygenation as discussed above and in the manuscript. In the revised manuscript we can try to make these points more forcefully and to explain more clearly why the Phanerozoic sediments are applicable to sediments 1 billion years older. Perhaps a broader discussion of the types of organic matter available in the Mesoproterozoic would help.

This become especially important when you are discussing processes that can influence HI in more modern sediments (Lines 274-278), as these depositional processes could have been very different for the sediments you are comparing and using to build your argument. As the vary least, you need to discuss that the depositional regimes were likely sim- ilar for the sediments even though Eons apart.

Response: It's not clear to us that the reviewer is after here? What kind of changes in depositional conditions where organic carbon characteristics are comparable, other than oxygen, can influence similar relationships between HI and TOC in sediments, even though they are separated in time 1 billion years? We cannot come with any plausible alternatives without starting to make stuff up. In this case we feel it is best to apply Occam's razor and appeal to the most obvious explanation, which is oxygen, and a link between the Phanerozoic examples we discuss and the Mesoproterozoic samples that we report. Please recall that the Phanerozoic examples we discussed are not influenced by terrestrial organic matter. This point was made clearly in both of the publications that we quote.

I feel that much of this discussion in Section 4.1 could be condensed, strengthened and over all clarified. I feel like your conclusion in Lines 289-291, that there are fluctuations between anoxic and oxic de- position conditions, seems a bit of a leap from the arguments presented in the section above.

Response: We disagree, but hopefully a revised discussion will clarify this point.

Lines 292-293- Do you have a figure or reference to support this claim?

Response: Sure, we can reference the original iron speciation papers and some of the numerous trace metal papers that link trace metal enrichments to the presence or absence of oxygen in the bottom waters depositing sediments.

Lines 295-300- How do you know this is Fe enrichment and not Al depletion? A depletion in Al would give you the same ratio as an enrichment as Fe, potentially.

Response: We cannot be hundred percent certain. Indeed, as discussed above, the dynamics of Fe/Al are consistent with our interpretations of the dynamics of bottom water oxygenation, but are not, in themselves, proof of it.

What is happening at the 15 and 35 m mark in your core? The ratios also seem to overlap within the same range between the two types of shales. How do you know the variation you see is significant and indicative of the changes in depositional environment?

Response: Good point. As mentioned above, the Fe/Al ratio is supportive, not definitive evidence for bottom water chemical conditions. We actually debated whether or not to report the Fe/Al at all in the manuscript. However, as also mentioned above, the Fe/Al ratio results are completely consistent with our other geochemical indicators of bottom water oxygenation and they help to tie the outcrop and core material results together..

Line 408: Is the use of this density value in this context something that is already published in the literature? Cite if yes, but if not, this needs to be further validated!

Response: This is a typical density for dried mud. It is also the density most commonly used in sediment calculations. The actual value we use makes very little difference given the broad range of sedimentation rates that we explore. But, on revision, we will further validate this value.

Line 426-430 (Equation 3): A is not defined in the equation description

Response: It is defined in line 434. We can move this up into the previous paragraph.

Lines 519-527: I don't follow the logic of this argument. How can you compare your oxygen estimates with modern observations? Isn't it likely the mid-Proterozoic ocean was different (in terms of oxygen penetration depth especially) than your modern analogues, as we now live in a more oxygen rich world? So how does this comparison support your conclusions?

Response: There are lots of modern observations such as the Borderline Basins of

California and the low-oxygen regions where oxygen minimum zones impinge on sediments where bottom water oxygen levels are much lower than typical in the marine environment. Oxygen penetration depths and oxygen uptake rates have been determined in many of these environments, as shown in Figure 3C.

Line 550: I feel like it should be mentioned that there is no precise dating of unit 1 MUCH earlier in the text!

Response: OK. No problem.

Technical Corrections: This paper would benefit from a careful proofreading and calibration of citation software used. For example, there are many citation-related errors that probably have resulted from the citation software used: e.g. Lines 99, 147, etc. where parenthetical citations are inappropriate, lines 115-116 where the exact citations are presented twice in a row, lines 594-599 where the same exact reference is presented as two distinct references (Cole et al., 2016a and 2016b), line 542 where two references in a row are presented separately "(Planavsky et al., 2014) (Cole et al., 2016a)" as opposed to "(Planavsky et al., 2014; Cole et al., 2016)".

Response: Oops. Thanks

Lines 41-43: Sentence starting with "The original idea ... " is not a complete thought. Line 43, 44, 46, etc: Please define the element names before using their symbols at first use. There is a shift back and forth between the element name and symbol (e.g. in line 46 "chromium-associated" and "Fe-enriched"). Please at least be consistent with the usages. Line 50, 56: Please provide more background on your "unit" numbers. The lack of context what unit 3 is vs unit 1 is confusing for the readers to follow. A figure might help with this! Lines 68-72: Sentence starting with "These intrusive sills ... " is wordy and confusing! Line 73: " ... like the sediments depositing just before" I would consider rewording this phrase as it sounds a bit weird. Line 79-80: Add space between the degree symbol and N Lines 84-88: Sentence starting with "Previous work ... .". This sentence is 5 lines long! Break into shorter sentences. Line 103: This

should read " ... drilling depth, angle, and cross calibration ... " Line 115: "HI is defined as S2*100/TOC" (i.e. HI = S2*100/TOC) This is an equation, shouldn't this have an equation number? Line 148: " ... highly iron ... " do you mean highly reactive iron? Line 159: You already defined FeHR on line 148 Lines 218-220: This not a complete thought. Line 220-223: This sentence is confusing ! C7 Line 266: After organic carbon burial flux maybe place the abbreviation used in the equation? Cbur? Line 354: This transition "Returning to the sediment model ... " is awkward. I would consider re-wording it. Line 543: " ... chrome, and it's isotopes ... " Do you mean chromium? "It's isotopes" should read "its isotopes", as it's used as a possessive pronoun here. Line 546: " ... chrome component" do you mean chromium? Line 548: the usage of "square" is awkward. I would consider rewording this sentence. Interactive comment on Biogeosciences Discuss., doi:10.5194/bg-2016-413, 201

Anonymous Referee #2 "General comments" This paper aims to determine oxygen levels prevailing in oceans 1.4 Gyr ago. This topic is of interest and debated as evidenced by the numerous publications in high-rank journals. As stressed by the authors, the use of the chromium isotopes as a redox indicator can be discussed implying the need for complementary or new approaches as the one published by the same authors in PNAS (Zhang et al., 2016). However, this manuscript shares the same conclusion than Zhang et al., 2016. There are too many similarities between the two manuscripts to fully consider this one as a new manuscript. In my opinion, the manuscript must refocus on the approach rather than the already published conclusion. The two studies are only distinguished by (i) the studied geological units and (ii) the determination of oxygen exposure time and penetration depth: (i) Their previous study dealt with units 2,3 and 4 while this one focus on the unit 4. Unfortunately and as stressed by the authors (L550), there is no precise dating of unit 1 in the Xiamaling Formation. Hence, the main original conclusion of the manuscript (persistent atmo- spheric oxygen over million years) is not convincing especially in the light of the alter- nating gray (oxidizing conditions) and black shales (anoxic conditions; see Zhang et al., 2016).

Response: With all due respect to the reviewer's concerns, we are puzzled by the idea that "there are too many similarities between the two manuscripts[this manuscript and an earlier one by our group published in PNAS" to fully consider this one is a new manuscript" because ".. .this manuscript shares the same conclusion than Zhang et al. 2016." We stress: 1) that there is no consensus on levels of oxygen during the Meso-proterozoic Era. Basically, there is a pair of chromium isotope studies suggesting very low levels of atmospheric oxygen, and our PNAS contribution suggesting much higher levels, 2) that the relationship, therefore, between the history of atmospheric oxygen and animal evolution is currently unresolved, 3) the current manuscript uses a completely different methodology to also establish oxygen concentrations – that turns out to be similar - as the PNAS paper, which focused on on a completely different section of the Xiamaling Formation, 4) the methodology explored in the current manuscript is completely novel and is the first to evaluate the consequences of oxygen exposure on carbon preservation during the Precambrian. In short, the method is novel and holds promise to future application, which is a gain for the community no doubt, and additionally these independently achieved results contribute to resolve oxygen concentrations in the Mesoproterozoic. We also point to the impassioned comment to this manuscript by Devon Cole, the author of one of the chromium isotope papers referenced above. This comment is further demonstration that additional evidence on the levels of Meso-proterozoic oxygenation is critically needed.

Therefore, we view our manuscript as a novel contribution to an important scientific debate. The fact that our two contributions, utilizing completely different methods, offer similar conclusions as to levels of Mesoproterozoic atmospheric oxygen strengthens the idea that atmospheric oxygen levels were higher than predicted from chromium isotope studies and reinforce the idea that sufficient oxygen for animal respiration was available in the environment long before the evolution of animals themselves.

The reviewer also states that "The two studies are only distinguished by (i) the studied geological units and (ii) the determination of oxygen exposure time and penetration

depth" Yes, the studies do look at different parts of the Xiamaling Formation, but the methodology used to constrain atmospheric oxygen is completely and fundamentally different. In the PNAS paper we utilized a water column model to determine the minimum levels of atmospheric oxygen required to allow bottom water oxygenation. In the present manuscript we utilize a sediment diagenetic approach to explore the minimum levels of bottom water oxygen required to generate the amounts of carbon oxidation needed to reproduce the HI values we observe in the sediments

(ii) Although very interesting, providing a fair review on the determination oxygen exposure time and penetration depth is too difficult in the current state of the manuscript. The overall presentation is too confusing as for the distinction between anoxic and oxic depositional environments.

Response: The first reviewer also raised concerns about our discussion of the distinction between anoxic and anoxic depositional environments. We accept that a clearer discussion is in order, but we fully stand by our conclusions.

Finally, the quality of figures and tables does not stand for the publication standard. In my opinion, this manuscript is not fully original (because of its redundant conclusion with the PNAS paper) but presents an innovating approach. I suggest the authors to modify this manuscript focusing on the new unique feature of this version (TOC-derived calculations) with a significant improvement of the overall presentation.

Response: We have already addressed most of this comment above. However, we agree that the manuscript needs more focus on the nature and sources of organic matter to unit 1 of the Xiamaling Formation.

"Specific comments": Introduction The introduction is very short. There is an overall lack of contextualization. Moreover, authors finalize their introduction by evidencing that they will present equiv- alent results than their former publication. It is quite destabilizing.

Response: Good point. Reviewer 1 also raised the concern that the introduction is too short. We will provide a longer introduction with a better development of the methods that we finally use in our paper to determine oxygen levels.

L19-20: Where? It is not so clear in the manuscript. L43: Uranium (U) L44: Molybdenum (Mo) L49 "in contrast, sediments from unit 3": It is probably too specific for the global purpose of an introduction. Do you have other references than yours ?

Response: These points and will be addressed on revision.

Study site and methods L113-122: What were the samples analyzed by Rock-Eval pyrolysis ? kerogens or crushed rocks ?

Response: Yes, they were crushed rocks.

According to the logic of the manuscript, it may be crushed rocks. L113 HI: Hydrogen Index L114: Using the HI index required the determination of the TOC by Rock-Eval Pyrolysis. To determine the TOC, carbona- ceous compounds are also determined during the combustion step (oxidation oven). It must be clarified.

Response: TOC was determined on whole rock samples after decarbonization. This will be clarified in the revised manuscript.

Moreover, why don't you use the Oxygen Index ? It cannot be as a consequence of the presence of carbonates since Rock-Eval device provide the possi- bility to distinguish oxygen from both carbonates and organic matter sources (Baudin et al., 2015; organic geochemistry). OI is often used as a proxy of oxidation of the organic matter during early diagenesis. Same as before comments

Response: No problem with including the oxygen Index as well. Relationships between HI and OI show the organic matter to be a mixture of Type 1 and Type 2, as would be expected. In any event, in the revised manuscript we will include in the supplement a graph comparing H/C to O/C in order to help place this organic matter within the context of known Phanerozoic examples.

L116-117: This assertion is not true. Following cited references, S2 corresponds to the amount of hydrocarbon released upon pyroly- sis without any distinction about the molecular weight of the hydrocarbons.

Response: It is generally accepted that S2 comprises of the longer-chained, non-volatile hydrocarbons, S1 consists of the free hydrocarbons. This distinction would be consistent with textbook descriptions of Rock-Eval analyses, and with the discussion provided by the developers of the method.

L131-139: Isolation procedure does not correspond to the classical procedure, why ?

Response: Our extraction procedures are the same as the "classical" procedures introduced by (Durand and Nicaise, 1980). This reference will be included in the revised manuscript.

L131-139: have you investigated the effect of the isolation procedure on the preservation of or- ganic matter ? HCl procedure can lead to artefactual degradation of aliphatic moieties implying in turn, a shift in the carbon isotope composition.

Response: We have discussed our extraction procedures and their possible influence on the isotopic composition of carbon in our samples above, in response to the first reviewer's comments. In short, we do not believe that our extractions have significantly influenced the isotopic composition of our samples, and especially, our ability to comment on the differences in the isotopic composition between the low TOC and high TOC samples.

Results Figures: The quality of Figure 1 and 2 is not acceptable.

Response: How so? It's not clear what we should do to improve the figures?

Discussion 4.1 Organic carbon preservation and water column chemistry This discussion section is too long and not really pertinent.

Response: We believe that the discussion in this section is central to the development

of our arguments into distinguishing between oxic and aanoxic depositional conditions in unit 1 of the Xiamaling Formation. Without more specific instructions as to what is pertinent and not pertinent in this discussion, it is difficult for us to try to presuppose the reviewer's concerns.

The TOC and HI high values are attributed to the preservation of organic matter under anoxic conditions. In turn, I have the feeling that the TOC and HI low values are allocated to oxic environments by de- fault.

Response: No, low TOC and low HI are not allocated to oxic environments by default. We have allocated these to oxygen conditions based on comparisons with Phanerozoic environments, and in particular those without a terrestrial plant influence, where the relationship between oxygen, TOC, and HI is clearly distinguishable and well discussed. Note also that the allocations that we have provided are completely consistent with a line of independent geochemical determinations. We really don't believe that one can do a much better job than this.

Factually, the FeHR/FeT is the best criterion to distinguish between oxic and anoxic depositional environments. As the distinction between the oxic and anoxic en- viron- ments is crucial to sustain the determination of O2 level, it is essential to provide a more thorough argumentation. For instance, why their results about element traces are not compared with data from literature ?

Response: We agree that the iron speciation data is perhaps the best tool to distinguish between anoxic and an anoxic depositional conditions. This is why we have conducted this work and reported this data, which is completely consistent with our interpretations based on HI, TOC and trace metal enrichments. We do not quite understand the reviewer's comment in relationship to comparisons with the literature. Our interpretation of the trace metals are derived from their behavior in modern environments as reported in the literature and as discussed in the text. We appreciate, however, that our argumentation may have been unclear, and we will work hard to clarify our discussion and

the relationship between our results and those from the literature.

Moreover, interpretation of element traces is by far more complicated. In this case, there is again a lack of contextualization.

Response: Again, we could discuss the trace metals in much more depth, but we don't believe that we would come further in our utilization of the trace metals in distinguishing between oxygenated and anoxic depositional conditions. It is the patterns of trace metal enrichments that are most critical here. We will, however, make more reference to literature studies on trace metal enrichments.

L210 "HI has often been linked": References are needed L211: higher=>high L211: more => better preservation of L212 poorer => a low L218-232: This paragraph can be shortened. In its present form, it is confusing and not really interesting.

Response: We don't see quite how it is confusing, but we will work hard to try and streamline this paragraph.

L248-261: the good preservation of organic matter in anoxic environment is not new and does not deserve such a large paragraph.

Response: We agree that this discussion is straightforward. We will try to balance the wishes of this reviewer to shorten this discussion and those of reviewer 1 to enhance it.

L269-270: This sentence is in contradiction with your previous statement (see comment on L210)

Response: We agree that this sentence is unclear and we will change it.

L274; If HI values are "blurred" by sediment re-suspension or transport, why is it not the case for the TOC values?

Response: It is true for TOC values! We will clarify.

Indeed, HI and TOC characterize the same organic matter. L279-280: It is redundant with a previous sentence. L280:

13C was determined on the insoluble organic matter not on the whole bulk rock in contrast to Rock-Eval parameters. During the isolation of the kerogen, the use of HCl can degrade then aliphatic content leading in turn to a bias in the determination of the 13C. Have you any evidence that such a bias does not affect the 13C values ?

Response: This is discussed above

Anyway, I don't really understand the relevance of the 13C.

Response: As mentioned in the text, we use the 13C to argue for a similar source of organic matter to the black and gray shales. Such comparisons are often presented in the literature.

For the 4.2 discussion section, an extensive rewriting effort is required to simplify the whole structure. Indeed, several readings of the draft text are required to fully understand the approach. Otherwise, I've no concern about the calculations.

Response: We agree that this is a complicated discussion and will do our best to try to streamline and make it more accessible.

References Barnes, C. E. and Cochran, J. K.: Uranium removal in oceanic sediments in the oceanic U balance Earth Planet. Sci. Lett., 97, 94-101, 1990. Durand, B. and Nicaise, G.: Procedures for kerogen isolation. In: Kerogen-Insoluble Organic Matter from Sedimentary Rocks, Durand, B. (Ed.), Editions Technip, Paris, 1980. Könitzer, S. F., Leng, M. J., Davies, S. J., and Stepherson, M. H.: An assessment of geochemical preparation methods prior to organic carbon concentration in carbon isotope ratio analysis of fine-grain sedimentary rocks, Geochem. Geophys. Geosyst., 13, 2012. Lyons, T. W. and Severmann, S.: A critical look at iron paleoredox proxies: New insights from modern euxinic marine basins, Geochim Cosmochim Ac, 70, 5698-5722, 2006.

[Figure]

[Figure]

**Fig. 1.**

[Figure]

**Fig. 2.**

---

## Author Response (AR1)

**Response to reviewer's comments**

We thank the reviewers for their detailed comments and criticisms of our manuscript. Please find our detailed response to these comments below. Some of these responses reiterate what we have written in our initial response to the reviews comments. However, we elaborate on these and highlight where we have made additions and corrections to the MS. All in all, we have made moderate revisions to the MS. Please note that none of these revisions has altered our approach or conclusions. The most important of our major additions are:

- We have substantially expanded the Introduction as requested by all of the reviews (including Devon Cole, who added a comment to the MS). Thus, we have added to the Introduction a review of existing information on the state of atmospheric oxygen concentrations during the Mesoproterozoic Era. This helps us highlight on the novelty of the current contribution as noted by reviewer 2 and the editor.
- We have added a section on the nature of the Mesoproterozoic Era carbon cycle. We do this
  to emphasize the similarities and difference between the Mesoproterozoic and modern carbon
  cycles. This is important because we use the modern carbon cycle to evaluate and quantify the
  ancient carbon cycle yielding our oxygen estimates. This is aimed at the comments of
  reviewer 1. We remain firm in our belief that this is an appropriate approach.
- We have expanded our discussion on the use of HI to assess the maturity and state of decomposition of sedimentary organic matter.
- We have sharpened our methods section and added a table (Table S1) summarizing the results of repeat analyses of many international standards for most of the geochemical parameters reported here. This is in response to both of the reviewers and to the editor's comments.
- We have both sharpened and expanded our discussion as to how we use our geochemical parameters to assess bottom water chemistry. This has included both the addition of new text and rearrangement of the original text.
- We have both clarified and re-arranged the discussion in many places.

Specific responses to reviewers' comments

Anonymous Referee #1 Received and published: 16 November 2016 Review of Zhang et al., Bioegosciences Discussion DOI: 10.5194/bg-2016-413

**General Comments:**

This paper provides analysis for support of a compelling idea that oxygen was sufficient in the mid-Proterozoic ocean for evolution of higher-order species long before their actual evolution, suggesting that biological evolution was not limited by oxygen levels, as has been the suggested and currently accepted paradigm. This idea has been the subject of much debate in recent publications with recent work on Cr isotopes published in Science and Nature suggesting the O2 levels in the Mesoproterozoic are even lower (<1% PAL) than is currently widely accepted (1% PAL), and, therefore, this topic certainly warrants further study and evidence to determine oxygen levels in this important Eon. The authors are leaders in their field, and have presented other lines of evidence to support the view of an oxygenated Mesoproterozoic world in related recent publications. However, in general, I feel that the overall presentation of this paper could be clearer, better structured and more fluent. Often both the sentence structure and the argument structure are confusing, weakening the overall presentation of very novel and interesting work. The discussions presented in this paper rely heavily on a diagenetic model to determine oxygen necessary for estimated carbon mineralization (inferred from measured [TOC] and HI) in the paleo-ocean. I applaud the efforts of the authors for their efforts to calibrate the model to many modern "analogues". However, the presentation of this model and the subsequent conclusions are often convoluted, and do not present strong enough arguments to back the authors claims, especially in regards to the recent controversial evidence from other paleo-reconstruction work of oxygen levels in this Eon. Therefore, it is hard to follow the applied methodology (diagenetic model) from the way the discussion is currently presented. As much of the conclusions of this work rely on heavy data interpretation, I would recommend a clearer dialogue throughout the manuscript, which would strengthen the conclusions this work has in our understanding of Mesoproterozoic oxygen levels.

Response: We appreciate these comments and the general support for our approach. We have taken these comments to heart in our revisions as outlined both above and below.

**Specific Comments:**

Comments for throughout the manuscript-

It seems like there are many "missing" references for methodology used and some discussion points. This opens room for debating the usage of the approaches outlined in the paper. e.g. lines 105, 110, 135-138 for methodology lines 211, 220, 250, 293 to support specific claims made in the paper.

Response: Thanks. We have carefully evaluated our Methods section and added the missing references.

**1 Introduction-**

Authors must keep in mind the broad readership of Biogeosciences as a journal and define the background and significance of this work in a bit more detail. The intro feels short and incomplete, especially in regards to framing the current state of the research and preparing the readers for the in-depth models that follow.

Response: We agree. This was also a comment of Reviewer 2. As noted above, the Introduction has been considerably expanded.

2 Methods-

2.1 Study Site: While the authors have obviously been working with the Xiamaling Formation and are vary familiar with its stratigraphy, the readers would benefit from a figure defining the "units" of the formation and it's overall place in the paleo-record (ie. dating).

Response: Good point. We have added a stratigraphic overview of the Xiamaling Formation including a general stratigraphy of the upper 4 units of the Formation, including an indication of where dating has been done, and a more detailed stratigraphy of the upper 45 meters of unit 1.

2.2 Sample Collection and Analytical Methods:

*Lines 103-105: for the geochemical data, do you have a reference for the preparation of these samples?*

Response: Yes, added.

Line 109: Are the accuracies presented adequate? For example, it is often accepted in organic geochemical analyses that over 10% variability is "too high". Is this acceptable for trace metal work? Do you have a reference to support that your methodology is acceptable?

Response: Prompted by this comment, we have gone back to reevaluate our methods and methods description. We realize that we should have been more careful in our presentation of the methods and their uncertainties. By far the most of our trace metal data was obtained by ICP-MS with uncertainties in the 1% range, not the ranges reported in the manuscript. We also generated some of our results with a hand-held XRF calibrated against numerous international standards and several splits of Xiamaling sediment independently calibrated with ICP-MS. Our uncertainties with this method were all better than 5%. We have redrafted this discussion, and we summarized our repeat analysis of standards and samples in Table S!

*Line 110: Define what major elements you looked at with X-ray fluorescence and a reference to the method used.*

Response: Done.

Line 113: Redefine HI, as you only have defined it in the abstract.

Response: Done.

*Line 115: How was TOC and S2 measured? Pyrolysis? This description is a little Confusing.*

Response: TOC was measured with an elemental analyzer, while S2 was measured with standard Rock-Eval pyrolysis. This has been clarified.

Line 125: How did you dry your samples?

Response: Samples were dried in a muffle furnace with the temperature <40°C. Statement added.

Line 125-130: Is this TOC measured on the C-S analyzer and the EA used for the HI presented in the paragraph above? If so, you should present this first so the readers

aren't wondering where the data came from. Is the uncertainty presented for the same standards an inter-lab comparison, or is it for the uncertainty between replicates in the same lab?

Response: Good point, we have our methods description as suggested. The uncertainties in TOC were presented for replicates run in the same lab.

Lines 131-139: Are these methods previously published? References?

Response: Yes, (Durand and Nicaise, 1980).

Line 138: Has there been any work done on losses or transformations of kerogen OM in regards to the removal methods of the carbonates, silicates, and fluorides? For example, in modern terrestrial and aquatic samples, significant losses and alterations to OM content, composition and isotopic fractionation have been observed with use of concentrated acids and rinse steps (See e.g.: Gélinas et al., Organic Geochemistry, 2001 DOI: 10.1016/S0146-6380(01)00018-3 and discussion in Brodie et al., Chemical Geology, 2011 DOI: 10.106/j.chemgeo.2011.01.007).

Response: We have followed standard protocols in all of our analytical techniques. It is true that these techniques could have removed or transformed some organic matter during their application. As noted in the MS, the procedures we use lead to carbon loss of less than 5% (Durand and Nicaise, 1980), and we doubt whether this small amount of loss will have seriously impacted our isotopic interpretations, particularly as the methods were applied in the same way throughout the stratigraphic column and we are most interested in relative differences in isotopic compositions between the different sediment types rather than the absolute compositions. Also, simple acid treatment, typically, has a negligible influence on sediment organic isotopic composition of bulk sediment TOC and kerogen TOC for unit 1 organic matter, the isotopic values are almost always within 1 per mil of one another, and typically within 0.5 per mil (see figure 1). We can view these differences as the maximum influence of analyses conducted in different laboratories (bulk sediment in Odense and kerogen in Beijing) as well as all of the accumulated extraction steps.

**Lines 161-172: What is the rational behind the 3 methods of iron determination? Why did you not do a split comparison on the unit 1?**

Response: The rationale is that we did not have access to ICP-MS for all of our samples as this is a very expensive method. The utilization of concentrated HCl after sample heating is a common method for "total" Fe, and we find that it routinely extracts >95% of the total iron in standard reference materials. We prefer this to routine methods utilizing concentrated HF acid. Recognizing, however, that the HCl method leaves a small Fe residual, we have also begun to calibrate and use our hand-held x-ray fluorescence unit for total iron analyses. This device is excellent for iron with great reproducibility, stability, and accuracy when properly calibrated. This data set has evolved over the span of two years as we have added more data to best

understand its geochemistry. Thus, the evolution of our methodology is in part due to the evolution of the data set and our understanding of unit 1. All of our methods have been carefully evaluated against standard materials. We have added a discussion of this in the revived MS.

*Lines 173-182: This should not be in the Methods section, please move to results and/or discussion.*

**Response: Done.**

**3 Results**

Line 184: Please describe a bit of the results rather than just referring me to a supplemental table. For example, averages and standard deviations could be given for black and green-gray shales within the text.

Response: Done. Added Table 1.

Figure 1: How does the stratigraphic height on the Y axis relate within your or to your unit number and the lithology? Descriptions of variations in the constituents with stratigraphic heights are largely lacking. For example in lines 188-189 the U/Al ratios are not always near the crustal values (e.g. around 30 m). Why?

Response: The lithology changes on a centimeter-decimeter scale which is way too detailed to indicate in a lithological reconstruction. Because of the tight correlation between lithology and TOC content, the TOC content of a sample is an excellent indicator of its lithology, be it black shale or green-gray shale. There could be many reasons as to why U/Al is not always at the crustal average. One could be small differences in the U/Al ratio of the depositing clastic material. Another could be that U can also be enriched in sediments also depositing in oxygenated environments (Barnes and Cochran, 1990). The main point, however, is the difference between uranium enrichment in the gray versus black shales. Most of this discussion has been re-written.

Lines 189-191: What is the significance of the higher Fe/Al values? No context for what a "higher value" even is- for example, is the difference observed between the ratios with shale type significant? Then what's the explanation of the differences with depth, which is often greater than differences between shale types?

Response: Higher values of Fe/Al are taken to indicate iron enrichments in a similar fashion to the ratio of FeHR/FeT. This has been discussed in the literature, and we reference these discussions (also referenced in the text; e.g. (Lyons and Severmann, 2006)), but have also expanded this discussion in the MS. We note, however, that Fe/Al has not been as carefully calibrated as the FeHR/FeT, and is not generally the method of choice for determining iron enrichments and for elucidating bottom water chemical conditions. However, in the present case, there is a clear trend of higher Fe/Al ratios in the outcrop samples experiencing also large trace metal enrichments. Therefore, the Fe/Al is consistent with our trace metal determinations in differentiating oxic and anoxic depositional conditions. We did not perform our standard iron

extraction procedure on the outcrop samples, as they are weathered, which would compromise the results.

Figure 2: This figure could benefit greatly from color separation of the green-gray and the black shales in ALL panels (such as in Figure 1). The figure description and much of the subsequent text is not clearly supported by the actual figure (e.g. the conclusion in line 309 is impossible to tell from the Figure 2 as is). Color-coding would help alleviate some of this confusion (for example discussions on lines 299-305, lines 307-309). Don't refer to another figure in a figure caption! The figure caption should be self-sufficient to the figure. Also, please explain in the text the significance of the line denoting the TOC wt % of 1%.

Response: These are all good points. One of the difficulties in the core samples is that the black and green-gray shales were not always easy to distinguish. This color differentiation becomes much more vivid and easy to see in the field samples where weathering has influenced the coloration of the rocks. However, for the field samples, by far most of the green-gray samples had a TOC content 0f <0.5 wt% while the black shales had a TOC content of >2 wt%. As a guide helping to relate the field samples to the core samples, we have indicated with boxes these TOC ranges in Figure (now) 3a.

**4 Discussion**

Line 210: The figures referenced in this sentence don't support the claim. The best figure to support this statement is probably Figure 2, but based on this figure, HI and TOC don't look that correlated among like-shale types, as the different shale types separate out completely. The "linear" relationship across the two shale types don't support your conclusions here.

Response: Perhaps "relationship" is a better word than "correlation". Also, there was intention to suggest that such a "relationship" is linear. But, it is clear, however, that there is a strong "relationship" between HI and TOC among different shale types and this is the main point; that high TOC shales are associated with high HI and Low TOC shales are associated with low HI. Such a "relationship" is common in Phanerozoic-aged sediments when the source of the organic matter is not heavily influenced by terrestrial plant material. This text has been significantly reworked and expanded in the revised MS.

**Line 215-217: What are the potential sources of OM for these age sediments?**

Response: Good point. The potential sources are prokaryotic biomass, including cyanobacteria and other microbes involved in producing and degrading organic matter, as well as eukaryotic algae. There is no biomarker evidence for eukaryotes in this formation, but we cannot rule out the possibility that they were a part of the ecosystem as they likely have evolved by 1.4 Ga. We have added a whole new section in the discussion describing the Mesoproterozoic Era carbon cycle.

Lines 223-232: I don't understand the purpose of this whole discussion. How can you compare your sediments to these when the terrestrial/aquatic assemblages are totally different and the depositional regimes were likely vastly different? For example, you stated in line 215 that your sediments were deposited before land-plant evolution, so how can you compare the HI index (even with relation to O2 availability) you see in your samples to these other, newer sediments?

Response: We believe that this discussion is highly relevant. Please note that in both of the cases from Phanerozoic sediments that we highlight, terrestrial plant material is a minor component of the TOC pool. Thus, relationships between TOC and HI in these cases are a result of the influence of oxygen on marine organic matter degradation. We have expanded our discussion to bring even more example of the relationship between HI, organic matter preservation, and oxygen. The same principles would have influenced organic matter preservation in the Mesoproterozoic. Indeed, there is nothing particularly weird to be expected about this marine microbial biomass. It would have contained lipids, proteins and carbohydrates like Phanerozoic eukaryotic algae, and all expectations would be that it decomposes like any other lignin and cellulose-free algal biomass. There will of course be some differences in the exact nature of and relative proportions of these biomass components. But still, these are basic biomass components. As mentioned above, we have added a new section describing the nature of the Mesoproterozoic Era carbon cycle. Remember also that the trends in HI in the Xiamaling Fm correlate with other independent geochemical indicators of bottom water oxygenation, precisely as would be expected based on modern analogues.

Couldn't the HI index be similar/different for completely different reasons at this point? If this is not the case, please make a less confusing and sounder argument for why high TOC and HI samples are deposited under anoxic conditions and how this can be utilized for older sediments.

Response: We have absolutely no alternative explanation as to our observed trends in HI is related to bottom water oxygenation. As noted above, the trends in HI are completely compatible with independent geochemical evidence for the presence and absence of bottom water oxygenation and with the expectations based on modern analogues. Honestly, we do not feel that it gets much better or more compelling than this. One can argue that because the sediments deposited a long time ago, when the dominant sources of organic matter were different than today that we cannot apply modern analogues to this ancient record. However, this would be an assertion without any justification, and it would not square with our independent geochemical observations of bottom water oxygenation as discussed above and in the manuscript. As mentioned above, in the revised MS we have expanded our discussion of both the Mesoproterozoic Era carbon cycle and of the relationship between Hi, oxygen and carbon preservation.

**This become**

especially important when you are discussing processes that can influence HI in more

modern sediments (Lines 274-278), as these depositional processes could have been very different for the sediments you are comparing and using to build your argument. As the vary least, you need to discuss that the depositional regimes were likely similar for the sediments even though Eons apart.

Response: It's not clear to us what the reviewer is after here? What kind of changes in depositional conditions where organic carbon characteristics are comparable, other than oxygen, can influence similar relationships between HI and TOC in sediments, even though they are separated in time of 1 billion years? We cannot come with any plausible alternatives without starting to make stuff up. In this case we feel it is best to apply Occam's razor and appeal to the most obvious explanation, which is oxygen, and a link between the Phanerozoic examples we discuss and the Mesoproterozoic samples that we report. Please recall that the Phanerozoic examples we discussed are not influenced by terrestrial organic matter. This point was made clearly in both of the publications that we quote.

**I feel that much of this discussion in**

Section 4.1 could be condensed, strengthened and over all clarified. I feel like your conclusion in Lines 289-291, that there are fluctuations between anoxic and oxic deposition conditions, seems a bit of a leap from the arguments presented in the section above.

Response: We disagree and have rearranged and expanded our discussion to more clearly develop the relationship between our geochemical indicators and our assessment of bottom water oxygenation. We believe that out data provides about the most compelling evidence for fluctuating bottom water conditions that exists in the literature.

**Lines 292-293- Do you have a figure or reference to support this claim?**

Response: Sure, we have referenced the original iron speciation papers and more of the numerous trace metal papers that link trace metal enrichments to the presence or absence of oxygen in the bottom waters depositing sediments.

*Lines* 295-300- *How do you know this is Fe enrichment and not Al depletion? A depletion in Al would give you the same ratio as an enrichment as Fe, potentially.*

Response: We cannot be hundred percent certain. Indeed, as discussed above, the dynamics of Fe/Al are consistent with our interpretations of the dynamics of bottom water oxygenation, but are not, in themselves, proof of it.

What

is happening at the 15 and 35 m mark in your core? The ratios also seem to overlap within the same range between the two types of shales. How do you know the variation you see is significant and indicative of the changes in depositional environment?

Response: Good point. As mentioned above, the Fe/Al ratio is supportive, not definitive evidence for bottom water chemical conditions. We actually debated whether or not to report the Fe/Al at all in the manuscript. However, as also mentioned above, the Fe/Al ratio results are completely consistent with our other geochemical indicators of bottom water oxygenation and they help to tie the outcrop and core material results together.

**Line 408: Is the use of this density value in this context something that is already published in the literature? Cite if yes, but if not, this needs to be further validated!**

Response: This is a typical density for dried mud. It is also the density most commonly used in sediment calculations. The actual value we use makes very little difference given the broad range of sedimentation rates that we explore. We have further validated this value in the revised MS.

**Line 426-430 (Equation 3): A is not defined in the equation description**

Response: It is defined in line 434. We have reorganized the text to make the reference to the various equations more clear.

Lines 519-527: I don't follow the logic of this argument. How can you compare your oxygen estimates with modern observations? Isn't it likely the mid-Proterozoic ocean was different (in terms of oxygen penetration depth especially) than your modern analogues, as we now live in a more oxygen rich world? So how does this comparison support your conclusions?

Response: There are lots of modern observations such as the Borderline Basins of California and the low-oxygen regions where oxygen minimum zones impinge on sediments where bottom water oxygen levels are much lower than typical in the marine environment. Oxygen penetration depths and oxygen uptake rates have been determined in many of these environments, as shown in Figure 4C.

**Line 550: I feel like it should be mentioned that there is no precise dating of unit 1 MUCH earlier in the text!**

Response: Mentioned in site description and obvious from new Figure 1.

**Technical Corrections:**

This paper would benefit from a careful proofreading and calibration of citation software used. For example, there are many citation-related errors that probably have resulted from the citation software used: e.g. Lines 99, 147, etc. where parenthetical citations are inappropriate, lines 115-116 where the exact citations are presented twice in a row, lines 594-599 where the same exact reference is presented as two distinct references (Cole et al., 2016a and 2016b), line 542 where two references in a row are presented separately "(Planavsky et al., 2014) (Cole et al., 2016a)" as opposed to "(Planavsky et al., 2016)".

Response: Corrected. Also, we have dealt with all of the corrections and suggestions presented below.

Lines 41-43: Sentence starting with "The original idea

" is not a complete thought.

Line 43, 44, 46, etc: Please define the element names before using their symbols at first use. There is a shift back and forth between the element name and symbol (e.g. in line 46 "chromium-associated" and "Fe-enriched"). Please at least be consistent with the usages. Line 50, 56: Please provide more background on your "unit" numbers. The lack of context what unit 3 is vs unit 1 is confusing for the readers to follow. A figure might help with this! Lines 68-72: Sentence starting with "These intrusive sills " is wordy and confusing! Line 73: " like the sediments depositing just before" I would consider rewording this phrase as it sounds a bit weird. Line 79-80: Add space between the degree symbol and N Lines 84-88: Sentence starting with "Previous work .". This sentence is 5 lines long! Break into shorter sentences. Line 103: This should read " drilling depth, angle, and cross calibration ... ,, Line 115: "HI is defined as S2\*100/TOC" (i.e. HI = S2\*100/TOC) This is an equation, shouldn't this have an equation number? Line 148: " ... highly iron " do you mean highly reactive iron? Line 159: You already defined FeHR on line 148 Lines 218-220: This not a complete thought. Line 220-223: This sentence is confusing ! C7 Line 266: After organic carbon burial flux maybe place the abbreviation used in the equation? Cbur? Line 354: This transition "Returning to the sediment model " is awkward. I would

consider re-wording it. Line 543: " ... chrome, and it's isotopes ... "Do you mean chromium? "It's isotopes" should read "its isotopes", as it's used as a possessive pronoun here. Line 546: "

chrome component" do you mean chromium?

Line 548: the usage of "square" is awkward. I would consider rewording this sentence. Interactive comment on Biogeosciences Discuss., doi:10.5194/bg-2016-413, 201

**Anonymous Referee #2**

Received and published: 22 December 2016

"General comments"

This paper aims to determine oxygen levels prevailing in oceans 1.4 Gyr ago. This topic is of interest and debated as evidenced by the numerous publications in high-rank journals. As stressed by the authors, the use of the chromium isotopes as a redox indicator can be discussed implying the need for complementary or new approaches as the one published by the same authors in PNAS (Zhang et al., 2016). However, this manuscript shares the same conclusion than Zhang et al., 2016. There are too many similarities between the two manuscripts to fully consider this one as a new manuscript. In my opinion, the manuscript must refocus on the approach rather than the already published conclusion. The two studies are only distinguished by (i) the studied geological units and (ii) the

determination of oxygen exposure time and penetration depth: (i) Their previous study dealt with units 2,3 and 4 while this one focus on the unit 4. Unfortunately and as stressed by the authors (L550), there is no precise dating of unit 1 in the Xiamaling Formation. Hence, the main original conclusion of the manuscript (persistent atmospheric oxygen over million years) is not convincing especially in the light of the alternating gray (oxidizing conditions) and black shales (anoxic conditions; see Zhang et al., 2016).

Response: With all due respect to the reviewer's concerns, we are puzzled by the idea that "there are too many similarities between the two manuscripts [this manuscript and an earlier one by our group published in PNAS" to fully consider this one is a new manuscript" because "... this manuscript shares the same conclusion than Zhang et al. 2016." We stress: 1) that there is no consensus on levels of oxygen during the Mesoproterozoic Era. Basically, there is a pair of chromium isotope studies suggesting very low levels of atmospheric oxygen, and our PNAS contribution suggesting much higher levels, 2) that the relationship, therefore, between the history of atmospheric oxygen and animal evolution is currently unresolved, 3) the current manuscript uses a completely different methodology to also establish oxygen concentrations – that turns out to be similar - as the PNAS paper, which focused on a completely different section of the Xiamaling Formation, 4) the methodology explored in the current manuscript is

completely novel and is the first to evaluate the consequences of oxygen exposure on carbon preservation during the Precambrian. In short, the method is novel and holds promise to future application, which is a gain for the community no doubt, and additionally these independently achieved results contribute to resolve oxygen concentrations in the Mesoproterozoic. We also point to the impassioned comment to this manuscript by Devon Cole, the author of one of the chromium isotope papers referenced above. This comment is further demonstration that additional evidence on the levels of Mesoproterozoic oxygenation is critically needed.

Therefore, we view our manuscript as a novel contribution to an important scientific debate. The fact that our two contributions, utilizing completely different methods, offer similar conclusions as to levels of Mesoproterozoic atmospheric oxygen strengthens the idea that atmospheric oxygen levels were higher than predicted from chromium isotope studies and reinforce the idea that sufficient oxygen for animal respiration was available in the environment long before the evolution of animals themselves.

The reviewer also states that "The two studies are only distinguished by (i) the studied geological units and (ii) the determination of oxygen exposure time and penetration depth". Yes, the studies do look at different parts of the Xiamaling Formation, but the methodology used to constrain atmospheric oxygen is completely and fundamentally different. In the PNAS paper we utilized a water column model to determine the minimum levels of atmospheric oxygen required to allow bottom water oxygenation. In the present manuscript we utilize a sediment diagenetic approach to explore the minimum levels of bottom water oxygen required to generate the amounts of carbon oxidation needed to reproduce the HI values we observe in the sediments.

We have expanded in our introduction evidence related to the history of atmospheric oxygen. This makes a robust case for the need of the present contribution.

(ii) Although very interesting, providing a fair review on the determination oxygen exposure time and penetration depth is too difficult in the current state of the manuscript. The overall presentation is too confusing as for the distinction between anoxic and oxic depositional environments.

Response: The first reviewer also raised concerns about our discussion of the distinction between anoxic and anoxic depositional environments. As noted above in response to reviewer, our discussion of the evidence related to the presence or absence of bottom water oxygen during the deposition of unit 1 has been reorganized and expanded. We fully stand by our conclusions.

Finally, the quality of figures and tables does not stand for the publication standard. In my opinion, this manuscript is not fully original (because of its redundant conclusion with the PNAS paper) but presents an innovating approach. I suggest the authors to modify this manuscript focusing on the new unique feature of this version (TOC-derived calculations) with a significant improvement of the overall presentation.

Response: We have already addressed most of this comment above. However, we agree that the manuscript needs more focus on the nature and sources of organic matter to unit 1 of the Xiamaling Formation.

"Specific comments":

Introduction The introduction is very short. There is an overall lack of contextualization. Moreover, authors finalize their introduction by evidencing that they will present equivalent results than their former publication. It is quite destabilizing.

Response: Good point. The Introduction has been considerably expanded:

L19-20: Where? It is not so clear in the manuscript. L43: Uranium (U) L44: Molybdenum (Mo) L49 "in contrast, sediments from unit 3": It is probably too specific for the global purpose of an introduction. Do you have other references than yours ?

Response: This part of the Introduction has been rewritten and expanded.

Study site and methods L113-122: What were the samples analyzed by Rock-Eval pyrolysis ? kerogens or crushed rocks ?

Response: Yes, they were crushed rocks.

According to the logic of the manuscript, it may be crushed rocks. L113 HI: Hydrogen Index L114: Using the HI index required the determination of the TOC by Rock-Eval Pyrolysis. To determine the TOC, carbonaceous compounds are also determined during the combustion step (oxidation oven). It must be clarified.

Response: No, TOC was determined on whole rock samples after decarbonization. We have made this statement explicitly in the revised MS.

Moreover, why don't you use the Oxygen Index ? It cannot be as a consequence of the presence of carbonates since Rock-Eval device provide the possibility to distinguish oxygen from both carbonates and organic matter sources (Baudin et al., 2015; organic geochemistry). OI is often used as a proxy of oxidation of the organic matter during early diagenesis. Same as before comments

Response: The oxygen index is compromised in outcrop samples, so is of little values here. These samples contained negligible carbonate.

L116-117: This assertion is not true. Following cited references, S2 corresponds to the amount of hydrocarbon released upon pyrolysis without any distinction about the molecular weight of the hydrocarbons. Response: It is generally accepted that S2 comprises of the longer-chained, non-volatile hydrocarbons, S1 consists of the free hydrocarbons. This distinction would be consistent with textbook descriptions of Rock-Eval analyses, and with the discussion provided by the developers of the method.

**L131-139: Isolation procedure does not correspond to the classical procedure, why?**

Response: Our extraction procedures are the same as the "classical" procedures introduced by (Durand and Nicaise, 1980). This reference will be included in the revised manuscript.

**L131-139:**

have you investigated the effect of the isolation procedure on the preservation of organic matter ? HCl procedure can lead to artefactual degradation of aliphatic moieties implying in turn, a shift in the carbon isotope composition.

Response: We have discussed our extraction procedures and their possible influence on the isotopic composition of carbon in our samples above, in response to the first reviewer's comments. In short, we do not believe that our extractions have significantly influenced the isotopic composition of our samples, and especially, our ability to comment on the differences in the isotopic composition between the low TOC and high TOC samples. This discussion has been included in the revised text.

**Results Figures: The quality of Figure 1 and 2 is not acceptable.**

Response: How so? It's not clear what we should do to improve the figures?

Discussion 4.1 Organic carbon preservation and water column chemistry This discussion section is too long and not really pertinent.

Response: We believe that the discussion in this section is central to the development of our arguments into distinguishing between oxic and anoxic depositional conditions in unit 1 of the Xiamaling Formation. Without more specific instructions as to what is pertinent and not pertinent in this discussion, it is difficult for us to try to presuppose the reviewer's concerns.

**The TOC and HI high values are**

attributed to the preservation of organic matter under anoxic conditions. In turn, I have the feeling that the TOC and HI low values are allocated to oxic environments by default.

Response: No, low TOC and low HI are not allocated to oxic environments by default. We have allocated these to oxygen conditions based on comparisons with Phanerozoic environments, and

in particular those without a terrestrial plant influence, where the relationship between oxygen, TOC, and HI is clearly distinguishable and well discussed. Note also that the allocations that we have provided are completely consistent with a line of independent geochemical determinations. We really don't believe that one can do a much better job than this. We have, however, revised our text make these arguments clearer.

Factually, the FeHR/FeT is the best criterion to distinguish between oxic and anoxic depositional environments. As the distinction between the oxic and anoxic environments is crucial to sustain the determination of O2 level, it is essential to provide a more thorough argumentation. For instance, why their results about element traces are not compared with data from literature ?

Response: We agree that the iron speciation data is perhaps the best tool to distinguish between anoxic and an anoxic depositional conditions. This is why we have conducted this work and reported this data, which is completely consistent with our interpretations based on HI, TOC and trace metal enrichments. We do not quite understand the reviewer's comment in relationship to comparisons with the literature. Our interpretation of the trace metals are derived from their behavior in modern environments as reported in the literature and as discussed in the text. We appreciate, however, that our argumentation may have been unclear, and we reorganized our discussion to better describe how our results are used to distinguish the chemistry of depositional environments.

**Moreover, interpretation of element traces is by far more complicated. In this case, there is again a lack of contextualization.**

Response: Again, we could discuss the trace metals in much more depth, but we don't believe that we would come further in our utilization of the trace metals in distinguishing between oxygenated and anoxic depositional conditions. It is the patterns of trace metal enrichments that are most critical here.

L210 "HI has often been linked": References are needed L211: higher=>high L211: more => better preservation of L212 poorer => a low L218-232: This paragraph can be shortened. In its present form, it is confusing and not really interesting.

Response: This discussion has been completely reworked.

L248-261: the good preservation of organic matter in anoxic environment is not new and does not deserve such a large paragraph.

Response: We agree that this discussion is straightforward. In the revised MS we have tried to balance the wishes of this reviewer to shorten this discussion and those of reviewer 1 to enhance it.

with your previous statement (see comment on L210)

Response: This discussion has been rewritten.

L274; If HI values are "blurred" by sediment re-suspension or transport, why is it not the case for the TOC values?

Response: This whole discussion has been rewritten and sharpened.

Indeed, HI and TOC characterize the same organic matter. L279-280: It is redundant with a previous sentence. L280:

13C was determined on the insoluble organic matter not on the whole bulk rock in contrast to Rock-Eval parameters. During the isolation of the kerogen, the use of HCl can degrade then aliphatic content leading in turn to a bias in the determination of the 13C. Have you any evidence that such a bias does not affect the 13C values ?

Response: This is discussed above.

Anyway, I don't really understand the relevance of the 13C.

Response: As mentioned in the text, we use the 13C to argue for a similar source of organic matter to the black and gray shales. Such comparisons are often presented in the literature.

For the 4.2 discussion section, an extensive rewriting effort is required to simplify the whole structure. Indeed, several readings of the draft text are required to fully understand the approach. Otherwise, I've no concern about the calculations.

Response: Agreed and we have rewritten to simplify the structure and to clarify the arguments.

This difference could relate to differences in the relative efficiencies of oxic vs anoxic mineralization, or We would expect this to differences in the initial composition of the particles originating at the different sites. If oxic vs anoxic decomposition is the main factor driving these TOC differences, then the differences would likelydifference to be even smaller for particles settling to the shallower water depths of 50 to 200 meters as we surmise for unit 1 of the ; representing the likely water depth range for the parts of the Xiamaling Formation unit 1 we are exploring.\_\_The Arabian Sea results also reinforce a general observation that throughout the global ocean, particles settling though the upper 100s of meterslayer of theoxygenated marine water columns (100s of meters) are quite TOC-enriched, with values much closer higher (typically 3 to 15 wt%;\_Armstrong et al., 2002; Honjo et al., 1982-), to{Honjo, 1982

29

**1245;Armstrong, 2002 #12828] than those observed in the black shales than those observed in the green-gray shales of unit 1 (Table 1). OverallThus, we argue that the differences in the TOC content between the green-gray and black shales in unit 1 were likely most likelymostly driven by differences in sediment organic carbon preservation, as determined by the presence or absence of bottom\_-water oxygen, and not by differences in water column processes. This assessment is based on: 1) the relatively small differences in the TOC content of particles settling through oxic and anoxic waters of the Arabian Sea, and 2) the observation that the green-gray shales of unit 1 have TOC contents much reduced compared to particles settling through the upper 100s of meters of the marine water column.**

**-4.6 Constraining oxygen levels**

**Our goal now is to determine the levels of bottom-water**

We will now draw from modern observations and develop a simple sediment diagenetic model to constrain the oxygen levels required to account for the patterns of carbon preservation in Xiamaling unit 1. First, however, we note that even though the TOC content of particles settling through the oxic and anoxic water columns of Xiamaling unit 1 was likely similar, some 20% to 60 % of the settling organic matter likely mineralized as particles settled from the base of the upper mixed layer to the sediment surface at some 50 to 200 meters water depth (e.g. {Martin, 1987 #1776;Marsay, 2015 #13088;Lamborg, 2008 #13091;Keil, 2015 #13217}). This organic matter mineralization amounts to an oxygen sink and a reduction in water column oxygen levels, that is not included in our model. To include this oxygen sink would raise our minimum oxygen estimates by the magnitude of the sink. Thus, ignoring this oxygen sink is one way in which our model provides a conservative minimum estimate for atmospheric oxygen levels.

**Formatted: Indent: First line: 0 cm**

Returning to the sediment model, we now estimate the amount of oxygen required to account for vield the differences in carbon preservation between the green-gray and black shales of unit 1, which we assume, from the discussion above (Sect.4.2), to be a factor of 10. Our model is constrained from modern observations through a multi-step process. Our first step is to revisit the observation that organic carbon TOC preservation in modern marine sediments scales is controlled by with both sedimentation rate and with sedimentary environment as shown in (Figure 43a). To utilize these trends, To proceed, therefore, we must first estimate the rate of sedimentdepositionation 
[revised manuscript text omitted]

afrom Rudnick (2004)

| scenario       | Sed rate                    | %C    | %C    | O 2 |
|----------------|-----------------------------|-------|-------|----------------|
|                |                             | pres  | pres  | exposure       |
|                |                             |       |       | time           |
|                | $g \text{ cm}^2 \text{y}^1$ | black | gray  | у              |
|                |                             | shale | shale |                |
| XML*0.5        | $0.8 \times 10^{-3}$        | 12    | 1.2   | 700-6000       |
| XML            | $1.7 \times 10^{-3}$        | 20-30 | 2-3   | 400-5000       |
| XML*10         | $1.7 \times 10^{-2}$        | 40-80 | 4-8   | 150-2000       |
| XML (factor 5) | $1.7 \times 10^{-3}$        | 20-30 | 4-6   | 200-2000       |

Table 24. Carbon preservation at various rates of sediment deposition

| XMI                                                                            | L*0.5                                                                                                                                                                                                                                                                                                    |                                                                                                                                                                                                                                                                                                                                                                                                                                                                                                                                                                                                                                               |  |  |  |  |  |
|--------------------------------------------------------------------------------|----------------------------------------------------------------------------------------------------------------------------------------------------------------------------------------------------------------------------------------------------------------------------------------------------------|-----------------------------------------------------------------------------------------------------------------------------------------------------------------------------------------------------------------------------------------------------------------------------------------------------------------------------------------------------------------------------------------------------------------------------------------------------------------------------------------------------------------------------------------------------------------------------------------------------------------------------------------------|--|--|--|--|--|
|                                                                                | 700                                                                                                                                                                                                                                                                                                      | 6000                                                                                                                                                                                                                                                                                                                                                                                                                                                                                                                                                                                                                                          |  |  |  |  |  |
|                                                                                |                                                                                                                                                                                                                                                                                                          |                                                                                                                                                                                                                                                                                                                                                                                                                                                                                                                                                                                                                                               |  |  |  |  |  |
| Sed rate                                                                       | O 2 pen                                                                                                                                                                                                                                                                                       | O 2 pen                                                                                                                                                                                                                                                                                                                                                                                                                                                                                                                                                                                                                            |  |  |  |  |  |
| cm y -1                                                             | cm                                                                                                                                                                                                                                                                                                       | cm                                                                                                                                                                                                                                                                                                                                                                                                                                                                                                                                                                                                                                            |  |  |  |  |  |
| 1.1x10 -3                                                           | 0.77                                                                                                                                                                                                                                                                                                     | 7.7                                                                                                                                                                                                                                                                                                                                                                                                                                                                                                                                                                                                                                           |  |  |  |  |  |
| 1.7x10 -3                                                           | 1.19                                                                                                                                                                                                                                                                                                     | 11.9                                                                                                                                                                                                                                                                                                                                                                                                                                                                                                                                                                                                                                          |  |  |  |  |  |
| 3.4x10 -3                                                           | 2.38                                                                                                                                                                                                                                                                                                     | 23.8                                                                                                                                                                                                                                                                                                                                                                                                                                                                                                                                                                                                                                          |  |  |  |  |  |
| XML sed rate                                                                   |                                                                                                                                                                                                                                                                                                          |                                                                                                                                                                                                                                                                                                                                                                                                                                                                                                                                                                                                                                               |  |  |  |  |  |
|                                                                                | 400                                                                                                                                                                                                                                                                                                      | 5000                                                                                                                                                                                                                                                                                                                                                                                                                                                                                                                                                                                                                                          |  |  |  |  |  |
|                                                                                |                                                                                                                                                                                                                                                                                                          |                                                                                                                                                                                                                                                                                                                                                                                                                                                                                                                                                                                                                                               |  |  |  |  |  |
| Sed rate                                                                       | O 2 pen                                                                                                                                                                                                                                                                                       | O 2 pen                                                                                                                                                                                                                                                                                                                                                                                                                                                                                                                                                                                                                            |  |  |  |  |  |
| cm y -1                                                             | cm                                                                                                                                                                                                                                                                                                       | cm                                                                                                                                                                                                                                                                                                                                                                                                                                                                                                                                                                                                                                            |  |  |  |  |  |
| 2.2x10 -3                                                           | 0.88                                                                                                                                                                                                                                                                                                     | 11.0                                                                                                                                                                                                                                                                                                                                                                                                                                                                                                                                                                                                                                          |  |  |  |  |  |
| 3.4x10 -3                                                           | 1.36                                                                                                                                                                                                                                                                                                     | 17.0                                                                                                                                                                                                                                                                                                                                                                                                                                                                                                                                                                                                                                          |  |  |  |  |  |
| 6.8x10 -3                                                           | 2.72                                                                                                                                                                                                                                                                                                     | 34.0                                                                                                                                                                                                                                                                                                                                                                                                                                                                                                                                                                                                                                          |  |  |  |  |  |
| XMI                                                                            | _ *10                                                                                                                                                                                                                                                                                                    |                                                                                                                                                                                                                                                                                                                                                                                                                                                                                                                                                                                                                                               |  |  |  |  |  |
|                                                                                | 150                                                                                                                                                                                                                                                                                                      | 2000                                                                                                                                                                                                                                                                                                                                                                                                                                                                                                                                                                                                                                          |  |  |  |  |  |
|                                                                                |                                                                                                                                                                                                                                                                                                          |                                                                                                                                                                                                                                                                                                                                                                                                                                                                                                                                                                                                                                               |  |  |  |  |  |
| Sed rate                                                                       | O 2 pen                                                                                                                                                                                                                                                                                       | O 2 pen                                                                                                                                                                                                                                                                                                                                                                                                                                                                                                                                                                                                                            |  |  |  |  |  |
| cm y -1                                                             | cm                                                                                                                                                                                                                                                                                                       | cm                                                                                                                                                                                                                                                                                                                                                                                                                                                                                                                                                                                                                                            |  |  |  |  |  |
| 2.2x10 -2                                                           | 3.3                                                                                                                                                                                                                                                                                                      | 44                                                                                                                                                                                                                                                                                                                                                                                                                                                                                                                                                                                                                                            |  |  |  |  |  |
| 3.4x10 -2                                                           | 5.1                                                                                                                                                                                                                                                                                                      | 68                                                                                                                                                                                                                                                                                                                                                                                                                                                                                                                                                                                                                                            |  |  |  |  |  |
| 6.8x10 -2                                                           | 10.2                                                                                                                                                                                                                                                                                                     | 136                                                                                                                                                                                                                                                                                                                                                                                                                                                                                                                                                                                                                                           |  |  |  |  |  |
| XML (factor 5)                                                                 |                                                                                                                                                                                                                                                                                                          |                                                                                                                                                                                                                                                                                                                                                                                                                                                                                                                                                                                                                                               |  |  |  |  |  |
|                                                                                |                                                                                                                                                                                                                                                                                                          |                                                                                                                                                                                                                                                                                                                                                                                                                                                                                                                                                                                                                                               |  |  |  |  |  |
|                                                                                | 200                                                                                                                                                                                                                                                                                                      | 2000                                                                                                                                                                                                                                                                                                                                                                                                                                                                                                                                                                                                                                          |  |  |  |  |  |
|                                                                                | 200                                                                                                                                                                                                                                                                                                      | 2000                                                                                                                                                                                                                                                                                                                                                                                                                                                                                                                                                                                                                                          |  |  |  |  |  |
| Sed rate                                                                       | 200
O 2 pen                                                                                                                                                                                                                                                                                | 2000
O 2 pen                                                                                                                                                                                                                                                                                                                                                                                                                                                                                                                                                                                                                    |  |  |  |  |  |
| Sed rate
cm y -1                                                 | 200
O 2 pen
cm                                                                                                                                                                                                                                                                          | 2000
O 2 pen
cm                                                                                                                                                                                                                                                                                                                                                                                                                                                                                                                                                                                                              |  |  |  |  |  |
| Sed rate
$cm y^{-1}$
2.2x10 -3                                | 200
O 2 pen
cm
0.44                                                                                                                                                                                                                                                                  | 2000
O 2 pen
cm
4.42                                                                                                                                                                                                                                                                                                                                                                                                                                                                                                                                                                                                      |  |  |  |  |  |
| Sed rate
cm y -1
2.2x10 -3
3.4x10 -3 | $     \begin{array}{r}       200 \\       \overline{O_2 \text{ pen}} \\       cm \\       0.44 \\       0.68 \\       \end{array} $                                                                                                                                                                      | $\begin{array}{c} 2000 \\ \hline O_2 \text{ pen} \\ \hline cm \\ \hline 4.42 \\ \hline 6.8 \end{array}$                                                                                                                                                                                                                                                                                                                                                                                                                                                                                                                                       |  |  |  |  |  |
|                                                                                | XMI         Sed rate $cm y^{-1}$ $1.1x10^{-3}$ $1.7x10^{-3}$ $3.4x10^{-3}$ XML s         Sed rate $cm y^{-1}$ $2.2x10^{-3}$ $3.4x10^{-3}$ $6.8x10^{-3}$ Sed rate $cm y^{-1}$ $2.2x10^{-2}$ $3.4x10^{-2}$ $8x10^{-2}$ $3.4x10^{-2}$ $3.4x10^{-2}$ $3.4x10^{-2}$ $3.4x10^{-2}$ $3.4x10^{-2}$ $3.4x10^{-2}$ | XML*0.5         700         Sed rate $O_2$ pen         cm y -1 cm         1.1x10 -3 0.77         1.7x10 -3 1.19         3.4x10 -3 2.38         XML sed rate       400         Sed rate       0.2 pen         cm y -1 cm         2.2x10 -3 0.88         3.4x10 -3 1.36         6.8x10 -3 2.72         XML *10       150         Sed rate $O_2$ pen         cm y -1 cm         2.2x10 -2 3.3         3.4x10 -2 5.1         6.8x10 -2 5.1         6.8x10 -2 10.2         XML (factor 5)       5.1 |  |  |  |  |  |

Table 32. Linear sedimentation rates and oxygen penetration depths (O2 pen) for the different mass fluxes explored in our modelling.

| XML*0.5                                              |              |              |                                 |              |  |  |  |
|------------------------------------------------------|--------------|--------------|---------------------------------|--------------|--|--|--|
| Oxygen exposure (y)                                  | 700          | 6000         |                                 |              |  |  |  |
| Carbon preservation (%)                              | 1.2          | 1.2          | -                               |              |  |  |  |
| $O_2$ -flux (mmol cm -2 y -1 ) | 0.019        | 0.019        |                                 |              |  |  |  |
|                                                      |              |              | -                               |              |  |  |  |
| Porosity ( $\phi$ )                                  | $[O_2]_{BW}$ | $[O_2]_{BW}$ | -                               |              |  |  |  |
|                                                      | μΜ           | μΜ           |                                 |              |  |  |  |
| 0.7                                                  | 19           | 160          |                                 |              |  |  |  |
| 0.8                                                  | 19           | 170          |                                 |              |  |  |  |
| 0.9                                                  | 27           | 230          |                                 |              |  |  |  |
|                                                      | XML          |              |                                 |              |  |  |  |
| Oxygen exposure (y)                                  | 400          | 5000         | 400                             | 5000         |  |  |  |
| Carbon preservation (%)                              | 2            | 2            | 3                               | 3            |  |  |  |
| $O_2$ -flux (mmol cm -2 y -1 ) | 0.029        | 0.029        | 0.015                           | 0.015        |  |  |  |
|                                                      | 10.1         | 10.1         |                                 |              |  |  |  |
| Porosity (\$)                                        | $[O_2]_{BW}$ | $[O_2]_{BW}$ | $[O_2]_{BW}$                    | $[O_2]_{BW}$ |  |  |  |
|                                                      | μΜ           | μM           | μΜ                              | μM           |  |  |  |
| 0.7                                                  | 35           | 440          | 18                              | 230          |  |  |  |
| 0.8                                                  | 36           | 450          | 19                              | 230          |  |  |  |
| 0.9                                                  | 51           | 630          | 26                              | 330          |  |  |  |
|                                                      | XML*1        | 0            |                                 |              |  |  |  |
| Oxygen exposure (y)                                  | 150          | 2000         | 150                             | 2000         |  |  |  |
| Carbon preservation (%)                              | 4            | 8            | 4                               | 8            |  |  |  |
| $O_2$ -flux (mmol cm -2 y -1 ) | 0.11         | 0.11         | 0.053                           | 0.053        |  |  |  |
|                                                      |              |              |                                 |              |  |  |  |
| Porosity ( $\phi$ )                                  | $[O_2]_{BW}$ | $[O_2]_{BW}$ | $[O_2]_{BW}$                    | $[O_2]_{BW}$ |  |  |  |
|                                                      | μΜ           | μΜ           | μΜ                              | μΜ           |  |  |  |
| 0.7                                                  | 500          | 6600         | 240                             | 3200         |  |  |  |
| 0.8                                                  | 510          | 6800         | 250                             | 3300         |  |  |  |
| 0.9                                                  | 720          | 9600         | 350                             | 4600         |  |  |  |
|                                                      |              |              |                                 |              |  |  |  |
|                                                      | XML (facto   | or 5)        |                                 |              |  |  |  |
| Oxygen exposure (y)                                  | 200          | 2000         | 200                             | 2000         |  |  |  |
| Carbon preservation (%)                              | 4            | 4            | 6                               | 6            |  |  |  |
| $O_2$ -flux (mmol cm -2 y -1 ) | 0.011        | 0.011        | 0.073                           | 0.073        |  |  |  |
|                                                      |              | 10.1         | 10.1                            |              |  |  |  |
| Porosity ( $\phi$ )                                  | $[O_2]_{BW}$ | $[O_2]_{BW}$ | [O 2 ] BW | $[O_2]_{BW}$ |  |  |  |
|                                                      | μM           | μM           | μΜ                              | μM           |  |  |  |
| 0.7                                                  |              |              | 4.4                             |              |  |  |  |
|                                                      | 6.6          | 66           | (10.1-15.7) a        | 44           |  |  |  |
| 0.8                                                  | 6.8          | 68           | 4.5                             | 45           |  |  |  |
| 0.9                                                  | 9.6          | 96           | 6.4                             | 64           |  |  |  |

Table 4. Calculations of  $[O_2]_{BW}$  ( $\mu M$ ) for our different assumptions of sedimentation rate (lowest value for each sedimentation rate in **red**).

avalues in parenthesis after considering diffusion through the benthic boundary layer

**Figure Captions:**

Figure 1. General stratigraphy for the upper 4 units of the Xiamaling Formation (abstracted from Zhang et al., 2015) with a more detailed stratigraphy for the upper 45 meters of unit 1.

Figure 2<del>1</del>. Total organic carbon (TOC), HI,  $\delta^{13}$ C (relative to PDB) and metal data (Mo/Al, V/Al and U/Al, F-e/Al for unit 1 of the Xiamaling Formation. T; where the dashed line represents upper crust values from Rudnick (2004).

**, for unit 1 of the Xiamaling Formation.**

Figure 32. a) TOC vs the ratio of highly reactive to total iron (FeHR/FeT) from fresh core material in uUnit 1 of the Xiamaling Formation. The horizontal dashed line represents a FeHR/FeT of 0.38. The range of TOC values for green-gray shales from outcrop samples is shown in the green rectangular field, while the range in values for the black shales from outcrop is shown in the gray field., while the vertical dashed line indicates a TOC of 1 wt%. The green-gray shales are predominantly represented at TOC < 1 wt% and the black shales with TOC > 1 wt% (see Figure 1). b) V/Al vs FeHR/FeT, with the horizontal line as in a) and the vertical line the V/Al crustal average {Rudnick, 2004 #8862}. be) TOC vs HI for outcrop material, with black and green-gray shales separately indicated. The red dots mark the averages for the black and green-gray shale groups.

Figure 43. a) Preservation of organic carbon in modern marine sediments calculated as % of carbon buried in a sediment compared to the % deposited to the sediment surface. Redrafted from Canfield (1994). The vertical lines represent the different modelled sedimentation rates. The upper red rectangles highlight the carbon preservation for the anoxic environments in the compilation, while the lower blue rectangles are 10 times less than this, representing the

estimated range of carbon preservation in Xiamaling oxic sediments. For the XML sedimentation rate a blue rectangle at 5 times less carbon preservation is also shown, b) Oxygen exposure time versus organic carbon preservation in marine sediments. The horizontal blue boxes reflect the range of oxic sediment carbon preservation at the different sedimentation rates from used in the modelling (see Figure 4a-a), while the dark blue fields outline the range of associated oxygen exposure times. Oxygen exposure time data summarized in Table S4., -c) oxygen penetration depth versus oxygen uptake rate from modern marine sediments with variable bottom-water oxygen concentrations. Data is from Table S54. Black line indicates the best power-function fit to the data. Red lines indicate fits from Equation- 5 to the data at different bottom water oxygen concentrations and A=4. Green line represents model fit from Equation- 5 with A=2 and 250 μM O2.